# Trifunctional cross-linker for mapping protein-protein interaction networks and comparing protein conformational states

Dan Tan[1,2†], Qiang Li[2,3,4,5†], Mei-Jun Zhang[2], Chao Liu[6], Chengying Ma[7], Pan Zhang[1,2], Yue-He Ding[1,2], Sheng-Bo Fan[6], Li Tao[1,2], Bing Yang[2], Xiangke Li[2], Shoucai Ma[2], Junjie Liu[7], Boya Feng[7], Xiaohui Liu[2], Hong-Wei Wang[7], Si-Min He[6], Ning Gao[7], Keqiong Ye[2], Meng-Qiu Dong[1,2*], Xiaoguang Lei[2,3,4,5*]

[1]Graduate Program, Peking Union Medical College, Chinese Academy of Medical Sciences, Beijing, China; [2]National Institute of Biological Sciences, Beijing, China; [3]Synthetic and Functional Biomolecules Center, Peking University, Beijing, China; [4]Peking-Tsinghua Center for Life Sciences, Peking University, Beijing, China; [5]Department of Chemical Biology, College of Chemistry and Molecular Engineering, Peking University, Beijing, China; [6]Key Lab of Intelligent Information Processing of Chinese Academy of Sciences, Institute of Computing Technology, Chinese Academy of Sciences, Beijing, China; [7]Ministry of Education Key Laboratory of Protein Sciences, School of Life Sciences, Tsinghua University, Beijing, China

**Abstract** To improve chemical cross-linking of proteins coupled with mass spectrometry (CXMS), we developed a lysine-targeted enrichable cross-linker containing a biotin tag for affinity purification, a chemical cleavage site to separate cross-linked peptides away from biotin after enrichment, and a spacer arm that can be labeled with stable isotopes for quantitation. By locating the flexible proteins on the surface of 70S ribosome, we show that this trifunctional cross-linker is effective at attaining structural information not easily attainable by crystallography and electron microscopy. From a crude Rrp46 immunoprecipitate, it helped identify two direct binding partners of Rrp46 and 15 protein-protein interactions (PPIs) among the co-immunoprecipitated exosome subunits. Applying it to *E. coli* and *C. elegans* lysates, we identified 3130 and 893 inter-linked lysine pairs, representing 677 and 121 PPIs. Using a quantitative CXMS workflow we demonstrate that it can reveal changes in the reactivity of lysine residues due to protein-nucleic acid interaction.

**\*For correspondence:**
dongmengqiu@nibs.ac.cn (M-QD);
xglei@pku.edu.cn (XL)

[†]These authors contributed equally to this work

**Competing interests:** The author declares that no competing interests exist.

## Introduction

Proteins execute diverse functions by interacting with multiple protein partners in different complexes. The study of protein complex structures and protein-protein interactions is critical for understanding their functions. Recently, chemical cross-linking of proteins coupled with mass spectrometry analysis (CXMS) has emerged as a powerful tool for the analysis of such structures and interactions (*Sinz, 2006*; *Leitner et al., 2010*; *Petrotchenko and Borchers, 2010*; *Singh et al., 2010*; *Rappsilber, 2011*; *Bruce, 2012*). CXMS methods are less time-consuming and less demanding of sample purity than are traditional methods; this technology has thus been increasing in popularity.

Recent progress in the development of analytical instruments, cross-linking reagents, and software has catapulted CXMS from obscurity to prominence, as witnessed by an explosion of successful applications (*Bohn et al., 2010*; *Chen et al., 2010*; *Kao et al., 2011*; *Lauber and Reilly, 2011*;

**eLife digest** Proteins fold into structures that are determined by the order of the amino acids that they are built from. These structures enable the protein to carry out its role, which often involves interacting with other proteins. Chemical cross-linking coupled with mass spectrometry (CXMS) is a powerful method used to study protein structure and how proteins interact, with a benefit of stabilizing and capturing brief interactions.

CXMS uses a chemical compound called a linker that has two arms, each of which can bind specific amino acids in a protein or in multiple proteins. Only when the regions are close to each other can they be "cross-linked" in this way. After cross-linking, the proteins are cut into small pieces known as peptides. The cross-linked peptides are then separated from the non cross-linked ones and characterized.

Although CXMS is a popular method, there are aspects about it that limit its use. It does not work well on complex samples that contain lots of different proteins, as it is difficult to separate the cross-linked peptides from the overwhelming amounts of non cross-linked peptides. Also, although it can be used to detect changes in the shape of a protein, which are often crucial to the protein's role, the method has not been smoothed out.

Tan, Li et al. have now developed a new cross-linker called Leiker that addresses these limitations. Leiker cross-links the amino acid lysine to another lysine, and contains a molecular tag that allows cross-linked peptides to be efficiently purified away from non cross-linked peptides. As part of a streamlined workflow to detect changes in the shape of a protein, Leiker also contains a region that can be labeled.

Analysing a bacterial ribosome, which contains more than 50 proteins, showed that Leiker-based CXMS could detect many more protein interactions than previous studies had. These included interactions that changed too rapidly to be studied by other structural methods. Tan, Li et al. then applied Leiker-based CXMS to the entire contents of bacterial cells at different stages of growth, and identified a protein interaction that is only found in growing cells.

In future, Leiker will be useful for analyzing the structure of large protein complexes, probing changes in protein structure, and mapping the interactions between proteins in complex mixtures.

*Herzog et al., 2012*; *Jennebach et al., 2012*; *Kalisman et al., 2012*; *Kao et al., 2012*; *Leitner et al., 2012*; *Bui et al., 2013*; *Murakami et al., 2013*; *Tosi et al., 2013*). However, CXMS is still limited by sample complexity and by low abundances of cross-linked peptides. Extensive fractionation is often required to reduce the complexity of samples that contain macromolecular complexes (*Chen et al., 2010*; *Lauber and Reilly, 2011*; *Jennebach et al., 2012*; *Kalisman et al., 2012*; *Kao et al., 2012*; *Murakami et al., 2013*; *Tosi et al., 2013*). The identification of cross-linked peptides in more heterogeneous samples such as crude immunoprecipitates and whole-cell lysates is even more difficult (*Rinner et al., 2008*; *Luo et al., 2012*; *Yang et al., 2012*; *Liu et al., 2015*).

Given the sparsity of cross-linked peptides in samples, it would be beneficial to purify them from complex mixtures using affinity tags after cross-linking. However, despite increased efforts to develop chemical cross-linkers with enrichment functions (*Luo et al., 2012*; *Trester-Zedlitz et al., 2003*; *Fujii et al., 2004*; *Chowdhury et al., 2006*; *Chu et al., 2006*; *Chowdhury et al., 2009*; *Kang et al., 2009*; *Nessen et al., 2009*; *Yan et al., 2009*; *Vellucci et al., 2010*; *Petrotchenko et al., 2011*; *Sohn et al., 2012*; *Kaake et al., 2014*), few such agents have been shown to improve identification capabilities in complex samples. Two exceptions include Azide-A-DSBSO, which is used with biarylazacyclooctynone (*Kaake et al., 2014*), and the protein interaction reporter (PIR) (*Chavez et al., 2013*; *Weisbrod et al., 2013*). However, special instrument control is recommended for their application (*Chavez et al., 2013*; *Weisbrod et al., 2013*).

In this work, we developed a series of chemical cross-linkers with a modular design as pioneered previously (*Trester-Zedlitz et al., 2003*). They each contain a biotin tag for affinity purification and a cleavage site that can be used to release cross-linked peptides from streptavidin beads. We selected the cross-linker with the best performance and developed a robust enrichment protocol with >97% enrichment efficiency. We termed it Lysine-targeted enrichable cross-linker (Leiker). Using our

previously developed pLink identification software (*Yang et al., 2012*), we here demonstrate that the use of Leiker effectively facilitates CXMS analysis in a variety of sample types, from purified complexes, crude immunoprecipitates, to highly complex whole-cell lysates.

Quantification of cross-linker modified peptides has the potential to detect protein conformational changes and changes in molecular interactions, though these methods are not mature. To address this potentially critical application of our technology, we synthesized stable isotope-labeled Leiker. Also, we established an automated data analysis workflow for the relative quantitation of light and heavy Leiker cross-links. As a proof of concept, we carried out a quantitative CXMS analysis of an RNA-binding protein L7Ae. Using deuterium-labeled Leiker, we found that for the three L7Ae lysine residues that are buried upon RNA binding, their mono-links decreased dramatically in the presence of RNA, exactly as expected. We further extended the application of quantitative CXMS to a highly complex system consisting of log-phase and stationary-phase *E. coli* cells and identified a growth phase specific protein interaction.

## Results

### Design, synthesis and evaluation of Leiker

We aimed to develop a cross-linker similar to the widely used BS$^3$ but that had two major advantages: first, a biotin tag for affinity purification of cross-linked peptides, and second, a cleavage site to release cross-linked peptides after enrichment on streptavidin beads without carrying the biotin group; biotin can interfere with subsequent LC-MS/MS analysis. After experimenting with different designs of Leiker (*Figure 1*, *Figure 1—figure supplements 1–6*, and Appendix), we found that bAL1 and bAL2 worked the best and there was no difference in performance between these two (*Figure 1—figure supplement 5*). Hereafter, Leiker refers to either bAL1 or bAL2. In this study, bAL2 was used in most of the experiments and a bAL2-based CXMS workflow is illustrated in *Figure 2*. Both bAL1 and bAL2 feature a one-piece design with an azobenzene-based chemical cleavage site (*Yang et al., 2010*) and a 9.3-Å carbon chain that connects two sulfo-NHS esters. This spacer arm is shorter than that of BS$^3$ (11.4 Å), so it may confer a higher specificity to Leiker in capturing protein-protein interactions. Inter-, loop-, and mono-linked peptides generated by either all produce a reporter ion of *m/z* 122.0606 in higher-energy collisional dissociation (HCD) spectra (*Figure 2D*). It can be used to verify the identification of Leiker-cross-linked peptides. For quantitative CXMS analysis, we synthesized isotope-labeled bAL2 in which six hydrogen atoms in the spacer arm were replaced with deuterium (*Figure 1* and *Figure 1—figure supplement 6*). The six-dalton difference was sufficient to separate peptides cross-linked by [d$_0$]-Leiker from the same peptides cross-linked by [d$_6$]-Leiker.

### Leiker enabled robust enrichment of cross-linked peptides

To assess to what extent Leiker could improve the identification of low-abundance cross-linked peptides from a complex background, a mixture of ten standard proteins (*Figure 3—source data 1*), consisting of RNase A, lysozyme, PUD-1/PUD-2 heterodimer, GST, aldolase, BSA, lactoferrin, β-galactosidase, mouse monoclonal antibody, and myosin, was treated with Leiker, digested with trypsin, and then diluted or not with a tryptic digest of non-cross-linked *E. coli* lysates at different ratios (1:1, 1:10, and 1:100, w/w). The peptide mixture was then incubated with streptavidin agarose. After extensive washes, cross-linked peptides were released using the Na$_2$S$_2$O$_4$ elution buffer. BS$^3$ was used in parallel as a control. As is shown in *Figure 3A* the number of BS$^3$-linked peptide pairs identified decreased dramatically, from 109 in the undiluted sample to only one in the 100-fold diluted sample. The number of Leiker-linked peptide pairs identified after enrichment was in no way affected by increasing background complexity, with >160 inter-links detected in each sample. To be noted, these inter-linked peptide pairs, or inter-links for abbreviation, can result from either intra-protein or inter-protein cross-linking (illustrated in *Figure 2B*). Strikingly, cross-linking products, including inter-, loop- and mono-links (*Figure 2C*) constituted over 97% of all peptides identified post enrichment (*Figure 3A*). Of the Leiker-linked lysine pairs that can be mapped to the pdb structures (*Figure 3—source data 1*), 82% have Cα – Cα distance ≤22 Å and 93% have Cα – Cα distance ≤30 Å (FDR < 5%, E-value < 0.01), which is comparable to BS$^3$ (*Figure 3—figure supplement 1*). This result demonstrated that Leiker enables effective enrichment of cross-linked peptides.

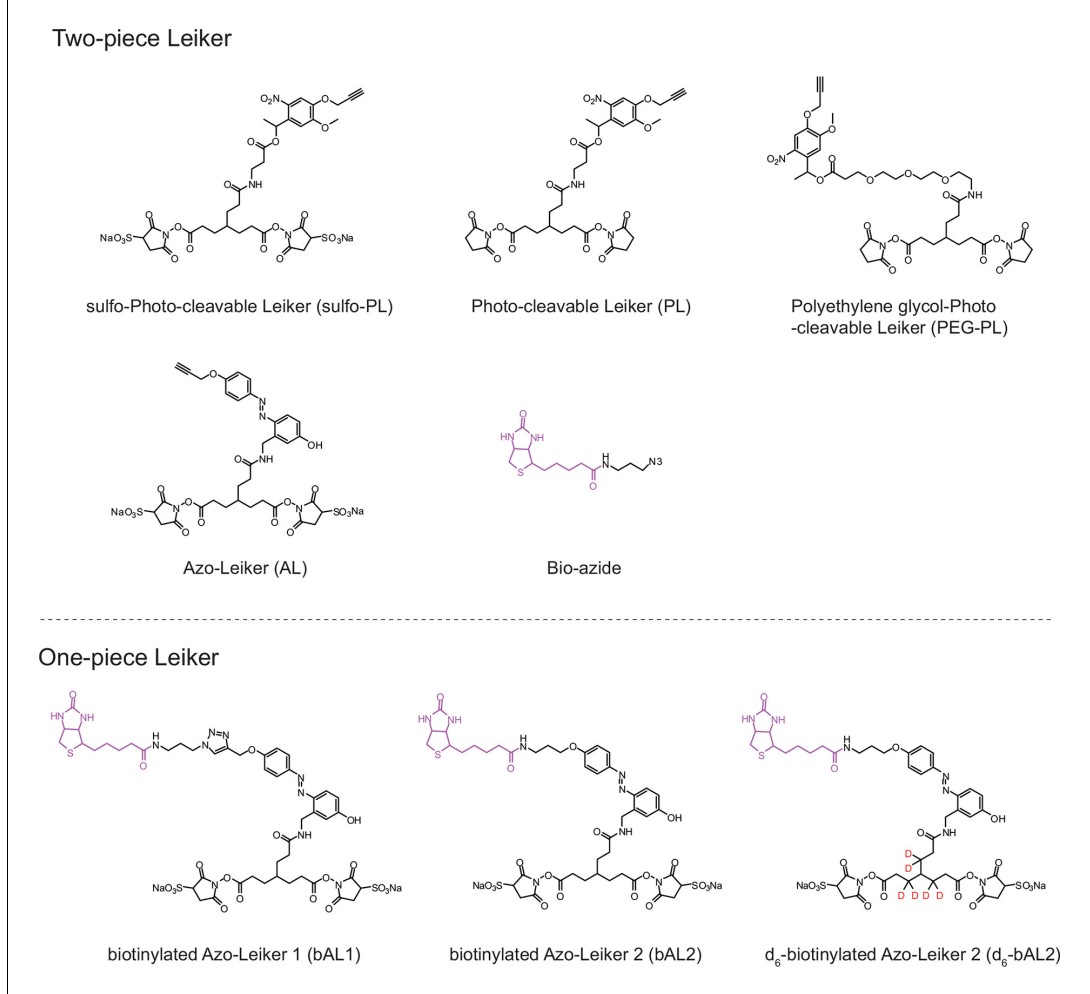

**Figure 1.** Chemical structures of different designs of Leiker. The top panel shows four designs of two-piece Leiker with a photo-cleavage site (sulfo-PL, PL, and PEG-PL) or an azobenzene-based cleavage site (AL). Biotin is attached via click chemistry by reacting with bio-aizde. The bottom panel shows two unlabeled (bAL1, bAL2) and deuterium-labeled ([$d_6$]-bAL2) one-piece Leiker molecules. The biotin moiety is colored magenta.

The following figure supplements are available for figure 1:

**Figure supplement 1.** Optimization of protein-to-cross-linker ratio (w/w) for (A) sulfo-PL, (B) AL, (C) bAL1, and (D) bAL2.

**Figure supplement 2.** Evaluation of azobenzene-based chemical cleavage.

**Figure supplement 3.** The one-piece Leiker (bAL1) outperformed the two-piece Leiker (AL) in the CXMS analysis of a mixture of ten standard proteins.

**Figure supplement 4.** Evaluation of the two piece Azo-Leiker (AL).

**Figure supplement 5.** bAL1 and bAL2 performed similarly.

**Figure supplement 6.** MS1 spectra of (A) [$d_0$]-bAL2 and (B) [$d_6$]-bAL2.

The ten standard proteins also allowed us to assess the specificity of Leiker. Because Leiker has more functional groups than BS[3] does, a concern arises that Leiker may produce more cross-linking artifacts. Cross-links between non-interacting proteins are surely artifacts, which include all the inter-protein cross-links identified from the ten-protein mixture except those between the light-chain and the heavy-chain of myosin, between the light-chain and the heavy-chain of an IgG antibody, and

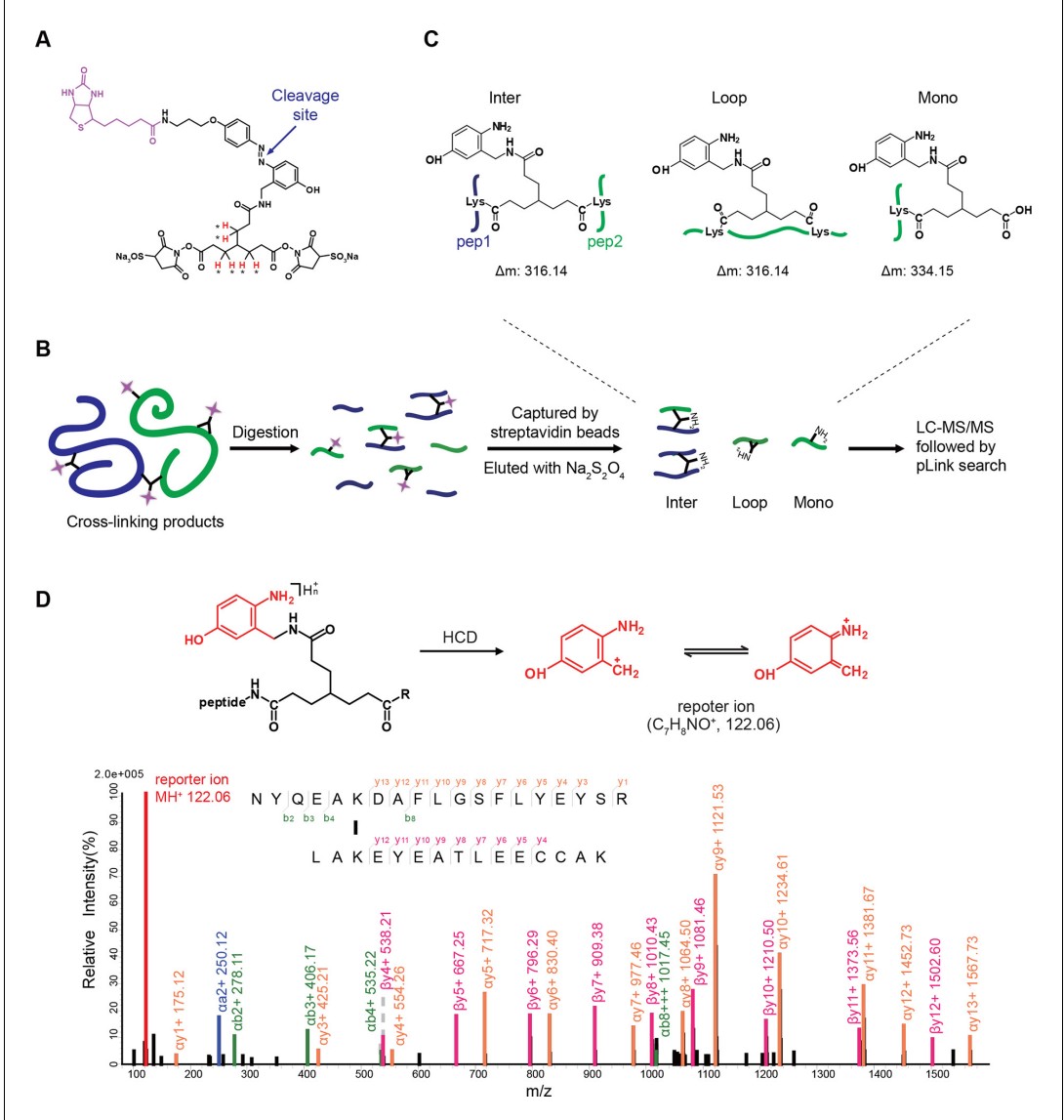

**Figure 2.** Scheme of the Leiker-based CXMS workflow. (A) Leiker contains a biotin moiety (magenta), a cleavage site (arrows), and six hydrogen atoms that are accessible to isotope labeling (asterisks). (B) The workflow for purification of Leiker-linked peptides. (C) Three types of Leiker-linked peptides. (D) Leiker-linked peptides generate a reporter ion of 122.06 m/z in HCD, as shown in the spectrum of an inter-linked peptide NYQEAKDAFLGSFLYEYSR-LAKEYEATLEECCAK (+4 charged, MH[+] 4433.0553), in which C denotes carbamidomethylated cysteine.

between PUD-1 and PUD-2, which form a heterodimer. We found that the percentage of artifactual cross-links is 3% for both Leiker and BS[3] (*Figure 3—source data 2*), fitting with the filtering criteria that were applied (FDR cutoff 0.05 followed by E-value cutoff 0.01). The results demonstrate that Leiker is as specific as BS[3].

Further, we cross-linked highly complex *E. coli* lysates with either Leiker or BS[3] for a side-by-side comparison. After enrichment and a single reverse phase LC-MS/MS analysis, Leiker yielded at least a fourfold increase in the number of inter-links identified (*Figure 3B*).

## Application of Leiker to large protein assemblies and immunoprecipitates

Next, we applied Leiker to real-world samples, starting with purified *E. coli* 70S ribosome, a 2.5 MDa ribonucleoprotein (RNP) complex consisting of more than 50 proteins. A total of 222 inter-

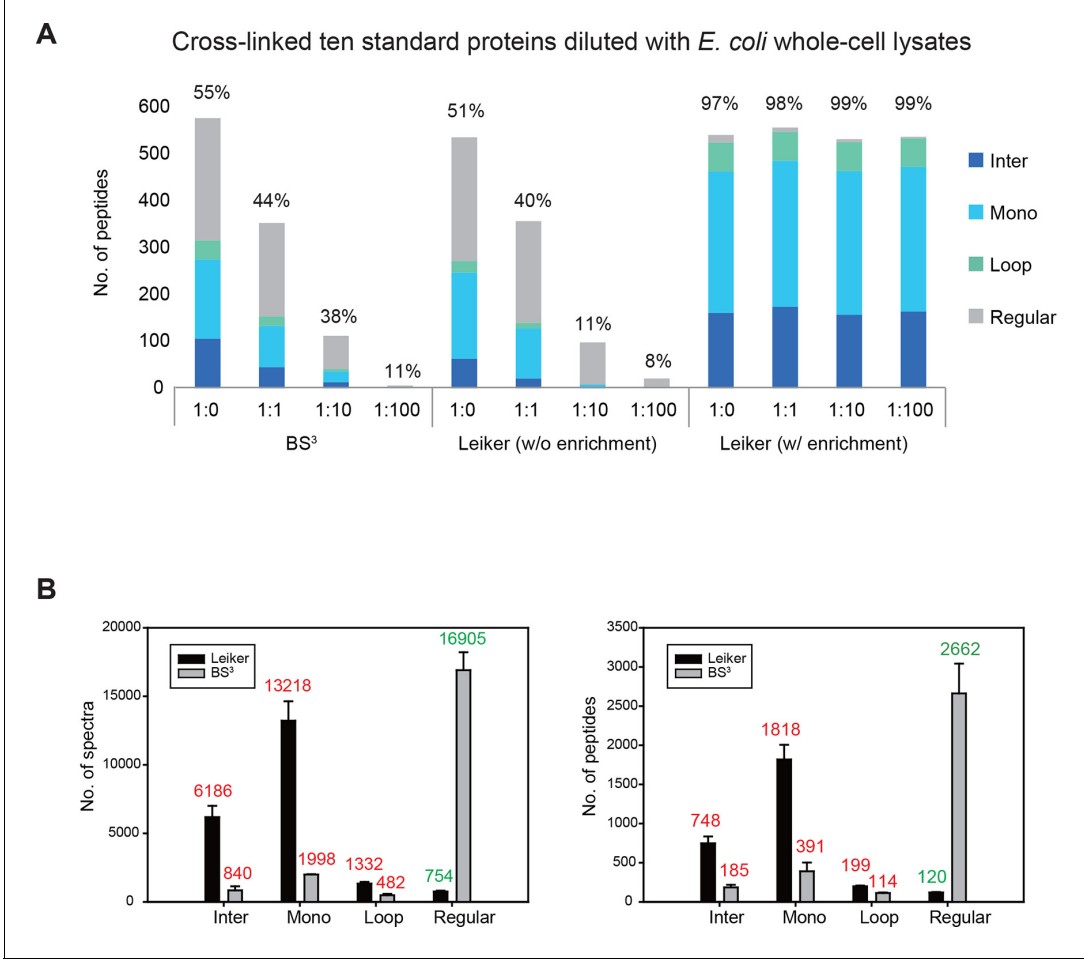

**Figure 3.** Evaluating the performance of Leiker. (**A**) Leiker allowed near 100% enrichment of target peptides from a cross-linked ten-protein mixture diluted with increasing amounts of non-cross-linked *E. coli* lysates. Dark blue, inter-links; light blue, mono-links; green, loop-links; grey, regular peptides not modified by Leiker. (**B**) Number of cross-link identifications from *E. coli* lysates treated with Leiker or BS³. Shown in the left and right panels are the identified spectra and peptides, respectively.

The following source data and figure supplement are available for figure 3:

**Source data 1.** Ten standard proteins used to evaluate Leiker, mixed at equal amounts by mass.

**Source data 2.** Summary of identified spectra from the ten-protein mixture.

**Figure supplement 1.** Distance distributions of cross-linked lysine pairs in the undiluted ten-protein mixture.

linked lysine pairs were identified with high confidence, including 95 inter-molecular and 127 intra-molecular cross-links (*Figure 4—source data 1*). This is three times as many as in a previous study (*Lauber and Reilly, 2011*). Of the 95 cross-links connecting two lysine residues that are both present in the crystal structure of a 70S ribosome (*Fischer et al., 2015*) (PDB code: 5AFI), 75% are compatible with the crystal structure with a Cα-Cα distance ≤22 Å, which is the length of the spacer arm of Leiker plus two lysine side chains. Among the subset of intra-molecular cross-links, 84% have Cα-Cα distances ≤22 Å; among the subset of inter-molecular cross-links, 50% have Cα-Cα distances ≤22 Å and 73% have Cα-Cα distances ≤30 Å, which could be a reasonable cutoff considering conformation flexibility of proteins in solution (*Figure 4—source data 1* and *Figure 4—figure supplement 1*). One particular ribosomal protein L9 is a good example to illustrate conformational flexibility and the dynamic nature of interactions between proteins or protein complexes. A large *b*-factor in the crystal structure has suggested that L9 is highly mobile. It has been observed to adopt an extended, rod-

like conformation in the crystal structure (*Schuwirth et al., 2005*) and a strikingly different bent conformation in the solution structure of the ribosome determined using cryo-EM (*Fischer et al., 2015*; *Seidelt et al., 2009*). Bending of L9 was echoed in this study, as reflected in the cross-links bridging L9 and L2 and the cross-links bridging the two termini of L9 (*Figure 4—figure supplement 2*). Three additional cross-links involving L9 have Cα-Cα distances >50 Å if measured within a ribosomal particle (*Figure 4—source data 1*). We propose these apparently long distance cross-links, which are similar to the ones observed in a previous CXMS study (*Lauber and Reilly, 2011*), reflect interactions between ribosomal particles. L9 locates at the interface between ribosomal particles in higher-order configurations (*e.g.* polysome) (*Brandt et al., 2009*). Dimerization or oligomerization of 70S ribosomes in the absence of mRNA was also observed using negative staining EM from highly purified non-cross-linked 70S ribosomes (*Figure 4—figure supplement 3*).

The peripheral regions of the ribosome are critical for protein translation and regulation (*Savelsbergh et al., 2000*; *Valle et al., 2003*; *Kothe et al., 2004*). However, despite of extensive studies on the ribosome structures, these peripheral parts are still largely missing because they are either too dynamic or refractory to crystallography. For *E. coli* ribosomal proteins currently lacking well-defined coordinates in the 70S crystal structures, Leiker-based CXMS provided remarkably more linkages than other ribosomal proteins. The top four proteins with the most inter-molecular cross-links identified are S1, L1, L7/12, and L31, all of which are mobile components in the peripheral regions and often invisible in the crystal structure (*Figure 4A* and *Figure 4—source data 2*). S1 is the largest ribosomal protein, which binds to mRNA and initiates translation, but has no high-resolution structures available either alone or in the context of the 70S ribosome (*Fischer et al., 2015*; *Lauber et al., 2012*). A previous CXMS analysis of the 30S subunit revealed interaction between S1 and a region near the 3' end of 16S rRNA (*Lauber et al., 2012*), but it is unknown whether or not S1 interacts with the 50S subunit. Our analysis of the 70S ribosome revealed extensive contacts between the C-terminal mRNA binding domain of S1 and L1 in the 50S subunit (*Figure 4—figure supplement 4A*). Since the two of them localize to a region where both tRNA and mRNA leave the ribosome, the observation of six cross-links between them hints that there might be a coordination between deacylated tRNA release and mRNA exit from the ribosome. The 30S proteins that were found to interact with S1 in this study were largely consistent with those identified in the previous study (*Lauber et al., 2012*). In particular, four cross-links were identified between the N-terminal peptide of S1 (M1-K14) and the N-terminal peptide of S2 (M1-K11) (*Figure 4—source data 1* and *Figure 4—figure supplement 4A*). This result agrees perfectly with a recent structural finding on the direct interaction between S1 and S2 (*Byrgazov et al., 2015*). L1 had the highest number of cross-links with S1, followed by cross-links with L33, L5, L9, S13, and S2 (*Figure 4—figure supplement 4B*). The proximity of L1 to L33 and L5 implicates a rotated conformation of L1 in the sample (*Figure 4—source data 3*), which was repeatedly observed in various structures of the 70S ribosome in different functional states (*Valle et al., 2003*). Furthermore, beyond the expected interactions between L7/12 and L6, L10, or L11 (*Diaconu et al., 2005*), we also found novel interactions between L7/12 and L19 or S3 (*Figure 4A* and *Figure 4—figure supplement 4C*). These findings suggest that the highly flexible L7/12 stalk might be able to contact the 30S subunit, given the predicted large length of this dynamic stalk (*Diaconu et al., 2005*). Nine cross-links between *E. coli* L31 and L5 placed L31 in the central protuberance region (*Figure 4—figure supplement 4D*), which is supported by the crystal structure of *T. thermophilus* 70S ribosome (*Voorhees et al., 2009*) and the newly revealed structure of 70S ribosome (*Fischer et al., 2015*). Together, these results demonstrate that Leiker-based CXMS analysis can provide structural information that is highly complementary to crystallography and cryo-EM, especially for the flexible or dynamic regions that cannot be deduced using traditional methods.

Combining CXMS and immunoprecipitation (IP) has great potential for the detection of binding partners in close proximity among co-immunoprecipitated proteins; such method may be widely adopted in biology laboratories. Much progress has been made recently in this area by the use of a modified anti-GFP single-chain antibody that cannot be cross-linked so that GFP-tagged protein complexes can be cross-linked on beads and separated away from the antibody for CXMS analysis (*Shi et al., 2015*). For highly heterogeneous IP samples, however, cross-linked peptides can be inundated by non-cross-linked peptides even if the antibody is removed from the background. As a test, we prepared a crude immunoprecipitate of a TAP-tagged yeast exosome subunit Rrp46 (*Figure 4—figure supplement 5*), from which 740 proteins were identified at 0.1% protein FDR. The

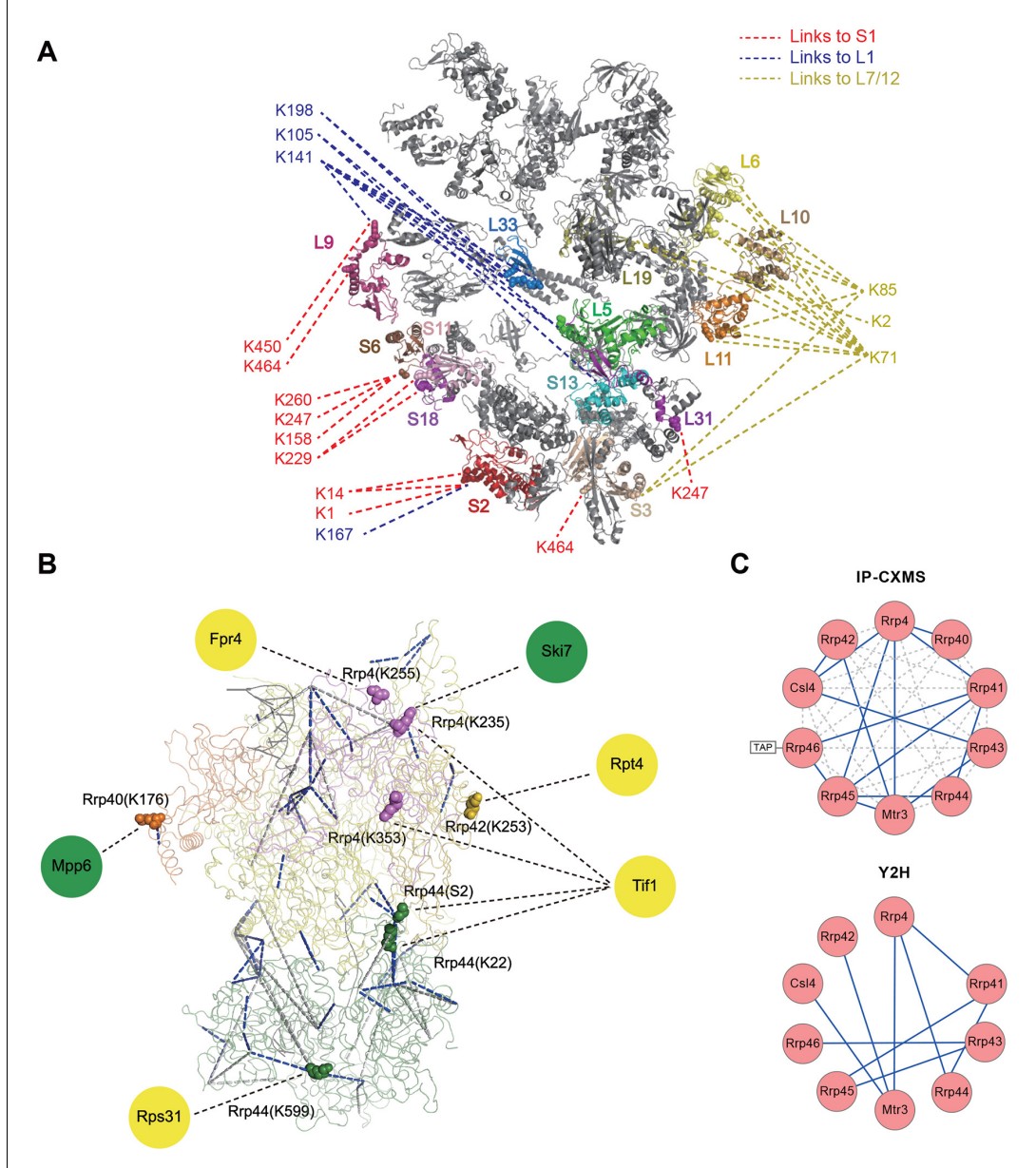

**Figure 4.** Leiker-based CXMS analyses of large protein assemblies. (**A**) Analysis of a purified *E. coli* 70S ribosome revealed the locations of highly dynamic periphery ribosomal proteins S1, L1, and L7/12 that were refractory to crystallography and cryo-EM analysis. Cross-links to S1, L1, and L7/12 are colored red, blue, and yellow, respectively, and the cross-linked residues on these three proteins are numbered according to the Uniprot sequences. (**B**) Analysis of a crude immunoprecipitate of the yeast exosome complex. Dashed blue and grey lines denote 50 compatible and 22 incompatible cross-links, respectively, according to the structure of the RNA-bound 11-subunit exosome complex (PDB code: 4IFD). Rrp44, green; Rrp40, orange; Rrp4, violet; Rrp42, gold; other exosome subunits, yellow; RNA, black. Known and candidate exosome regulators revealed by Leiker-cross-links are shown along the periphery and highlighted in green and yellow circles, respectively. (**C**) Connectivity maps of the ten-subunit exosome core complex based on the inter-molecular cross-links identified in the current IP-CXMS experiments or on previous yeast two-hybrid (Y2H) studies (*Stark et al., 2006*; *Uetz et al., 2000*; *Oliveira et al., 2002*; *Luz et al., 2007*; *Yu et al., 2008*). Blue solid lines: experimentally identified putative direct protein-protein interactions; grey dashed lines: theoretical cross-links according to the crystal structure; Cα-Cα distance cutoff ≤30 Å.

The following source data and figure supplements are available for figure 4:

**Source data 1.** CXMS analysis of *E. coli* 70S ribosomes.

**Source data 2.** Number of cross-linked lysine pairs classified by ribosomal proteins.

*Figure 4 continued on next page*

*Figure 4 continued*

**Source data 3.** Identified cross-linked lysine pairs involving L1.
**Source data 4.** CXMS analysis of the *Saccharomyces cerevisiae* exosome complex.
**Figure supplement 1.** Distance distribution of the inter-molecular and intra-molecular cross-links identified in 70S ribosomes.
**Figure supplement 2.** Alignment of L9 and L2 from the crystal structure (L9, orange; L2, wheat; PDB code: 2AW4) and their counterparts from the cryo-EM reconstruction (L9, blue; L2, lightblue; PDB code: 5AFI).
**Figure supplement 3.** Negative staining of non-cross-linked *E. coli* 70S ribosome.
**Figure supplement 4.** Connectivity maps of cross-links involving (**A**) S1, (**B**) L1, (**C**) L7/12, and (**D**) L31.
**Figure supplement 5.** Silver-stained SDS-PAGE gel of the crude immunoprecipitate of TAP-tagged Rrp46.
**Figure supplement 6.** Number of identified inter-linked peptide pairs from decreasing amount of Leiker-cross-linked exosome immunoprecipitate (FDR < 0.05, E-value < 0.01).

immunoprecipitated proteins were eluted off IgG beads and cross-linked with Leiker. To evaluate the sensitivity of the method, we varied the amount of immunoprecipitates from 40 µg to 3 µg of proteins and found that the number of inter-link identifications did not change much as the input decreased from 40 to 20 µg (*Figure 4—figure supplement 6*). From three experiments starting with 40 µg of proteins, a total of 195 cross-linked lysine pairs (43 inter-molecular and 152 intra-molecular) were identified (*Figure 4B* and *Figure 4—source data 4*). Thanks to cross-linking, not only did we identify all ten exosome core subunits, but also 15 putative direct protein-protein interactions amongst the core subunits, which generated a connectivity map more complete than the one from yeast two-hybrid experiments (*Stark et al., 2006*; *Uetz et al., 2000*; *Oliveira et al., 2002*; *Luz et al., 2007*; *Yu et al., 2008*) and showed that among the co-immunoprecipitated proteins, Rrp41 and Rrp45 directly bind to the bait protein Rrp46 (*Figure 4C*). Of the cross-links identified, 69% were compatible with the crystal structure of an RNA-bound 11-subunit exosome complex (*Makino et al., 2013*) (PDB code: 4IFD). Among the cross-links that disagreed with the RNA-bound structure, 68% involved the catalytic subunit Rrp44, which has a large rotation relative to the rest of the exosome core between the RNA-bound and the RNA-free states (*Makino et al., 2013*; *Liu et al., 2014*). The crude Rrp46 immunoprecipitate should mainly contain apo exosome, because magnesium was included in the buffer to activate the nuclease activity of exosome. Therefore, the presence (in the crystal structure) or absence (in our exosome preparation) of bound RNA is likely to be the primary reason behind most of the seemingly inconsistent inter-molecular cross-links.

To fulfill different functions in multiple biological processes (*Houseley and Tollervey, 2009*), the core exosome complex must recruit additional regulators, of which only a few are known. Here we found two known (Mpp6 [*Milligan et al., 2008*] and Ski7 *Araki et al., 2001*) and four potential exosome regulators through nine cross-links with core exosome subunits (*Figure 4B* and *Figure 4—source data 4*). These cross-links revealed residues in close proximity. Ski7 was found to cross-link with Rrp4 via K111, which fits well with previous co-IP results obtained by using different fragments of Ski7 (*Araki et al., 2001*) and a recently published CXMS study of the yeast exosome (*Shi et al., 2015*). Among the newly identified candidate regulators, the translation initiation factor Tif1 stood out; it had interactions with the Rrp4 and Rrp44 exosome subunits (*Figure 4B*). Translation has been implicated in RNA quality control (*Shoemaker and Green, 2012*). The linkages identified here support the hypothesis that exosome complexes 'stand by' the translation machinery and recognize and degrade aberrant mRNA molecules.

## Application of Leiker to lysates

We further tested Leiker for the purpose of mapping protein-protein interaction networks using *E. coli* and *C. elegans* lysates. *E. coli* whole-cell lysates are commonly used for evaluating CXMS

methods (*Rinner et al., 2008*; *Yang et al., 2012*; *Weisbrod et al., 2013*). In three independent experiments, Leiker-treated, trypsin digested *E. coli* lysates were fractionated on a high pH reverse phase column, and cross-linked peptides were enriched from each of the 10 or 11 fractions (*Figure 5—figure supplement 1*). After filtering the data by requiring FDR < 0.05, E-value < 0.01, spectral count $\geq$ 3, we identified a total of 2003 non-redundant inter-linked lysine pairs including 1386 (69%) intra-molecular and 617 (32%) inter-molecular cross-links (*Figure 5—source data 1* and *Figure 5—figure supplement 2*). Protein structure information is available in the PDB database for 984 intra-molecular cross-links identified with Leiker, and is consistent with 80% of them, indicating the high quality of the results (*Figure 5—source data 1*). Of note, the inter-molecular cross-links represent 436 pairs of protein-protein interactions, and 25% of the cross-links are supported by the combined network of the bacteriome.org database (*Peregrín-Alvarez et al., 2009*) *Figure 5—source data 1*). Most of the inter-molecular cross-links suggest novel protein-protein interactions. Based on the Leiker cross-links, we constructed a protein-protein interaction network and extracted the most highly connected module (*Bader and Hogue, 2003*; *Saito et al., 2012*) (*Figure 5A*). This 12-protein module consists of 9 ribosomal proteins and two DNA-binding proteins (the Hu heterodimer DBHA/DBHB) organized around a translation elongation factor Tu (EF-Tu). Evidently, it is enriched with proteins that function in translation, suggesting that DBHA/DBHB also plays a role in this process. Indeed, previous studies reported that a small fraction of this Hu heterodimer is bound to ribosomes (*Rouvière-Yaniv and Kjeldgaard, 1979*) and that this protein can enhance or repress translation of the mRNA molecules that it binds to (*Balandina et al., 2001*). In contrast, the most connected module obtained from the previously identified BS$^3$ cross-links (*Yang et al., 2012*) comprised only three ribosomal proteins (*Figure 5A*). These results indicate the potential of Leiker in generating comprehensive protein-protein interaction networks using CXMS. Since ribosomal proteins dominated *E. coli* whole-cell lysates, we prepared samples in which ribosomes were removed by centrifugation through a layer of sucrose cushion. Analysis of the ribo-free samples (two repeats) with Leiker identified 1971 inter-links, 1127 of which were not identified in the whole-cell lysates (5% FDR, E-value < 0.01, spectral count $\geq$ 3) (*Figure 5B*, *Figure 5—figure supplement 2*, and *Figure 5—source data 2*). Together, we identified a total of 3130 non-redundant cross-linked lysine pairs from *E. coli*. This allowed us to construct a network comprising 677 protein-protein interactions (*Figure 5—figure supplement 3A*).

Applying Leiker to an even more complex lysate from *C. elegans*, which has a similar number of protein coding genes as human (~20,000), we identified 459 inter-links (5% FDR, E-value < 0.01, spectral count $\geq$ 3) (*Figure 5—source data 3*). We also analyzed a *C. elegans* mitochondrial fraction and identified 547 inter-linked lysine pairs (5% FDR, E-value < 0.01, spectral count $\geq$ 3), of which 434 were not detected in the whole-worm lysate (*Figure 5—source data 4*). Together, we identified 893 non-redundant cross-linked lysine pairs from *C. elegans* and constructed protein-protein interactions between 155 proteins (*Figure 5—figure supplement 3B*).

In order to compare with previous studies, we also applied a less stringent cutoff (5% FDR, E-value < 0.01, spectral count $\geq$ 1) to the data sets of *E. coli* and *C. elegans* whole-cell lysates. This allowed us to determine that the number of *C. elegans* cross-links identified in this study was 23 times as many as the previous record (*Figure 5C*) (*Yang et al., 2012*). The number of *E. coli* cross-links identified in this study is four times greater than the number of PIR-identified inter-links (*Chavez et al., 2013*) and eight times greater than the number of BS$^3$-identified inter-links (*Yang et al., 2012*). Half of the BS$^3$-identified cross-links (*Yang et al., 2012*) were recapitulated in this study (*Figure 5D*).

## Leiker-based quantitative CXMS analysis

Relative quantification of cross-linker modified peptides can reveal changes in protein conformation and/or interactions between a protein and another molecule (e.g. nucleic acid, ligand, or protein). To apply Leiker in quantitative CXMS, we synthesized deuterium-labeled Leiker ([d$_6$]-bAL2) in addition to the unlabeled version ([d$_0$]-bAL2). Few software tools reported to date directly support quantitative CXMS (*Fischer et al., 2013*; *Walzthoeni et al., 2015*). We therefore modified the quantification software pQuant (*Liu et al., 2014*) and established an automated data analysis workflow for quantitative CXMS (*Figure 6* and Materials and methods). As a proof-of-principle experiment, we compared the RNA-free and H/ACA RNA-bound states of a *Pyrococcus furiosus* ribosomal protein L7Ae (*Rozhdestvensky et al., 2003*; *Li and Ye, 2006*). We treated RNA-free L7Ae with [d$_0$]-

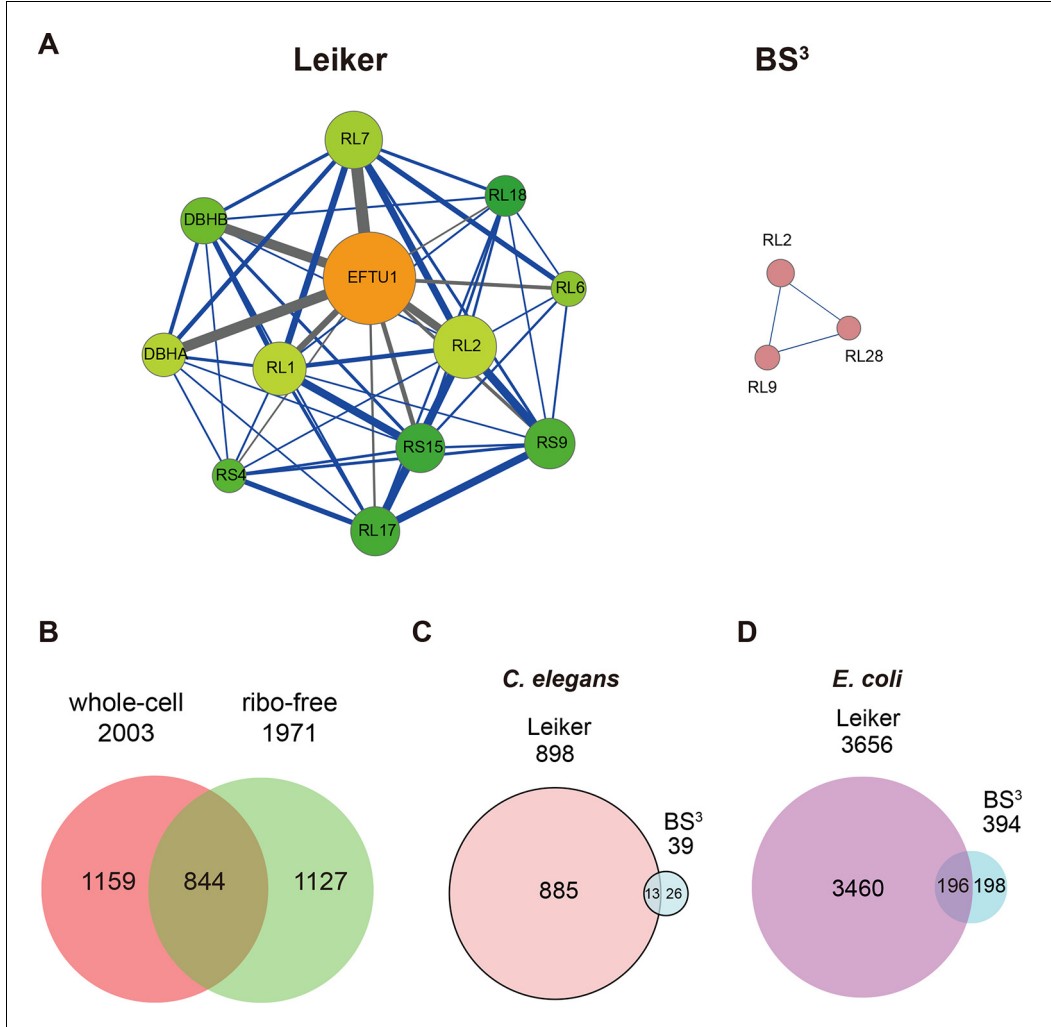

**Figure 5.** CXMS analyses of *E. coli* and *C. elegans* lysates. (**A**) The best protein-protein interaction cluster extracted from the Leiker-identified or BS³-identified (*Yang et al., 2012*) inter-links from *E. coli* whole-cell lysates. Node size represents the degree of connectivity of the indicated protein in the network. Line width represents the spectral counts of every inter-molecular cross-link. The line color is set to blue when the two peptides of an inter-link are both attributed to unique proteins, to grey if either could be assigned to multiple proteins. All the lines connected to EF-Tu1 are grey because EF-Tu1 differs from EF-Tu2 by only one amino acid. (**B**) Comparison of the identified inter-links in *E. coli* whole-cell lysates and ribo-free lysates (5% FDR, E-value < 0.01, spectral count ≥ 3). (**C** and **D**) Comparison of the number of Leiker-identified inter-links and that of BS³-identified inter-links (*Yang et al., 2012*) from *C. elegans* (**C**) and *E. coli* (**D**) whole-cell lysates (5% FDR, E-value < 0.01, spectral count ≥ 1).

The following source data and figure supplements are available for figure 5:

**Source data 1.** CXMS analysis of *E. coli* whole-cell lysates.
**Source data 2.** CXMS analysis of *E. coli* ribo-free lysates.
**Source data 3.** CXMS analysis of *C. elegans* whole-cell lysates.
**Source data 4.** CXMS analysis of *C. elegans* mitochondrial proteins.
**Figure supplement 1.** Fractionation of digested, Leiker-treated *E. coli* lysates.

*Figure 5 continued on next page*

*Figure 5 continued*

**Figure supplement 2.** Overlap of cross-linked lysine pairs between biological replicates of *E. coli* lysates (FDR < 0.05, E-value < 0.01, and spectral count ≥ 3).

**Figure supplement 3.** Protein-protein interaction networks constructed from the cross-links identified in (**A**) *E. coli* and (**B**) *C. elegans*.

bAL2 and the assembled L7Ae-RNA complex with [$d_6$]-bAL2 in the forward reaction, and switched the isotope labels in the reverse reaction (*Figure 7A*). An equal amount of BSA protein was included in each sample to control for possible difference in cross-linking efficiency between [$d_0$]- and [$d_6$]-Leiker. We expected that the formation of the protein-RNA complex would block the access of Leiker to lysine residues at the binding interface, which would be manifested as large abundance decrease of mono- or inter-linked peptides at these sites.

Mono-linked peptides are usually neglected in CXMS, but they are valuable because they indicate that the modified lysine residues are exposed to solvent. Mono-links at all 15 lysine residues and the N-terminus of L7Ae were reliably quantified (*Figure 6*) in both forward and reverse labeling experiments. Three mono-links at K35, K42, and K84 consistently had significantly higher abundance (>5 fold) in the RNA-free state (F) than in the RNA-bound state (B) (*Figure 7B–D*, *Figure 7—figure supplement 1* and *Figure 7—source data 1*). None of inter-links passed the quantification criteria described above. These results suggest that the three lysine residues are buried upon RNA binding, either due to direct protein-RNA binding or indirect protein conformational changes induced by RNA binding. This is in perfect agreement with the crystal structure (*Li and Ye, 2006*) (PDB code: 2HVY), which shows that K35, K42, and K84 all bind to the RNA, each with a buried area greater than 20 Å$^2$ (*Figure 7C*).

Lastly, we applied quantitative CXMS to *E. coli* lysates. The log phase and the stationary phase cell lysates were cross-linked, respectively, with [$d_0$]- and [$d_6$]-bAL2 in the forward labeling experiment, or with [$d_6$]- and [$d_0$]-bAL2 in the reverse labeling experiment. After a single enrichment step without pre-fractionation, a total of 161 inter-linked lysine pairs were quantified in both the forward and the reverse labeling experiments, and most of them had similar [log phase]/[stationary phase] ratios in the two experiments (*Figure 8* and *Figure 8—source data 1*). Noticeably, the cross-link between YqjD and ElaB increased at least 10 times in the stationary phase compared to the log phase. These two paralogous proteins are associated with the inner membrane of *E. coli* cells through their C-terminal transmembrane motifs and both bind to stationary phase ribosomes, probably through their N-terminal regions (*Yoshida et al., 2012*). It is suggested that YqjD binding to ribosomes inhibits translation (*Yoshida et al., 2012*). Association of YqjD and ElaB has been detected but the sites of interaction are not known (*Hu et al., 2009*). Here, our results not only confirm previous findings, but also provide new insights that YqjD and ElaB form a heterodimer through their central regions, presumably as a stronger, divalent anchoring site for ribosomes to inhibit protein translation in the stationary phase.

## Discussion

In this study, we developed an MS-friendly and isotope-encodable cross-linker called Leiker that enables the efficient enrichment of cross-linked peptides through biotin-based immobilization and azobenzene-based chemical cleavage. With an enrichment efficiency of 97% or more, Leiker yields a fourfold increase in the number of identified cross-linked peptide pairs from complex samples. Also established is a workflow for quantitative CXMS based on deuterium-labeled Leiker.

In theory, a comprehensive network of putative direct protein-protein interactions could be obtained by applying Leiker to lysates. However, the interaction networks obtained as such are limited, because the cross-links identified are dominated by those from highly abundant proteins, for example, EF-Tu and ribosomal proteins in *E. coli*. This can be overcome with subcellular fractionation, which can separate abundant proteins from less abundant ones. We increased the number of unique inter-link identifications by more than 50% (from 2003 to 3130) by simply removing ribosomes from the *E. coli* lysates (*Figure 5B*). This is also obvious by contrasting the CXMS results of

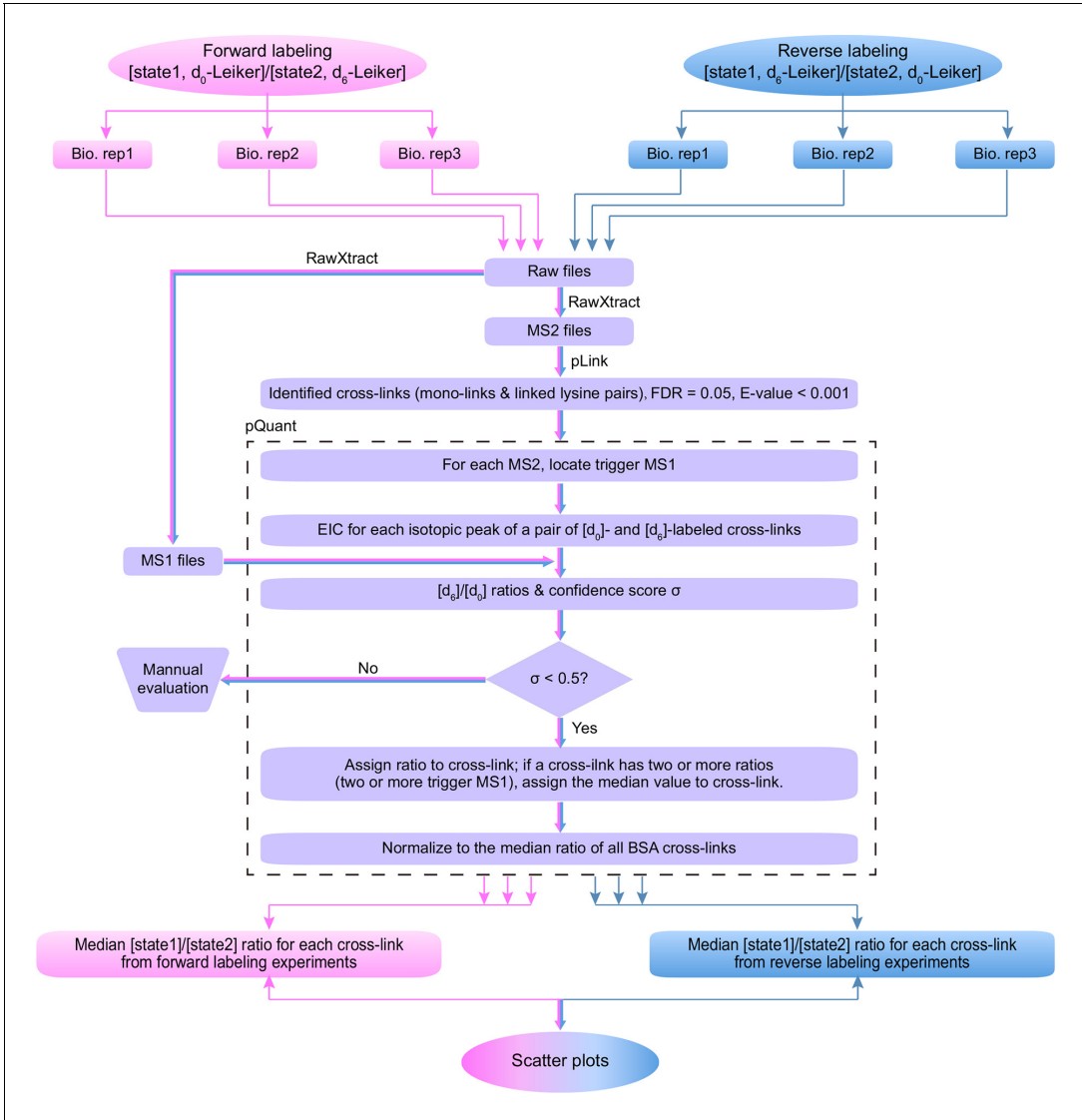

**Figure 6.** Workflow for quantification of cross-linked peptides using pQuant. For each identified cross-link spectrum, an extracted ion chromatogram (EIC) is constructed for each isotopic peak of the $[d_0]$- and $[d_6]$-labeled precursor. The $[d_6]/[d_0]$ ratios can be calculated based on the monoisotopic peak, the most intense peak, or the least interfered peak of each isotopic cluster as specified by users. The accuracy of the ratio calculation was evaluated with the confidence score σ (range: 0–1, from the most to the least reliable). If a cross-link have ratios with σ < 0.5, the median of these ratios is assigned to this cross-link. The cross-link ratios of the proteins of interest are normalized to the median ratio of all BSA cross-links. For each cross-link, the median [state1]/[state2] ratio of three independent forward labeling experiments is plotted against the median ratio of three independent reverse labeling experiments. Cross-links that are only present in state1 or state2 due to a dramatic conformational change cannot be quantified as described above because the ratios would be zero or infinite and their σ values would be 1. Therefore, if a cross-link does not have a valid ratio after automatic quantification, the EICs were manually inspected to determine if it was an all-or-none change.

the whole-worm lysate and the mitochondrial fraction of *C. elegans*, from which 459 and 547 inter-linked lysine pairs were detected, respectively, with an overlap of only 113. We anticipate that extensive protein fractionation coupled with Leiker-assisted CXMS will pave the way towards constructing comprehensive interactomes for different model organisms, and next-generation cross-link identification software of higher sensitivity will also help. Further, with the advantage of heavy isotope labeling for quantification in addition to the enrichment function, Leiker shows promise for use in differential interactome analysis (*Ideker and Krogan, 2012*).

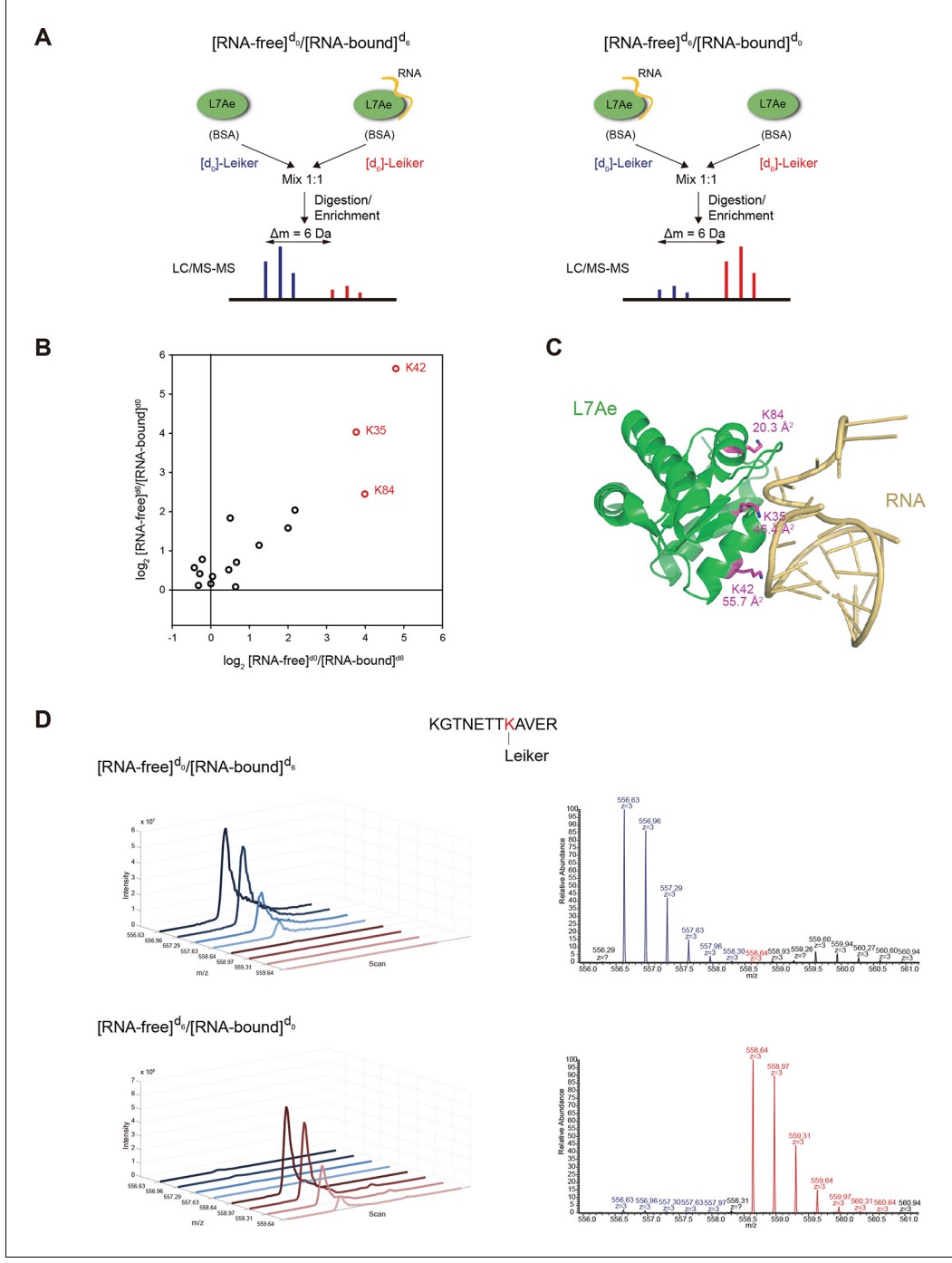

**Figure 7.** Quantitative CXMS analysis of the L7Ae-RNA complex. (**A**) Reciprocal labeling of RNA-free (**F**) and RNA-bound (**B**) L7Ae with $[d_0]/[d_6]$-Leiker. (**B**) Abundance ratios of mono-links (F/B) in the forward ($F^{[d0]}/B^{[d6]}$) and the reverse labeling experiment ($F^{[d6]}/B^{[d0]}$). Each circle represents a mono-linked lysine residue and is colored red if it has a ratio greater than five in both labeling schemes. (**C**) The three lysine residues affected by RNA binding are highlighted in the structure model (PDB code: 2HVY). The number below each such lysine residue indicates the buried surface area ($\text{Å}^2$) upon RNA binding. (**D**) Extracted ion chromatograms (left) and representative MS1 spectra (right) of a K42 mono-link.

The following source data and figure supplement are available for figure 7:

**Source data 1.** Quantitative CXMS analysis of L7Ae with or without the H/ACA RNA.

*Figure 7 continued on next page*

*Figure 7 continued*

**Figure supplement 1.** Extracted ion chromatograms (left) and representative MS1 spectra (right) of a mono-linked peptide corresponding to (**A**) K35 and (**B**) K84.

When we examined the cross-links identified from *E. coli* against the protein structures deposited in the PDB database, we noted that the intra-molecular cross-links in both the whole-cell lysates and the ribo-free samples had similar rates of structural compatibility (80% and 84%, respectively). This shows that the quality of our Leiker-based CXMS data is high. Interestingly, the inter-molecular cross-links detected from the ribo-free samples had a much higher rate of structural compatibility (69%) than those detected in the whole-cell lysates (12%). Given that 92% of the inter-links with existing structural information in the whole-cell lysate involved at least one ribosomal protein and many were between ribosomal proteins, we think that most of the apparently incompatible inter-molecular cross-links seen in the whole-cell lysates likely result from cross-linking of adjacent ribosomal particles.

Previous cross-linking studies have typically treated mono-linked peptides as by-products, and have ignored them. This is regrettable, as they carry structural information about proteins and always outnumber inter-links (*Figure 3*). Leiker also generates abundant mono-links. In this study, we demonstrate that mono-links are highly valuable in mapping RNA-binding lysine residues. As the positively charged lysine residue is frequently involved in binding the negatively charged phosphate backbone of DNA and RNA, relative quantification of lysine mono-links would be particularly suited for mapping the DNA or RNA binding surface on a protein. We suggest that mono-link data should be used in routine practice.

# Materials and methods

## Materials

Acetonitrile, methanol, formic acid, ammonium bicarbonate, and acetone were purchased from J.T. Baker (Center Valley, PA). Dimethylsulfoxide (DMSO), HEPES, urea, thiourea, and other general chemicals were purchased from Sigma-Aldrich (St. Louis, MO). Trypsin and Lys-C were purchased from Promega (Wisconsin, WI). Bis(sulfosuccinimidyl) suberate (BS$^3$), streptavidin agarose resin, and high capacity streptavidin agarose resin were purchased from Pierce (Rockford, IL). Dynabeads M-280 streptavidin was purchased from Invitrogen (Carlsbad, CA).

## Preparation of protein samples

RNase A, lysozyme, aldolase, BSA, lactoferrin, β-galactosidase, and myosin were obtained from Sigma-Aldrich. Recombinant GST containing an N-terminal His tag was expressed in *E. coli* BL21 cells from the pDYH24 plasmid and purified with glutathione sepharose (GE Healthcare, Piscataway, NJ). PUD-1/PUD-2 heterodimers were purified on a HisTrap column followed by gel filtration. Stock solutions of the ten standard proteins were individually buffer exchanged into 20 mM HEPES, pH8.0 by ultrafiltration, and then mixed to make a total protein mixture with a 2 μg/μl protein concentration.

Purification of 70S ribosomes from *E. coli* cells was performed as previously described (*Guo et al., 2011*). *E. coli* cells (DH5α) were grown in 2 L LB medium to an $OD_{600}$=0.8. Cells were collected by centrifugation, washed with 100 mL lysis buffer (50 mM HEPES-KOH, pH 7.5, 500 mM KCl, 12 mM MgCl$_2$, 1 mM DTT, 1 mM PMSF) and resuspended in 100 mL of lysis buffer. Cells were then disrupted with an Ultrasonic Cell Disruptor. The lysate was clarified at 13,000 rpm for 1 hr at 4°C in a JA 25.50 motor (Beckman Coulter, UK). The supernatant was layered on a sucrose cushion (50 mM HEPES-KOH, pH 7.5, 500 mM KCl, 12 mM MgCl$_2$, 33% sucrose) and centrifuged at 30,000 rpm for 18 hr in a 70Ti rotor (Beckman Coulter) at 4°C. The supernatant was collected as the ribo-free lysate. The pellet was resolved with a buffer containing 50 mM HEPES-KOH, pH 7.5, 500 mM KCl, and 12 mM MgCl$_2$. The crude ribosomes were then layered on a 10–50% sucrose gradient (50 mM HEPES-KOH, pH 7.5, 500 mM KCl, 12 mM MgCl$_2$, 10% to 50% sucrose) and centrifuged at 28,000 rpm for 5 hr in an SW28 rotor (Beckman Coulter) at 4°C. The gradient was scanned

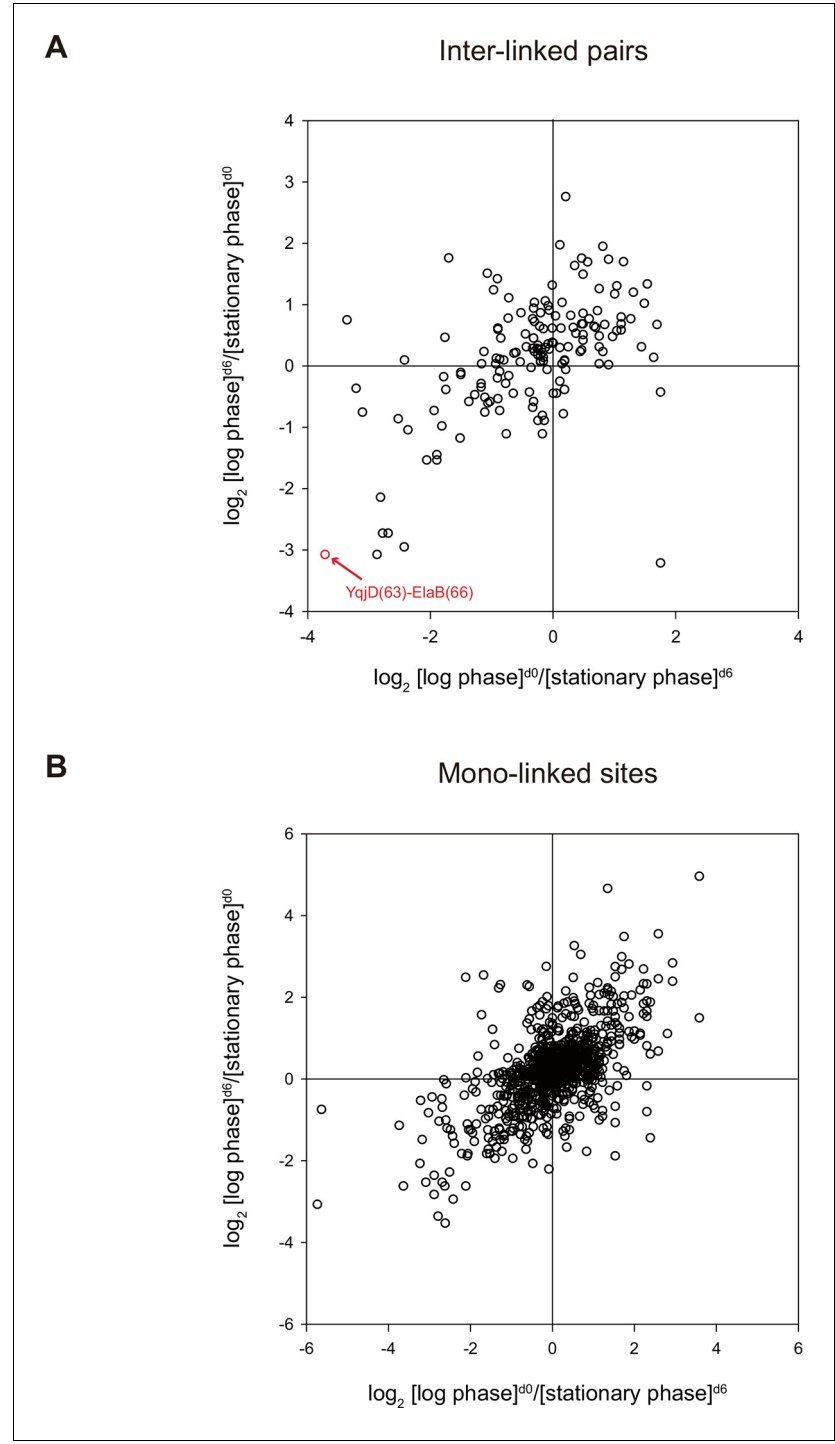

**Figure 8.** Quantitative CXMS analysis of *E. coli* lysates. Abundance ratios of (**A**) inter-linked lysine pairs and (**B**) mono-linked sites in the forward ([log phase]$^{d0}$/[stationary phase]$^{d6}$) and the reverse labeling experiment ([log phase]$^{d6}$/[stationary phase]$^{d0}$).

The following source data is available for figure 8:

**Source data 1.** Quantitative CXMS analysis of *E. coli* lysates.

at 260 nm and fractionated in an ISCO gradient collector. The fractions of 70S ribosomes were pooled and concentrated with Amicon Ultra centrifugation filters (Millipore, China) with a buffer containing 50 mM HEPES-KOH, pH 7.5, 500 mM KCl, and 12 mM MgCl$_2$.

The yeast exosome complex was immunoprecipitated with IgG beads as described previously (*Liu et al., 2014*), with the following modifications: a gentle wash buffer (150 mM NaCl) was applied and the mono-Q anion exchange step was not performed. These modifications were made in order to maintain the interaction of the proteins in the sample. Eluted proteins were exchanged into 20 mM HEPES, pH 8.0, 150 mM NaCl.

*E. coli* OP50 lysates and *C. elegans* N2 lysates were prepared following a protocol from Bing et al. (*Yang et al., 2012*; *Zhao et al., 2015*). Mitochondria were isolated from the wild-type N2 worms as described previously (*Shen et al., 2014*) and lysed by incubation in 100 mM HEPES pH 8.0, 1% NP-40, 10 mM CaCl$_2$ at 4°C for 30 min.

The *Pyrococcus furiosus* L7Ae and the H/ACA RNA were prepared as described previously (*Li and Ye, 2006*). The buffer was exchanged to 50 mM HEPES, pH 7.6, 1 M NaCl.

*E. coli* (MG1665) cells were grown at 37°C in 500 mL M9 minimal medium from a 1 mL overnight culture. Log phase cells were harvested after 11 hr at OD$_{600}$ 0.7; stationary phase cells were harvested after 26 hr at OD$_{600}$ 2.3. Cell lysates were prepared in 50 mM HEPES pH 8.0, 150 mM NaCl using a FastPrep system (MP Biomedicals, Santa Ana, CA) using two volumes of glass beads at 6.5 m/s, 20 s per pulse for four pulses, with 5 min of cooling on ice between pulses. The lysates were cleared by centrifugation at top speed in a tabletop microfuge for 30 min. Protein concentrations were determined using the bicinchoninic acid assay.

## Trypsin digestion

At room temperate (RT), protein pellets were dissolved (assisted by sonication) in 8 M urea, 20 mM methylamine (to reduce carbamylation), 100 mM Tris, pH 8.5, reduced with 5 mM TCEP for 20 min and alkylated with 10 mM iodoacetamide for 15 min in the dark. Then, the samples were diluted with 3 volumes of 100 mM Tris, pH 8.5 and digested with trypsin at 1/50 (w/w) enzyme/substrate ratio at 37°C for 16–18 hr.

## CXMS analysis of model proteins

The optimal protein-to-cross-linker mass ratio was determined by a titration experiment. 1 µl of cross-linker at increasing concentrations (2.5 µg/µl, 5 µg/µl, 10 µg/µl, 20 µg/µl, 40 µg/µl) in DMSO was incubated with 20 µl of 2 µg/µl of the ten-protein mixture at RT for 1 hr to make 16:1, 8:1, 4:1, 2:1, and 1:1 protein-to-cross-linker mass ratios, respectively. The reactions were quenched with 20 mM NH$_4$HCO$_3$ at RT for 20 min. Cross-linking products were analyzed by SDS-PAGE. The 4:1 ratio was ultimately chosen for both the one-piece and the two-piece Leiker. Higher dosages were avoided to minimize excessive cross-linking.

For comparison of the one-piece and two-piece Leikers, 50 µl of the 2 µg/µl ten-protein mixture was incubated with 0.5 µl of 50 µg/µl AL or bAL1 at RT for 1 hr. The reactions were quenched as described above. For AL, the solution was mixed with 350 µl of 8 M urea, 100 mM Tris, pH 8.5, and filtered with an Amicon Ultra-0.5 10-kD filter device (Millipore). Excess cross-linker molecules were removed by two additional washes with urea. Click chemistry was subsequently performed on the membrane. In a 100 µl reaction, 28 nmol of azide-biotin was added (an amount equal to the starting amount of the alkyne group of AL), followed by the addition of 2 mM CuSO$_4$, 2 mM TCEP, and 200 µM TBTA. Samples were gently rotated and incubated at RT for 2 hr. The excess free azide-biotin was then removed by washes with urea in the filter device. Finally, the proteins were collected by centrifugation with the filter device placed upside down inside the tube. Recovered proteins were transferred to a new 1.5 mL tube, precipitated at -20°C with four volumes of pre-cooled acetone for at least 30 min, and digested with trypsin. The bAL1 samples were processed in the same way except that the reaction mixture was precipitated directly without going through the 10-kD filter device.

The AL- and bAL1-cross-linked peptides were enriched in parallel. The tryptic digests, without formic acid (FA) acidification, were directly mixed with an equal volume of 20 mM HEPES, pH 8.0 and incubated with 40 µl pre-washed high capacity streptavidin agarose for 2 hr. Then, the beads were washed three times with 20 mM HEPES, pH 8.0, 1 M KCl, once with H$_2$O, three times with 10%

acetonitrile (ACN), and another three times with $H_2O$, each time with 1 mL buffer or $H_2O$, with 5-min rotation. Supernatants were removed carefully with a 1 mL syringe needle connected to a vacuum pump. Loss of beads was avoided by keeping the beveled surface of the needle tip in contact with the wall of the tube. After the extensive washes, the peptides were released by incubating the beads with 5× bed volumes of cleavage buffer (300 mM $Na_2S_2O_4$ in 6 M urea, 2 M thiourea, 10 mM HEPES, pH8.2) (*Yang et al., 2010*) at 37°C for 30 min, with end-to-end rotation. Recovered peptides were acidified with 5% FA and subsequently desalted on home-made C18 desalting columns, followed by elution with 70% ACN/0.1% FA. Eluates were vacuum dried and reconstituted in 0.1% FA for mass spectrometry analyses. The color of the beads could be used to monitor the entire enrichment process: a bright yellow color indicated the binding of Leiker-linked peptides; a return to a white color occurred when the cleavage reaction was successful.

Comparison of bAL1 and bAL2 was carried out in two samples. For the first comparison, 50 μg of the ten-protein mixture was cross-linked with bAL1 or bAL2 at 4:1 protein-to-cross-linker mass ratio and then digested with trypsin. After mixing with the tryptic digest of an *E. coli* lysate containing 500 μg of total proteins, the digested Leiker-linked peptides were affinity purified with 20 μl of high-capacity streptavidin agarose. For the second comparison, 30 μg of ribosome was treated with bAL1 or bAL2 at 8:1, 4:1, or 2:1 protein-to-cross-linker mass ratios, digested, and enriched using 20 μl of high-capacity streptavidin agarose.

For the serial dilution experiment (*Figure 3*), 200 μg of the ten-protein mixture was treated with 50 μg of bAL1 at RT for 1 hr. After quenching, the proteins were precipitated and digested with trypsin. Four equal aliquots of this digest were either not diluted to serve as a control (1:0) or diluted with the tryptic digest of a non-cross-linked *E. coli* lysate at 1:1, 1:10, or 1:100 (w/w) ratio. Each mixture was enriched with 200 μl of pre-washed streptavidin agarose.

## CXMS analysis of purified ribosomes and the immunoprecipitated exosome complex

30 μg of ribosome was treated with bAL2 at 8:1, 4:1, or 2:1 protein-to-cross-linker mass ratios. 40 μl of the exosome complex sample (1 μg/μl) was incubated with 0.25 μl of 40 μg/μl bAL2 at RT for 1 hr. 20 μl of high-capacity streptavidin agarose was used to enrich Leiker-linked peptides in each sample.

## Negative staining of *E. coli* 70S ribosome

70S ribosomes were negatively stained with 0.2% uranyl acetate. Carbon coated grids were first glow-discharged to increase the surface hydrophilicity using a Harrick Plasma cleaner. 4 μL aliquots of 70S ribosomes (~10 nM) were placed on grids for about 1 min, and excessive liquid was absorbed by filter paper. After that 0.2% uranyl acetate was applied on the grid for about 1 min and absorbed using filter paper. The grids were air-dried and examined using an FEI Tecnai Spirit BioTwin microscope (FEI, Hillsboro, OR) (120 KV) at 49,000× magnification.

## CXMS analysis of *E. coli* and *C. elegans* cell lysates

*E. coli* or *C. elegans* lysates prepared as described previously (*Yang et al., 2012*; *Zhao et al., 2015*) (1 mg of total proteins) were treated with 250 μg bAL1 at RT for 1 hr, in 300 μl reactions; $NH_4HCO_3$ was added to quench the reactions. Proteins were precipitated and digested with trypsin. After centrifugation in a bench top centrifuge at top speed for 30 min and filtering with a 50-kD cutoff filter, the digested peptides were brought to a volume of 3 mL with 2% ACN, 20 mM HEPES, pH 8.2; the pH was adjusted to 10.0 with ammonia prior to high-pH reverse phase separation on an Xtimate column (10×250 mm) packed with 5 μm C18 resin (Welch Materials, China) at a flow rate of 2 mL/min. A 70 min gradient was applied as follows: 0-6% B in 10 min, 6-40% B in 40 min, 40-100% B in 10 min, 100% B for 10 min (A = 4% ACN, 5 mM $NH_4COOH$, pH 10, B = 80% ACN, 5 mM $NH_4COOH$, pH 10). A total of 39 two-min fractions were collected, and then combined into 9–11 fractions of similar shades of color judging by naked eyes. These pooled samples were evaporated to 200–300 μl volumes before Leiker-linked peptides were enriched with 50 μl of high-capacity streptavidin beads from each sample. For the ribo-free lysates, 3 mg of proteins were cross-linked with 0.75 mg bAL2 at RT for 1 hr, and subjected to tryptic digestion and fractionation as described above.

*C. elegans* mitochondria were prepared as described previously (*Shen et al., 2014*), and the CXMS analysis was performed as described above except with two differences: 3.2 mg of total proteins was used as the starting material and the collected fractions were pooled into 5 fractions.

## Quantitative CXMS analysis of the L7Ae-RNA complex

In the forward and reverse labeling experiments, 0.7 nmol of RNA-free L7Ae was treated with $[d_0]$-bAL2 and $[d_6]$-bAL2, respectively; an equal amount of L7Ae was pre-incubated with 1 nmol of the 65 nt H/ACA RNA at 4°C for 30 min and then treated with $[d_6]$-bAL2 and $[d_0]$-bAL2, respectively. An equal amount of BSA was spiked into each cross-linking reaction. A 4:1 protein-to-cross-linker ratio (w/w) was used for each reaction. The cross-linking reactions were quenched with ammonium bicarbonate after 1 hr at RT. The paired $[d_0]$- and $[d_6]$-bAL2 samples were combined and subjected to acetone precipitation and trypsin digestion.

## Quantitative CXMS analysis of *E. coli* lysates

In the forward labeling experiment, the log phase and the stationary phase cell lysates (100 µg proteins each) were cross-linked with 50 µg of $[d_0]$-bAL2 and 50 µg of $[d_6]$-bAL2, respectively, with 1 µg of BSA spiked into each sample. After 1 hr at RT, the two reactions were quenched, mixed, precipitated with acetone, and digested with trypsin. The reverse labeling experiment was conducted in the same way except that the log phase lysate was cross-linked with $[d_6]$-bAL2 and the stationary phase lysate was cross-linked with $[d_0]$-bAL2.

## LC-MS/MS analysis

All protein samples were analyzed with an EASY-nLC 1000 system (Thermo Fisher Scientific, Waltham, MA) interfaced with a Q-Exactive mass spectrometer (Thermo Fisher Scientific). A two-column setup was used, consisting of a pre-column (100 µm×4 cm, 3 µm C18) with a frit at each end and an analytical column (75 µm×10 cm, 1.8 µm C18) with a 5 µm tip. For the Leiker-cross-linked samples after enrichment, typically one third of a reconstituted sample was injected and separated with a 65 min linear gradient at a flow rate of 300 nl/min as follows: 0–5% B in 2 min, 5–28% B in 41 min, 28–80% in 10 min, 80% for 12 min (A = 0.1% FA, B = 100% ACN, 0.1% FA). Slight modifications to the separation method were made for different samples. A 120 min gradient was used with a more gradual ramp to 28% buffer B. The Q-Exactive mass spectrometer was operated in data-dependent mode with one full MS scan at R = 70000 (m/z = 200), followed by ten HCD MS/MS scans at R = 17,500 (m/z = 200), NCE = 27, with an isolation width of 2 m/z. The AGC targets for the MS1 and MS2 scans were 3e6 and 1e5, respectively, and the maximum injection times for MS1 and MS2 were both 60 ms. For cross-linked samples, precursors of the +1, +2, +7 or above, or unassigned charge states were rejected; exclusion of isotopes was disabled; dynamic exclusion was set to 30 s.

For accurate mass analysis, 20 µg/ml of $[d_0]$-bAL2 or $[d_6]$-bAL2 in methanol was sprayed directly into a LTQ Orbitrap XL mass spectrometer (Thermo Fisher Scientific) operated in the negative mode with a spray voltage of 0.8 kV and a scan mass range of 150–1000 m/z.

## Identification of cross-linked peptides with pLink

The Xcalibur raw data was converted to ms2 files using RawExtract (*McDonald et al., 2004*). Cross-linked peptides were identified using pLink software as described previously (*Yang et al., 2012*), with the following modifications Cross-linker was set to AL, bAL1, bAL2, $[d_6]$-bAL2, or BS[3]; The minimum peptide length was 5 amino acids for lysate samples; oxidation on Met was set as a variable modification.

For the ten-protein mixture and ribosome complexes, the search databases consisted of the sequences of all of the proteins in question. The sequences were downloaded from NCBI or Uniprot.

Prior to the CXMS analysis of the exosome complex, LC-MS/MS analyses of digested, uncross-linked samples were carried out to identify the proteins present in the samples. For protein identification, the precursors of +1 or unassigned charge states were rejected; MS2 spectra were searched against a *S. cerevisiae* protein database (downloaded from Uniprot on 2013-04-03) using ProLuCID2 (*Xu et al., 2006*) and filtered using DTASelect 2.0 (*Tabb et al., 2002*) with a spectral false identification rate ≤1% and a minimum of two identified peptides for each protein. A restricted database containing only the identified proteins (740 in total) was generated using Contrast 2.0 (*Tabb et al.,*

*2002*). MS2 spectra from the cross-linked samples were then searched against this small database using pLink.

For the CXMS analysis of *E. coli* whole-cell lysates and ribo-free lysates, the sequences of the entire proteome of the K12 strain were downloaded from Uniprot on 2014-07-31 and used for searching.

For the CXMS analysis of *C. elegans* lysates, a database consisting of proteins identified from N2 *C. elegans* lysates generated with ProLuCID2 was used for searching (unpublished).

For the CXMS analysis of *C. elegans* mitochondrial proteins, a restricted database was constructed in a similar way as for the exosome complex.

## Quantification of cross-linked peptides with pQuant

pQuant (*Liu et al., 2014*) was used to determine the heavy-to-light ratio (H/L) of each cross-link. The regression model $Y = aX + e$ is used to calculate peptide ratios. The optimal value of $a$ is solved using the least-squares method as $\hat{a} = \sum X_j Y_j / \sum X_j X_j$, and the estimated standard error of $\hat{a}$ is $\hat{\sigma} = (K^{-1} \cdot \sum (Y_j - \hat{a} X_j)^2 / \sum X_j^2)^{1/2}$. is then normalized to the interval of [0,1], and is named confidence score. If the value of $\hat{\sigma}$ is zero (the highest confidence), there is no interference signal; if the value is one (the lowest confidence), the peptide signals are inundated by interference signals. For each identified cross-link spectrum, an extracted ion chromatogram (EIC) was constructed for each isotopic peak of the light- and heavy-labeled precursor. The H/L ratios can be calculated based on the monoisotopic peak, the most intense peak, or the least interfered peak of each isotopic cluster as specified by users. For L7Ae, all options yielded similar results and we selected the monoisotopic peak. For the highly complex samples of the log phase versus stationary phase *E. coli*, the option of the least interfered peak performed the best. For each cross-link, every identified spectrum (E-value < 0.001) will lead to a H/L ratio and a confidence score σ, because pQuant conducts the quantitation independently starting from each identified MS/MS spectrum. In most cases, the H/L ratios obtained for the same precursor ion are close, but sometimes the ratios may differ due to multiple reasons including local interference signals or a sudden decrease followed by recovery in signal intensity in the chromatograms, all of which can affect the calling of the start and the end of a chromatogram peak. Ratios with σ values above or equal to 0.5 were discarded. The median H/L ratio obtained from the remaining spectra was assigned to a cross-linked lysine pair or a mono-linked lysine residue as the final quantification value. If a cross-link had no assigned ratio value (i.e., none of its ratios had a σ value less than 0.5), we manually evaluated the reconstructed ion chromatograms to assess abundance changes. All the ratios were normalized against the median value of all the H/L ratios belonging to the spiked-in BSA.

## Acknowledgements

We thank Mingyan Zhao (NIBS) for NMR and HPLC-MS analysis. We also thank Dr. Li-Lin Du (NIBS), members of the pFind group (ICT, CAS), and members of the Dong lab (NIBS, Beijing) for discussions and experimental support. This work was supported by the National Natural Science Foundation of China (21375010 to M-QD, 31325007 to KY, 31422016 to NG, 21475141 to S-MH, 21222209, 21472010, and 91313303 to X-GL), the Ministry of Science and Technology of China (973 grants 2015CB856200 and 2012CB837400 to X-GL, 2014CB849800 to M-QD, 2012CB910602 to CL, 2010CB912701 to S-MH, 2010CB912401 to H-WW, the National Scientific Instrumentation Grant Program 2011YQ09000506 to M-QD), and the Chinese Academy of Sciences (CAS Knowledge Innovation Program and ICT-20126033 to S-MH, Strategic Priority Research Program XDB08010203 to KY).

## Additional information

### Funding

| Funder | Author |
|--------|--------|
| National Natural Science Foundation of China | Si-Min He<br>Ning Gao<br>Keqiong Ye<br>Meng-Qiu Dong<br>Xiaoguang Lei |
| Ministry of Science and Technology of the People's Republic of China | Chao Liu<br>Hong-Wei Wang<br>Si-Min He<br>Meng-Qiu Dong<br>Xiaoguang Lei |
| Chinese Academy of Sciences | Si-Min He<br>Keqiong Ye |

The funders had no role in study design, data collection and interpretation, or the decision to submit the work for publication.

### Author contributions

DT, Designed MS experiments, acquired and analyzed all the MS data, prepared samples, interpreted the data and wrote the manuscript; QL, Performed the chemical synthesis, interpreted the data, wrote the manuscript; M-JZ, CL, PZ, S-BF, Wrote data analysis programs or scripts, helped with data interpretation; CM, Prepared samples, performed the EM analysis; Y-HD, Wrote data analysis programs or scripts, helped with data interpretation, prepared samples; LT, BY, Contributed to MS analysis; XLi, XLiu, Performed the chemical synthesis; SM, JL, BF, Prepared samples; H-WW, NG, KY, Directed protein purification and data interpretation, revised the manuscript; S-MH, Directed software development; M-QD, XL, Designed and guided the study, interpreted the data, wrote the manuscript

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

## Appendix

### Synthesis of Leiker molecules

#### Instrumentation and methods

[1]H NMR spectra were recorded on a Varian 400 MHz spectrometer at ambient temperature with $CDCl_3$ as the solvent unless otherwise stated. [13]C NMR spectra were recorded on a Varian 100 MHz spectrometer (with complete proton decoupling) at ambient temperature. Chemical shifts are reported in parts per million relative to chloroform ([1]H, δ 7.26; [13]C, δ 77.00). Data for [1]H NMR are reported as follows: chemical shift, multiplicity (s = singlet, d = doublet, t = triplet, q = quartet, m = multiplet), integration and coupling constants. Infrared spectra were recorded on a Thermo Fisher FT-IR200 spectrophotometer. High-resolution mass spectra were obtained at Peking University Mass Spectrometry Laboratory using a Bruker APEX Flash chromatography. The samples were analyzed by HPLC/MS on a Waters Auto Purification LC/MS system (3100 Mass Detector, 2545 Binary Gradient Module, 2767 Sample Manager, and 2998 Photodiode Array (PDA) Detector). The system was equipped with a Waters C18 5μm SunFire separation column (150*4.6 mm), equilibrated with HPLC grade water (solvent A) and HPLC grade acetonitrile (solvent B) with a flow rate of 0.3 mL/min. Flash chromatography was performed using 200-400 mesh silica gel. Yields refer to chromatographically and spectroscopically pure materials, unless otherwise stated.

#### Reagents and solvents

All chemical reagents were used as supplied by Sigma-Aldrich, J&K and Alfa Aesar Chemicals. DCM, DMF, DMSO were distilled from calcium hydride; tetrahydrofuran was distilled from sodium/benzophenone ketyl prior to use. 4-(2-carboxyethyl) heptanedioic acid **1** (*Newkome et al., 1988*; *Karunaratne et al., 1992*), 1-(5-methoxy-2-nitro-4-(prop-2-yn-1-yloxy) phenyl)ethanol N-(3-azidopropyl)-5-(2- oxohexahydro-1H-thieno[3,4-d]imidazol-4-yl) pentanamide **4** (*Kaneko et al., 2011*), 4-(prop-2-yn-1-yloxy)aniline **9** (*Liu et al., 2005*), tert-butyl 3-hydroxybenzyl carbamate **10** (*Nhu et al., 2010*), **14** (*Wang et al., 2013*),(9H-fluoren-9-yl)methyl (3-hydroxy propyl)carbamate **19** (*Crestey et al., 2008*), biotin pentafluorophenyl ester **24** (*Kessler et al., 2009*), were prepared according to the literature reported procedures. All reactions were carried out in oven-dried glassware under an argon atmosphere unless otherwise noted.

## Synthetic procedures

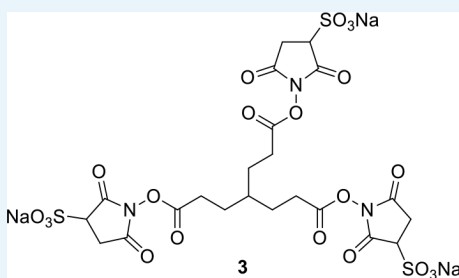

**Appendix 1—figure 1.** Synthesis of sulfo-Photo-cleavable Leiker (sulfo-PL, 8)

Reagents and conditions: (a) **2**, EDCI, DMSO, 24 h; (b) **5**, EDCI, DMAP, DCM, 12 h, 93%; (c) TFA/DCM, 2.5 h; (d) **3**, TEA, DMSO, 24 h, 56% for two steps.

**Appendix 1—figure 2.** Compound 3.

Compound **3**: To a solution of triacid **1** (23.2 mg, 0.1 mmol) in anhydrous DMSO (5 mL) was added **2** (71.6 mg, 0.33 mmol) and EDCI (67.1 mg, 0.35 mmol). The reaction was stirred at room temperature for 24 h. After that, 40 mL of anhydrous THF was added, and the solution became muddy. The solvents were poured out after standing overnight to give the crude product **3**, which was directly used for the next step.

**Appendix 1—figure 3.** Compound 6.

Compound **6**: To a solution of alcohol **4** (48.5 mg, 0.193 mmol) in DCM (2.5 mL) was added acid **5** (67.6 mg, 0.386 mmol), EDCI (74 mg, 0.386 mmol) and DMAP (0.7 mg, 0.0058 mmol). The reaction was stirred at room temperature for 12 h. $H_2O$ (5 mL) was added and the mixture

was extracted with $CH_2Cl_2$ (10 mL×3). The combined organic layers were washed with brine (5 mL). After dried over $Na_2SO_4$, the solution was concentrated *in vacuo* and purified by flash chromatography (silica gel, 30% EtOAc in petrol ether) to afford the desired compound **6** as a waxy solid (75.7 mg, 93%): [1]H NMR (400 MHz CDCl$_3$): δ 1.41 (s, 9H), 1.62 (d, *J* = 6.4 Hz, 3H), 2.58 (m, 3H), 3.38 (m, 2H), 3.97 (s, 3H), 4.82 (d, *J* = 2.4 Hz, 2H), 4.95 (br, 1H), 6.51 (q, *J* = 6.4 Hz, 1H), 7.03 (s, H), 7.75 (s, 1H); [13]C NMR (100 MHz CDCl$_3$): δ 22.0, 28.3, 34.7, 36.0, 56.4, 56.9, 68.5, 77.0, 77.1, 79.4, 108.4, 110.2, 133.8, 139.6, 145.6, 154.1, 155.8, 171.2; IR (neat) $\nu_{max}$ 3291, 2978, 1710, 1519, 1336, 1276, 1209, 1168, 1018, 791 cm$^{-1}$; HRMS (ESI): $[M+Na]^+$ calculated for $C_{20}H_{26}N_2NaO_8$: 445.1581, found: 445.1585.

**Appendix 1—figure 4.** Compound 7.

Compound **7**: To a solution of compound **6** (22.4 mg, 0.053 mmol) in DCM (2 mL) was added TFA (0.5 mL). The resulting mixture was stirred at room temperature for 2.5 h. The solvents were removed *in vacuo* to afford the crude product **7**, which was directly used for the next step.

**Appendix 1—figure 5.** Compound 8.

Compound **8**: To a solution of the crude product **3** (freshly prepared from 23.2 mg of triacid **1**) in DMSO (2 mL) was added compound **7** (freshly prepared from 22.4 mg of compound **6**) dissolved in DMSO (0.5 mL). Triethylamine (50 μL, 0.36 mmol) was added to the solution subsequently. The reaction was stirred at room temperature for 24 h. DMSO was removed by using the genevac HT-4X evaporator and the product was isolated by HPLC (20-40% CH$_3$CN in water over 18 min). CH$_3$CN was removed *in vacuo* and water was removed by using Christ ALPHA 1-4 LD plus to yield a white solid (26.7 mg, 56% for two steps): mp=142-145 °C; [1]H NMR (400 MHz DMSO): δ 1.48 (m, 3H), 1.58 (m, 7H), 2.01 (m, 2H), 2.64 (m,4H), 2.85 (m, 2H), 3.24 (m, 2H), 3.65 (t, *J* = 2.4 Hz, 1H), 3.94 (s, 5H), 4.93 (d, *J* = 2.4 Hz, 2H), 6.22 (q, *J* = 6.4 Hz, 1H), 7.14 (s, H), 7.69 (s, 1H), 7.94 (t, *J* = 5.6 Hz, 1H); [13]C NMR (100 MHz DMSO): δ 15.6, 21.4, 25.5, 27.1, 27.6, 30.9, 32.1, 33.9, 34.2, 34.5, 35.3, 36.1, 42.3, 54.6, 56.3, 56.4, 67.5, 78.4, 79.2, 109.0, 109.4, 132.9, 139.2, 145.3, 153.8, 158.5, 165.3, 168.8, 170.7, 172.1; IR (neat) $\nu_{max}$ 3274, 2924, 1782, 1732, 1651, 1518, 1205, 1066, 1016, 793 cm$^{-1}$; HRMS (ESI): $[M+Na]^+$ calculated for $C_{33}H_{36}N_4Na_3O_{21}S_2$: 957.1001, found: 957.1026.

**Appendix 1—figure 6.** Synthesis of Azo-Leiker (AL, 13).

Reagents and conditions: (a) (1) Con. HCl, NaNO$_2$, H$_2$O, 0~5 °C, 1.3 h; (2) **10**, NaOH, H$_2$O, 0~5 °C, 2 h, 88%; (b) TFA/DCM, 2.5 h; (c) **3**, TEA, DMSO, 24 h, 35% for two steps.

**Appendix 1—figure 7.** Compound 11.

Compound **11**: To a solution of compound **9** (131 mg, 0.89 mmol) in H$_2$O (4 mL) was added Con. HCl (220 µL, 2.64 mmol) at 0~5 °C, the reaction was stirred for 20 min. Subsequently, NaNO$_2$ (86 mg, 1.24 mmol) dissolved in H$_2$O (3 mL) was added. The resulting mixture was stirred at 0~5 °C for 1 h. After that, the resulting mixture was added to the solution of compound **10** (207 mg, 0.89 mmol) in 0.6 M NaOH solution (4 mL) at 0~5 °C dropwise. The reaction mixture was stirred at 0~5 °C for 2 h. The mixture was neutralized with 0.25 M HCl, extracted with DCM (20 mL×3). The combined organic layers were washed with brine (10 mL). After dried over Na$_2$SO$_4$, the solution was concentrated *in vacuo* and purified by flash chromatography (silica gel, 30% EtOAc in petrol ether) to afford the desired compound **11** as a bright yellow waxy solid (298 mg, 88%): mp=133-135 °C; $^1$H NMR (400 MHz CDCl$_3$): δ 1.45 (s, 9H), 2.57 (t, *J* = 2.4 Hz, 1H), 4.77 (m, 4H), 5.22 (br, 1H), 6.84 (dd, *J* = 8.8 Hz, 2.8 Hz, 1H), 7.01 (s, 1H), 7.08 (dd, *J* = 6.8 Hz, 2.0 Hz, 2H), 7.72 (d, *J* = 8.8 Hz, 1H), 7.86 (dd, *J* = 6.8 Hz, 2.0 Hz, 2H); $^{13}$C NMR (100 MHz CDCl$_3$): δ 28.4, 41.4, 56.0, 76.0, 78.0, 80.2, 115.1, 115.67, 115.72, 118.0, 124.3, 139.6, 143.7, 147.7, 156.4, 159.3, 159.5; IR (neat) ν$_{max}$ 3283, 2974, 1680, 1598, 1584, 1500, 1229, 1162, 1024, 837, 669 cm$^{-1}$; HRMS (ESI): [M+H]$^+$ calculated for C$_{21}$H$_{24}$N$_3$O$_4$: 382.1761, found: 382.1768.

**Appendix 1—figure 8.** Compound 12.

Compound **12**: To a solution of compound **11** (20.2 mg, 0.053 mmol) in DCM (2 mL) was added TFA (0.5 mL). The resulting mixture was stirred at room temperature for 2.5 h. The

solvents were removed *in vacuo* to afford the crude product **12**, which was directly used for the next step.

**Appendix 1—figure 9.** Compound 13.

Compound **13**: To a solution of the crude product **3** (prepared from 23.2 mg of triacid) in DMSO (2 mL) was added compound **12** (prepared from 20.2 mg of compound **11**) dissolved in DMSO (0.5 mL). Triethylamine (50 µL, 0.36 mmol) was added to the solution subsequently. The reaction was stirred at room temperature for 24 h. DMSO was removed by using the genevac HT-4X evaporator and the product was isolated by HPLC (20-40% $CH_3CN$ in water over 18 min). $CH_3CN$ was removed *in vacuo* and water was removed by using Christ ALPHA 1-4 LD plus to yield a yellow solid (16.3 mg, 35% for two steps): mp=148-150 °C; [1]H NMR (400 MHz DMSO): δ 1.62 (m, 7H), 2.17 (m, 2H), 2.65 (m, 6H), 2.86 (m, 2H), 3.61 (s, 1H), 3.94 (s, 2H), 4.76 (d, *J* = 4.8 Hz, 2H), 4.90 (s, 2H), 6.74 (d, *J* = 8.4 Hz, 1H), 6.85 (s, 1H), 7.15 (d, *J* = 8.8 Hz, 2H), 7.58 (d, *J* = 9.2 Hz, 1H), 7.85 (d, *J* = 8.8 Hz, 2.0 Hz, 2H), 8.35 (br, 1H), 10.17 (s, 1H); [13]C NMR (100 MHz DMSO): δ 27.2, 27.6, 28.1, 30.9, 32.4, 35.4, 37.9, 55.8, 56.3, 78.6, 78.9, 114.5, 114.7, 115.4, 116.6, 124.1, 141.0, 142.1, 147.2, 159.0, 160.5, 165.4, 168.8, 172.2; IR (neat) $v_{max}$ 3323, 2947, 1737, 1618, 1594, 1217, 1036, 845, 672 cm$^{-1}$; HRMS (ESI): [M-Na]$^-$ calculated for $C_{34}H_{33}N_5NaO_{17}S_2$: 870.1216, found: 870.1197.

**Appendix 1—figure 10.** Synthesis of biotinylated Azo-Leiker 1 (bAL 1, 17).

Reagents and conditions: (a) **14**, CuSO$_4$.5H$_2$O, sodium ascorbate, DCM/DMF/H$_2$O, 24 h, 91%; (b) TFA/DCM, 2.5 h; (c) **3**, TEA, DMSO, 24 h, 37% for two steps.

**Appendix 1—figure 11.** Compound 15.

Compound **15**: To a solution of **11** (69.2 mg, 0.181 mmol) in DMF/DCM/H$_2$O (2/2/2 mL) was added **14** (77 mg, 0.236 mmol), CuSO$_4$.5H$_2$O (2.3 mg, 0.009 mmol) and sodium ascorbate (5.4 mg, 0.027 mmol). The mixture was stirred at room temperature for 24 h. H$_2$O (4 mL) was added, extracted with EtOAc (12 mL×3). The combined organic layers were washed with brine (5 mL). After dried over Na$_2$SO$_4$, the solution was concentrated *in vacuo* and purified by flash chromatography (silica gel, 6~10% methanol in methylene chloride) to afford the desired compound **15** as a bright yellow waxy solid (117 mg, 91%): mp=120-122 °C; $^1$H NMR (400 MHz Methanol-d$_4$): δ 1.44 (m, 2H), 1.48 (s, 9H), 1.64 (m, 4H), 2.13 (m, 2H), 2.21 (t, J = 7.6 Hz, 2H), 2.69 (d, J = 12.4 Hz, 1H), 2.90 (dd, J = 12.8 Hz, 5.2 Hz, 1H), 3.18 (m, 1H), 3.23 (t, J = 6.8 Hz, 2H), 4.28 (m, 1H), 4.47 (m, 3H), 4.78 (s, 2H), 5.26 (s, 2H), 6.76 (dd, J = 8.8 Hz, 2.8 Hz, 1H), 6.91 (d, J = 2.8 Hz, 1H), 7.14 (d, J = 9.2 Hz, 2H), 7.66 (d, J = 8.8 Hz, 1H), 7.87 (dd, J = 6.8 Hz, 2.0 Hz, 2H), 8.13 (s, 1H); $^{13}$C NMR (100 MHz Methanol-d$_4$): δ 26.7, 28.9, 29.4, 29.8, 31.1, 36.7, 41.0, 41.2, 49.2, 57.0, 61.6, 62.6, 63.3, 80.3, 115.3, 115.8, 116.2, 118.1, 125.4, 125.6, 142.1, 144.2, 144.6, 149.0, 158.4, 161.6, 161.8, 166.0, 176.2; IR (neat) ν$_{max}$ 3299, 2930, 1694, 1582, 1597, 1500, 1243, 1147, 838 cm$^{-1}$; HRMS (ESI): [M+H]$^+$ calculated for C$_{34}$H$_{46}$N$_9$O$_6$S: 708.3286, found: 708.3277.

**Appendix 1—figure 12.** Compound 16.

Compound **16**: To a solution of compound **15** (37.5 mg, 0.053 mmol) in DCM (2 mL) was added TFA (0.5 mL). The resulting mixture was stirred at room temperature for 2.5 h. The solvents were removed *in vacuo* to afford the crude product **16**, which was directly used for the next step.

**Appendix 1—figure 13.** Compound 17.

Compound **17**: To a solution of the crude product **3** (prepared from 23.2 mg of triacid) in DMSO (2 mL) was added compound **16** (prepared from 37.5 mg of compound **15**) dissolved in DMSO (0.5 mL). Triethylamine (50 μL, 0.36 mmol) was added to the solution subsequently. The reaction was stirred at room temperature for 24 h. DMSO was removed by using the genevac HT-4X evaporator and the product was isolated by HPLC (10-30% $CH_3CN$ in water over 18 min). $CH_3CN$ was removed *in vacuo* and water was removed by using Christ ALPHA 1-4 LD plus to yield a yellow solid (23.8 mg, 37% for two steps): mp=158-161 °C; $^1$H NMR (400 MHz DMSO): δ 1.30 (m, 2H), 1.57 (m, 11H), 1.96 (m, 2H), 2.21 (t, *J* = 7.6 Hz, 2H), 2.17 (m, 2H), 2.57 (d, *J* = 12.4 Hz, 1H), 2.65 (m, 4H), 2.74 (s, 2H), 2.81 (m, 3H), 3.00 (m, 3H), 3.95 (d, *J* = 6.4 Hz, 2H), 4.12 (m, 1H), 4.30 (m, 1H), 4.38 (t, *J* = 7.2 Hz, 2H), 4.76 (d, *J* = 5.6 Hz, 2H), 5.24 (s, 2H), 6.35 (s, 1H), 6.42 (s, 1H), 6.74 (dd, *J* = 8.8 Hz, 2.4 Hz, 1H), 6.85 (d, *J* = 2.4 Hz, 1H), 7.20 (d, *J* = 8.8 Hz, 2H), 7.58 (d, *J* = 8.8 Hz, 1H), 7.84 (d, *J* = 8.8 Hz, 2H), 7.90 (t, *J* = 5.6 Hz, 1H), 8.29 (s, 1H), 8.36 (t, *J* = 5.6 Hz, 1H), 10.16 (s, 1H); $^{13}$C NMR (100 MHz DMSO): δ 15.6, 25.3, 25.5, 27.2, 27.6, 28.0, 28.2, 30.0, 30.9, 32.4, 34.2, 35.2, 35.4, 35.7, 36.1, 37.9, 42.3, 47.3, 54.7, 55.4, 56.2, 56.3, 59.2, 61.0, 61.5, 114.5, 114.7, 115.2, 116.6, 124.2, 124.8, 140.9, 142.1, 142.3, 146.9, 158.5, 160.0, 160.5, 162.7, 165.3, 168.8, 172.18, 172.25; IR (neat) $v_{max}$ 3321, 2939, 1717, 1685, 1601, 1524, 1257, 1174, 1142, 840 cm$^{-1}$; HRMS (ESI): [M-2Na]$^{2-}$ calculated for $C_{47}H_{55}N_{11}O_{19}S_3$: 586.6424, found: 586.6413.

**Appendix 1—figure 14.** Synthesis of biotinylated Azo-Leiker 2 (bAL 2, 27).

Reagents and conditions: (a) **19**, DEAD, PPh₃, THF, 0 °C, 1.5 h, 97 %; (b) FeSO₄.7H₂O, Fe, ethanol/H₂O, 80°C, 18 h, 85 %; (c) (1) Con. HCl, NaNO₂, H₂O, 0~5 °C, 1.3 h; (2) **10**, NaOH, H₂O, 0~5 °C, 2 h, 94 %; (d) TBAF, i-PrOH/DMF, 2 h, 95 %; (e) **24**, TEA, DMF, 2.5 h, 90 %; (f) TFA/DCM, 2.5 h; (g) **3**, TEA, DMSO, 24 h, 44% for two steps.

**Appendix 1—figure 15.** Compound 20.

Compound **20**: 4-nitrophenol **18** (211 mg, 1.52 mmol) was dissolved in anhydrous THF (37 mL), compound **19** (519 mg, 1.75 mmol), PPh₃ (344 mg, 1.97 mmol), DEAD (518 mg, 1.97 mmol) were added at 0 °C subsequently. The reaction mixture was stirred at 0 °C for 1.5 h. The mixture was quenched with sat. NH₄Cl solution (10 mL), extracted with CH₂Cl₂ (15 mL×3). The combined organic layers were washed with brine (10 mL). After dried over Na₂SO₄, the solution was concentrated *in vacuo* and purified by flash chromatography (silica gel, 25% EtOAc in petrol ether) to afford the desired compound **20** as a white solid (615 mg, 97%): mp=128-130 °C; $^1$H NMR (400 MHz CDCl₃): δ 2.04 (m, 2H), 3.41 (m, 2H), 4.09 (t, $J$ = 6.0 Hz, 2H), 4.21 (t, $J$ = 6.4 Hz, 1H), 4.44 (d, $J$ = 6.8 Hz, 2H), 4.94 (br, 1H), 6.93 (m, 2H), 7.31 (m, 2H), 7.40 (m, 2H), 7.58 (m, 2H), 7.77 (m, 2H), 8.19 (m, 2H); $^{13}$C NMR (100 MHz CDCl₃): δ 29.3, 38.1, 47.2, 66.3, 66.5, 114.4, 120.0, 124.9, 125.9, 127.0, 127.7, 141.3, 141.6, 143.8, 156.4, 163.7; IR (neat) $\nu_{max}$ 3325, 2951, 1702, 1592, 1509, 1448, 1335, 1256, 1109, 844, 741 cm$^{-1}$; HRMS (ESI): [M+Na]$^+$ calculated for C₂₄H₂₂N₂NaO₅: 441.1421, found: 441.1424.

**Appendix 1—figure 16.** Compound 21.

Compound **21**: To a solution of compound **20** (598 mg, 1.43 mmol) in ethanol/$H_2O$ (15 mL/4.5 mL) was added $FeSO_4.7H_2O$ (80 mg, 0.286 mmol) and iron dust (705 mg, 12. 6 mmol). The reaction mixture was stirred at 80 °C for 18 h. The reaction mixture was filtered over a short pad of silica gel with EtOAc as the eluant, the filtrate was removed by rotary evaporation. The residue was dissolved in EtOAc (50 mL), washed with $H_2O$ (10 mL). After dried over $Na_2SO_4$, the solution was concentrated *in vacuo* and purified by flash chromatography (silica gel, 30% EtOAc in petrol ether) to afford the desired compound **21** as a waxy solid (471 mg, 85%): [1]H NMR (400 MHz $CDCl_3$): δ 1.96 (m, 2H), 3.40 (m, 4H), 3.96 (t, *J* = 6.0 Hz, 2H), 4.22 (t, *J* = 7.2 Hz, 1H), 4.40 (d, *J* = 6.8 Hz, 2H), 5.14 (br, 1H), 6.64 (m, 2H), 6.70 (m, 2H), 7.31 (m, 2H), 7.40 (m, 2H), 7.60 (m, 2H), 7.76 (m, 2H); [13]C NMR (100 MHz $CDCl_3$): δ 29.3, 38.8, 47.3, 66.5, 66.6, 115.6, 116.4, 119.9, 125.0, 127.0, 127.6, 140.2, 141.3, 144.0, 151.8, 156.4; IR (neat) $v_{max}$ 3348, 2950, 1704, 1510, 1449, 1233, 1044, 825, 741 cm$^{-1}$; HRMS (ESI): $[M+H]^+$ calculated for $C_{24}H_{25}N_2O_3$: 389.1860, found: 389.1866.

**Appendix 1—figure 17.** Compound 22.

Compound **22**: To a solution of compound **21** (345 mg, 0.89 mmol) in $H_2O$ (4 mL) was added Con. HCl (220 μL, 2.64 mmol) at 0~5 °C, the reaction was stirred for 20 min. Subsequently, $NaNO_2$ (86 mg, 1.24 mmol) dissolved in $H_2O$ (3 mL) was added. the resulting mixture was stirred at 0~5 °C for 1 h. After that, the resulting mixture was added to the solution of compound **10** (207 mg, 0.89 mmol) in 0.6 M NaOH solution (4 mL) at 0~5 °C dropwise. The reaction mixture was stirred at 0~5 °C for 2 h. The mixture was neutralized with 0.25 M HCl, extracted with DCM (20 mL×3). The combined organic layers were washed with brine (10 mL). After dried over $Na_2SO_4$, the solution was concentrated *in vacuo* and purified by flash chromatography (silica gel, 30% EtOAc in petrol ether) to afford the desired compound **22** as a bright yellow solid (520 mg, 94%): mp=79-81 °C; [1]H NMR (400 MHz $CDCl_3$): δ 1.44 (s, 9H), 2.05 (m, 2H), 3.44 (m, 2H), 4.10 (m, 2H), 4.22 (t, *J* = 6.8 Hz, 1H), 4.44 (d, *J* = 6.8 Hz, 2H), 4.76 (d, *J* = 6.4 Hz, 2H), 5.03 (br, 1H), 5.17 (br, 1H), 6.83 (dd, *J* = 8.8 Hz, 2.8 Hz, 1H), 6.70 (m, 3H), 7.31 (m, 2H), 7.40 (m, 2H), 7.60 (m, 2H), 7.72 (d, *J* = 8.8 Hz, 1H), 7.77 (d, *J* = 7.2 Hz, 2H), 7.84 (d, *J* = 8.8 Hz, 2H); [13]C NMR (100 MHz $CDCl_3$): δ 28.4, 29.3, 38.4, 41.3, 47.2, 65.9, 66.6, 79.9, 114.6, 115.6, 115.7, 117.8, 119.9, 124.4, 124.9, 127.0, 127.7, 139.6, 141.3, 143.8, 147.3, 156.3, 156.6, 159.3, 160.6; IR (neat) $v_{max}$ 3298, 2928, 1686, 1597, 1501, 1467, 1240, 1144, 837, 741 cm$^{-1}$; HRMS (ESI): $[M+H]^+$ calculated for $C_{36}H_{39}N_4O_6$: 623.2864, found: 623.2855.

**Appendix 1—figure 18.** Compound 23.

Compound **23**: To a solution of compound **22** (494 mg, 0.79 mmol) in *i*PrOH/DMF (1.5 mL/15 mL) was added TBAF (0.04 M in DMF, 30 mL, 1.2 mmol) dropwise. The reaction was stirred at room temperature for 2 h. The mixture was quenched with sat. NH$_4$Cl solution (15 mL), extracted with EtOAc (40 mL×3). The combined organic layers were washed with brine (20 mL). After dried over Na$_2$SO$_4$, the solution was concentrated *in vacuo* and purified by flash chromatography (silica gel, 6~10% methanol in methylene chloride) to afford the desired compound **23** as a bright yellow solid (317 mg, 95%): mp=182-184 °C; $^1$H NMR (400 MHz DMSO): δ 1.41 (s, 9H), 1.93 (t, *J* = 6.4 Hz, 2H), 2.85 (t, *J* = 6.4 Hz, 2H), 3.33 (br, 2H), 4.14 (t, *J* = 6.4 Hz, 2H), 4.65 (d, *J* = 6.0 Hz, 2H), 6.73 (dd, *J* = 8.8 Hz, 2.8 Hz, 1H), 6.85 (d, *J* = 2.4 Hz, 1H), 7.09 (dd, *J* = 6.8 Hz, 2.0 Hz, 2H), 7.37 (t, *J* = 6.0 Hz, 1H), 7.57 (d, *J* = 8.8 Hz, 1H), 7.83 (dd, *J* = 6.8 Hz, 2.0 Hz, 2H); $^{13}$C NMR (100 MHz DMSO): δ 28.3, 29.6, 37.2, 65.5, 77.8, 114.0, 114.6, 114.9, 116.4, 124.1, 141.3, 141.8, 146.7, 155.8, 160.4, 160.7; IR (neat) ν$_{max}$ 3468, 2975, 1682, 1598, 1583, 1502, 1248, 1165, 836 cm$^{-1}$; HRMS (ESI): [M+H]$^+$ calculated for C$_{21}$H$_{29}$N$_4$O$_4$: 401.2183, found: 401.2187.

**Appendix 1—figure 19.** Compound 25.

Compound **25**: To a solution of biotin pentafluorophenyl ester **24** (46 mg, 0.11 mmol) in anhydrous DMF (1 mL) was added compound **23** (37 mg, 0.093 mmol) in anhydrous DMF (1.5 mL) dropwise. Triethylamine (26 µL, 0.187 mmol) was added to the solution subsequently. The reaction was stirred at room temperature for 2.5 h. The solvent was removed in vacuo and the crude product was purified by flash chromatography (silica gel, 3~9% methanol in methylene chloride) to afford the desired product **25** as a bright yellow solid (52 mg, 90%): mp=124-126 °C; $^1$H NMR (400 MHz DMSO): δ 1.30 (m, 2H), 1.41 (s, 9H), 1.54 (m, 4H), 1.88 (t, *J* = 6.4 Hz, 2H), 2.08 (t, *J* = 7.6 Hz, 2H), 2.59 (d, *J* = 12.4 Hz, 1H), 2.79 (dd, *J* = 12.4 Hz, 4.8 Hz, 1H), 3.07 (m, 1H), 3.23 (m, 2H), 4.09 (m,3H), 4.28 (m, 1H), 4.66 (d, *J* = 5.6 Hz, 2H), 6.37 (s, 1H), 6.44 (s, 1H), 6.74 (dd, *J* = 8.8 Hz, 2.4 Hz, 1H), 6.86 (d, *J* = 2.0 Hz, 1H), 7.07 (d, *J* = 8.8 Hz, 2H), 7.34 (t, *J* = 6.0 Hz, 1H), 7.58 (d, *J* = 8.8 Hz, 1H), 7.83 (d, *J* = 8.8 Hz, 2H), 7.90 (t, *J* = 5.6 Hz, 1H), 10.13 (s, 1H); $^{13}$C NMR (100 MHz DMSO): δ 25.3, 28.0, 28.2, 28.3, 28.9, 35.2, 35.3, 39.9, 55.4, 59.2, 61.0, 65.7, 77.8, 114.0, 114.5, 114.9, 116.5, 124.2, 141.3, 141.9, 146.7, 155.8, 160.4, 160.5, 162.7, 172.1; IR (neat) ν$_{max}$ 3287, 2930, 1692, 1596, 1581, 1501, 1243, 1163, 838 cm$^{-1}$; HRMS (ESI): [M+H]$^+$ calculated for C$_{31}$H$_{43}$N$_6$O$_6$S: 627.2959, found: 627.2951.

**Appendix 1—figure 20.** Compound 26.

Compound **26**: To a solution of compound **25** (33.3 mg, 0.053 mmol) in DCM (2 mL) was added TFA (0.5 mL). The resulting mixture was stirred at room temperature for 2.5 h. The solvents were removed in vacuo to afford the crude product **26**, which was directly used for the next step.

**biotinylated Azo-Leiker 2 (bAL 2, 27)**

**Appendix 1—figure 21.** Compound 27.

Compound **27**: To a solution of the crude product **3** (prepared from 23.2 mg of triacid) in DMSO (2 mL) was added compound **26** (prepared from 33.3 mg of compound **25**) dissolved in DMSO (0.5 mL). Triethylamine (50 μL, 0.36 mmol) was added to the solution subsequently. The reaction was stirred at room temperature for 24 h. DMSO was removed by using the genevac HT-4X evaporator and the product was isolated by HPLC (20-40% $CH_3CN$ in water over 18 min). $CH_3CN$ was removed *in vacuo* and water was removed by using Christ ALPHA 1-4 LD plus to yield a yellow solid (26.5 mg, 44% for two steps): mp=152-154°C; [1]H NMR (400 MHz DMSO): δ 1.31 (m, 2H), 1.57 (m, 11H), 1.88 (m, 2H), 2.07 (t, $J$ = 7.2 Hz, 2H), 2.18 (m, 2H), 2.56 (d, $J$ = 12.8 Hz, 1H), 2.65 (m, 4H), 2.76 (s, 2H), 2.83 (m, 3H), 3.00 (m, 1H), 3.20 (m, 2H), 3.95 (d, $J$ = 7.2 Hz, 2H), 4.09 (m, 3H), 4.28 (m, 1H), 4.76 (d, $J$ = 5.6 Hz, 2H), 6.34 (s, 1H), 6.41 (s, 1H), 6.74 (dd, $J$ = 8.8 Hz, 2.4 Hz, 1H), 6.85 (d, $J$ = 2.4 Hz, 1H), 7.08 (d, $J$ = 8.8 Hz, 2H), 7.58 (d, $J$ = 8.8 Hz, 1H), 7.83 (d, $J$ = 8.8 Hz, 2H), 7.89 (t, $J$ = 5.6 Hz, 1H), 8.36 (t, $J$ = 5.6 Hz, 1H), 10.14 (s, 1H); [13]C NMR (100 MHz DMSO): δ 15.6, 25.3, 25.5, 27.2, 27.6, 28.0, 28.2, 28.9, 30.9, 32.4, 34.2, 35.2, 35.3, 36.0, 37.9, 42.3, 54.6, 55.4, 56.3, 59.2, 61.0, 65.7, 114.4, 114.7, 114.9, 124.2, 140.8, 142.1, 146.7, 158.5, 160.4, 160.6, 162.7, 165.3, 168.8, 172.1, 172.2; IR (neat) $v_{max}$ 3328, 2937, 1737, 1714, 1650, 1597, 1234, 1040, 631 cm[-1]; HRMS (ESI): [M-2Na][2-] calculated for $C_{44}H_{52}N_8O_{19}S_3$: 546.1261, found: 546.1265.

**Appendix 1—figure 22.** Synthesis of $d_6$-biotinylated Azo-Leiker 2 (bAL 2, 31).

Reagents and conditions: (a) sat. KOH, 105 °C, 2 d, 84%; (b) **2**, EDCI, DMSO, 24 h; (c) **26**, TEA, DMSO, 24 h, 31% for two steps.

**Appendix 1—figure 23.** Compound 29.

Compound **29**: A solution of compound **28** (102 mg, 0.42 mmol, Note: compound **28** was synthesized with acrylonitrile-$d_3$ (purchased from CDN) as the starting material. The procedure was the same as compound **1**. Unfortunately it lost some deuterium during the procedure) in sat. KOH aqueous (2 mL) was stirred at 105 °C for 2 d. The mixture was acidified with con. HCl, extracted with DCM (100 mL×3). The combined organic layers were washed with brine (5 mL). After dried over $Na_2SO_4$, the solution was concentrated *in vacuo* to give the desired product **29** as a white solid (85 mg, 84%): mp=107-109°C; $^1$H NMR (400 MHz DMSO): δ 1.28 (s, 1H), 2.16 (s, 6H), 12.02 (s, 3H); $^{13}$C NMR (100 MHz DMSO): δ 30.7, 35.0, 174.6; IR (neat) $\nu_{max}$ 2925, 1696, 1413, 1283, 1217, 911 cm$^{-1}$; HRMS (ESI): [M+K]$^+$ calculated for $C_{10}H_{10}D_6KO_6$: 277.0955, found: 277.0956.

**Appendix 1—figure 24.** Compound 30.

Compound **30**: To a solution of $d_6$-triacid **29** (23.8 mg, 0.1 mmol) in anhydrous DMSO (5 mL) was added **2** (71.6 mg, 0.33 mmol) and EDCI (67.1 mg, 0.35 mmol). The reaction was stirred at

room temperature for 24 h. After that, 40 mL of anhydrous THF was added, and the solution became muddy. The solvents were poured out after standing overnight to give the crude product **30**, which was directly used for the next step.

**d₆-biotinylated Azo-Leiker 2 (d₆-bAL 2, 31)**

**Appendix 1—figure 25.** Compound 31.

Compound **31**: To a solution of the crude product **30** (prepared from 23.8 mg of d$_6$-triacid **29**) in DMSO (2 mL) was added compound **26** (prepared from 33.3 mg of compound **25**) dissolved in DMSO (0.5 mL). Triethylamine (50 μL, 0.36 mmol) was added to the solution subsequently. The reaction was stirred at room temperature for 24 h. DMSO was removed by using the genevac HT-4X evaporator and the product was isolated by HPLC (20-40% CH$_3$CN in water over 18 min). CH$_3$CN was removed *in vacuo* and water was removed by using Christ ALPHA 1-4 LD plus to yield a yellow solid (18.8 mg, 31% for two steps): mp=154-156°C; $^1$H NMR (400 MHz DMSO): δ 1.30 (m, 2H), 1.50 (m, 5H), 1.88 (m, 2H), 2.07 (t, *J* = 7.2 Hz, 2H), 2.16 (m, 2H), 2.56 (d, *J* = 12.4 Hz, 1H), 2.63 (m, 4H), 2.69 (s, 2H), 2.81 (m, 3H), 3.05 (m, 1H), 3.21 (m, 2H), 3.95 (d, *J* = 6.8 Hz, 2H), 4.09 (m, 3H), 4.28 (m, 1H), 4.76 (d, *J* = 5.6 Hz, 2H), 6.34 (s, 1H), 6.41 (s, 1H), 6.74 (dd, *J* = 8.8 Hz, 2.4 Hz, 1H), 6.85 (d, *J* = 2.4 Hz, 1H), 7.08 (d, *J* = 9.2 Hz, 2H), 7.58 (d, *J* = 8.8 Hz, 1H), 7.83 (d, *J* = 8.8 Hz, 2H), 7.90 (t, *J* = 5.6 Hz, 1H), 8.36 (t, *J* = 5.6 Hz, 1H), 10.16 (s, 1H); $^{13}$C NMR (100 MHz DMSO): δ 25.3, 27.4, 28.1, 28.2, 28.9, 30.9, 34.9, 35.2, 35.4, 37.9, 54.9, 55.4, 56.2, 56.3, 59.2, 61.1, 65.7, 114.5, 114.7, 114.9, 116.6, 124.2, 140.8, 142.1, 146.7, 160.4, 160.6, 162.7, 165.4, 168.8, 172.1, 172.3; IR (neat) ν$_{max}$ 3325, 2932, 1738, 1712, 1647, 1587, 1236, 1042, 636 cm$^{-1}$; HRMS (ESI): [M-2Na]$^{2-}$ calculated for C$_{44}$H$_{46}$D$_6$N$_8$O$_{19}$S$_3$: 549.1449, found: 549.1449.

# $^1$H NMR and $^{13}$C NMR spectra

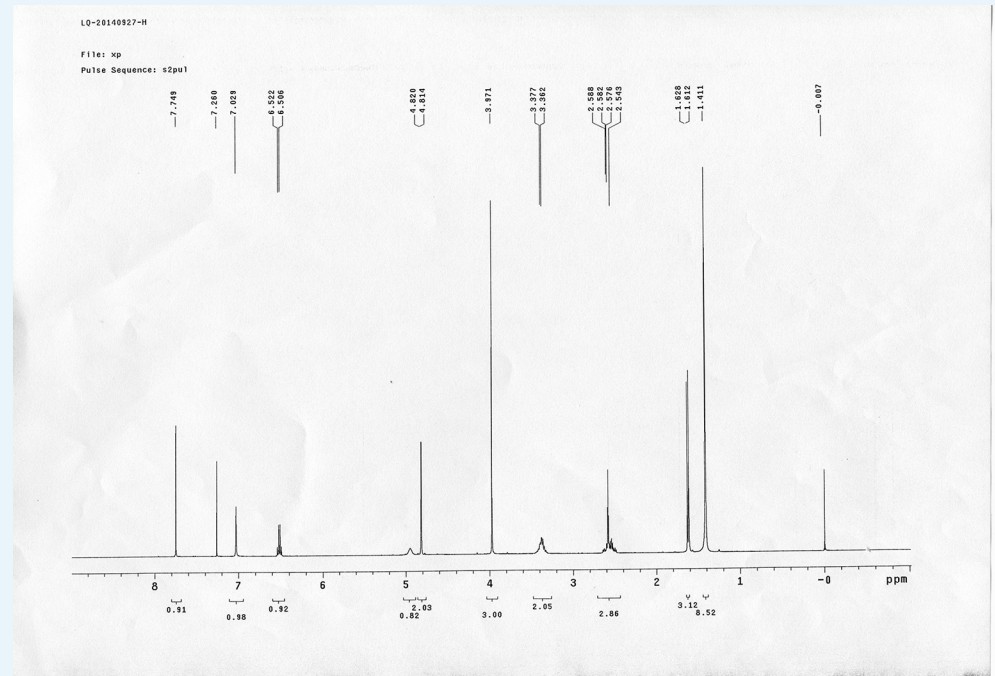

**Appendix 1—figure 26.** $^1$H NMR spectra of compound 6.

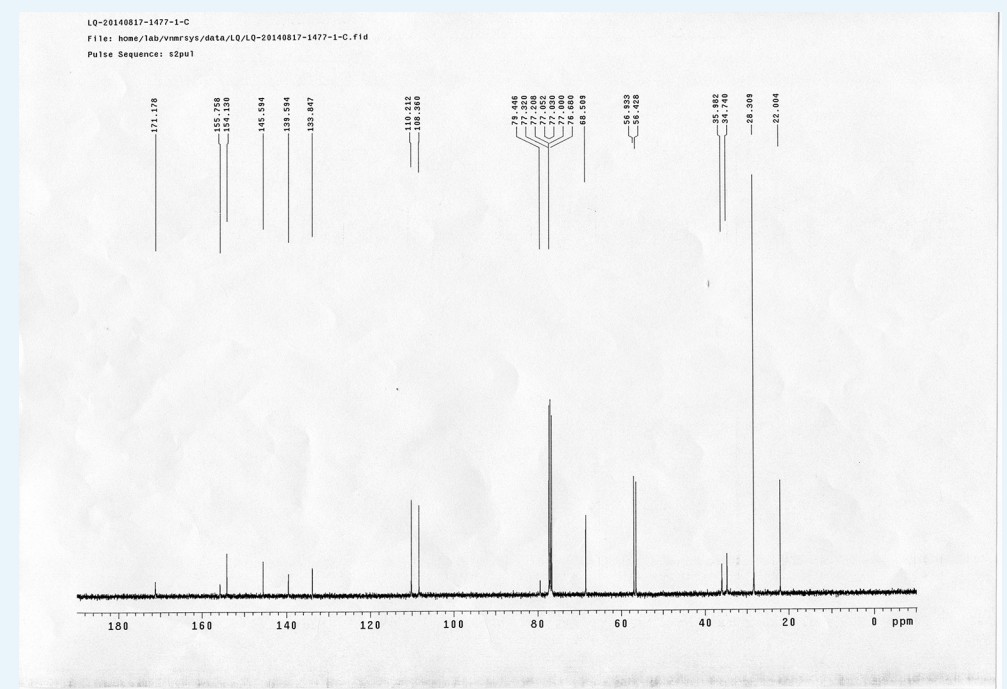

**Appendix 1—figure 27.** $^{13}$C NMR spectra of compound 6.

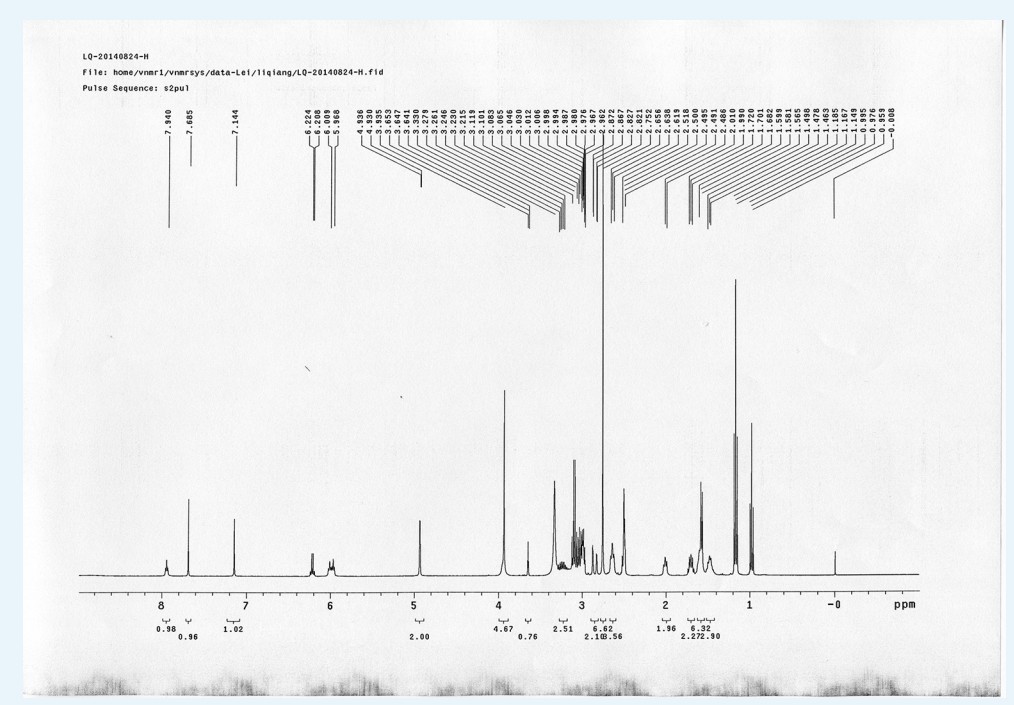

**Appendix 1—figure 28.** [1]H NMR spectra of sulfo-PL.

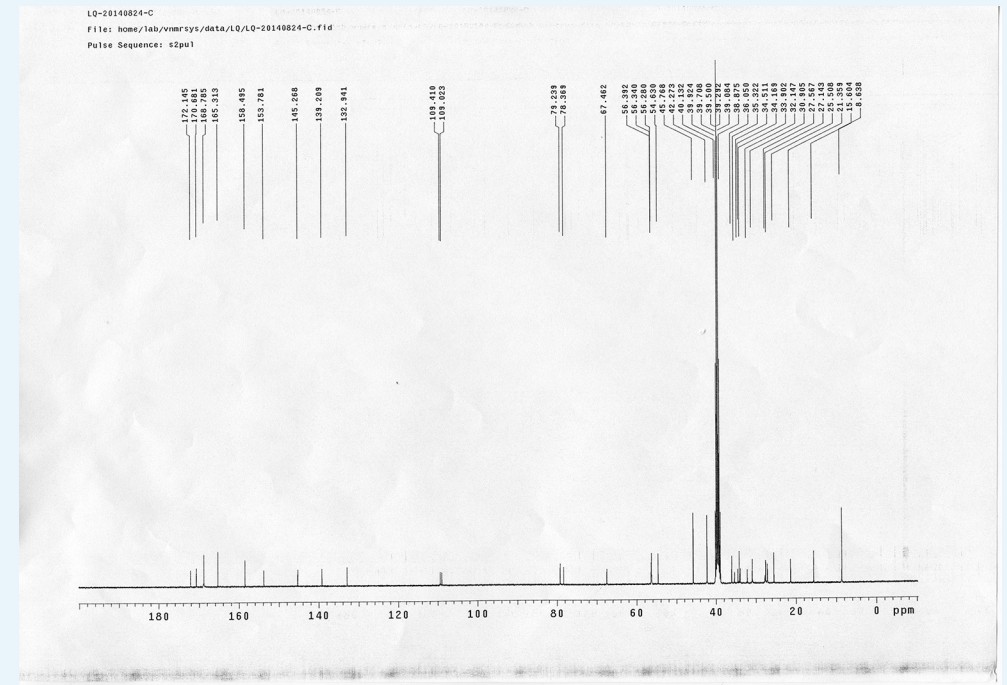

**Appendix 1—figure 29.** [13]C NMR spectra of sulfo-PL.

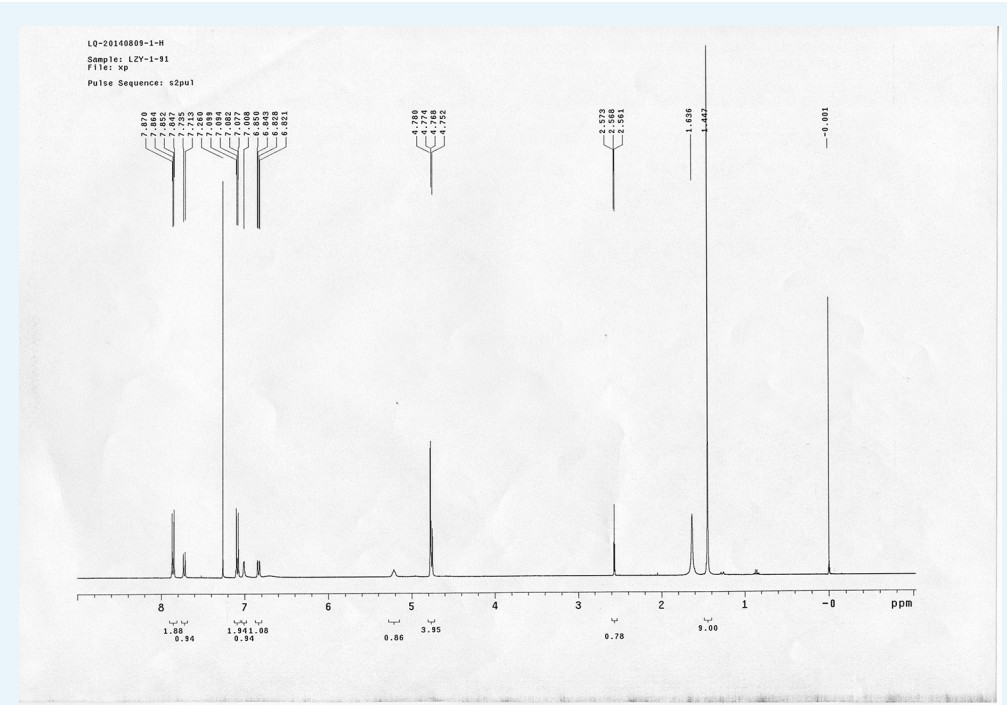

**Appendix 1—figure 30.** [1]H NMR spectra of compound 11.

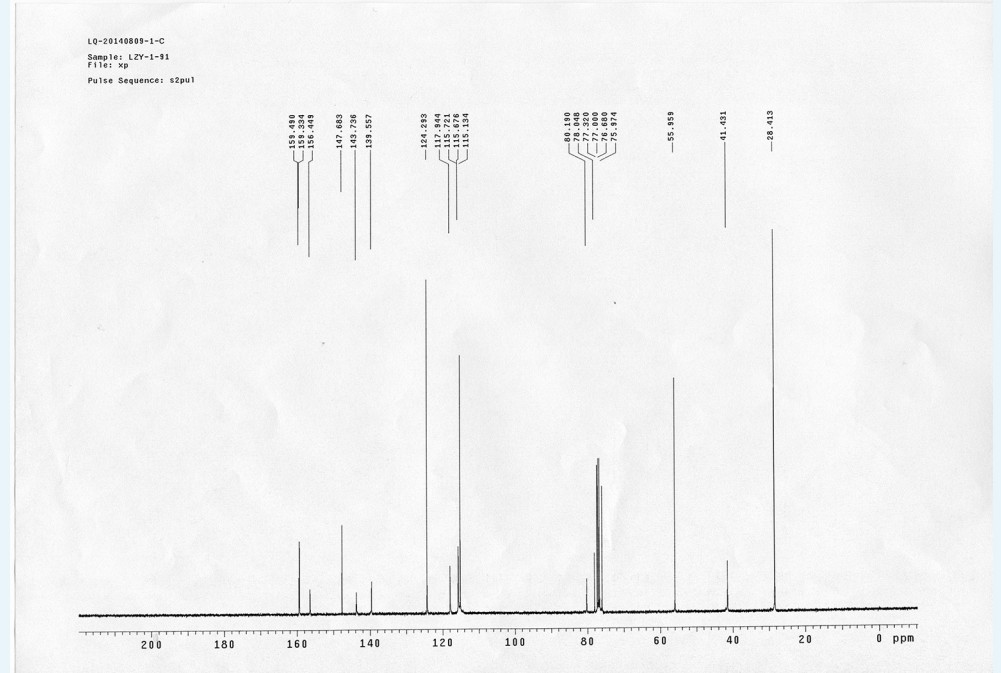

**Appendix 1—figure 31.** [13]C NMR spectra of compound 11.

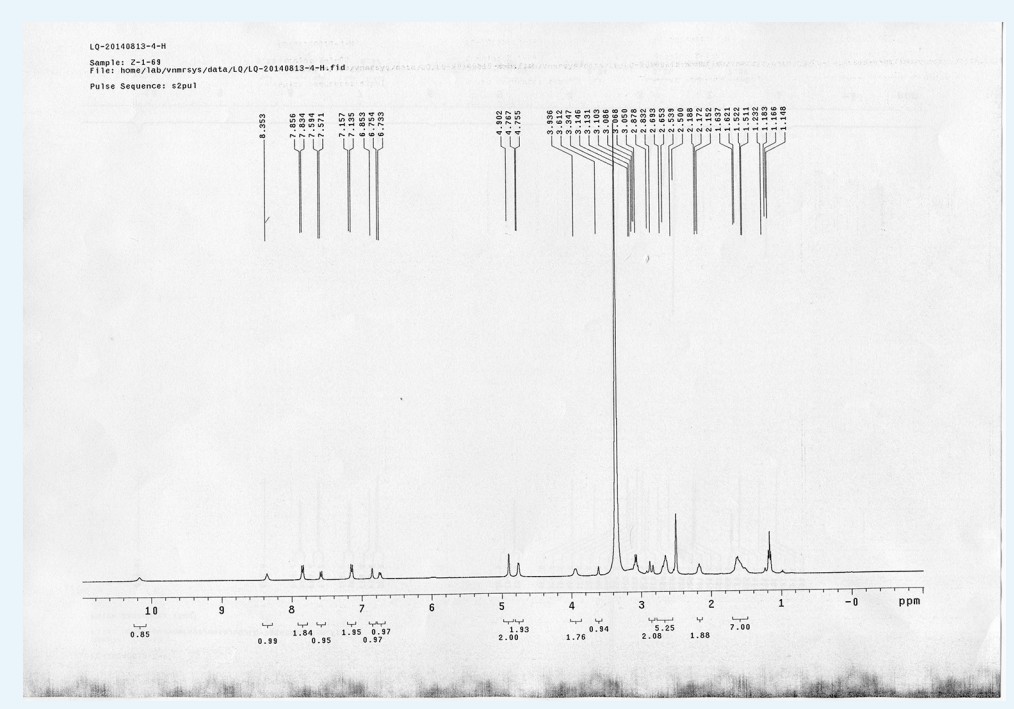

**Appendix 1—figure 32.** [1]H NMR spectra of AL.

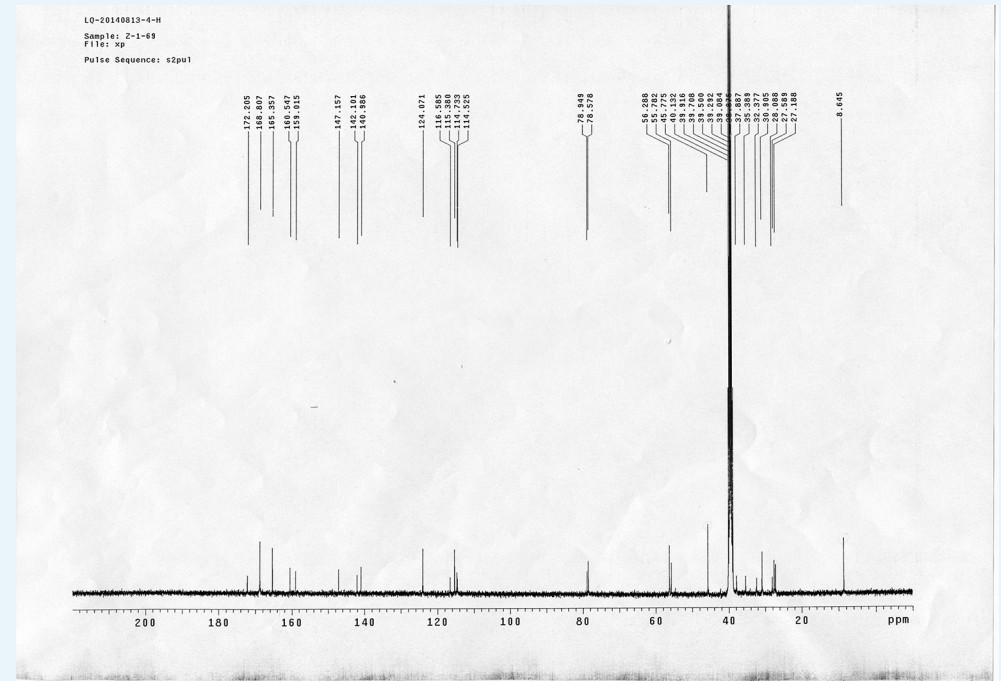

**Appendix 1—figure 33.** [13]C NMR spectra of AL.

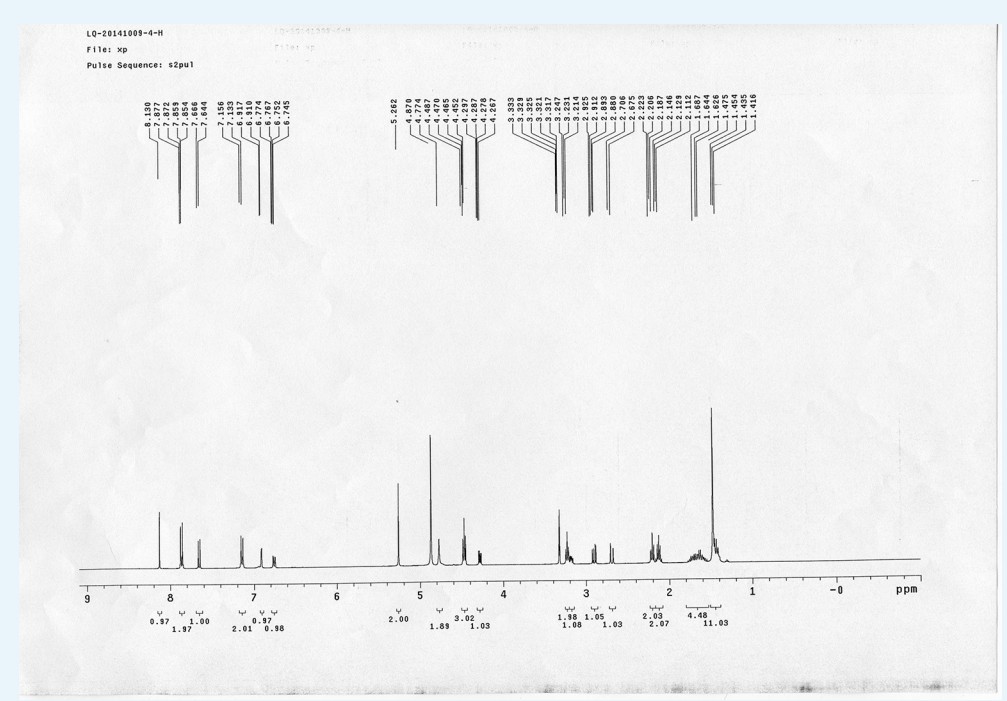

**Appendix 1—figure 34.** [1]H NMR spectra of compound 15.

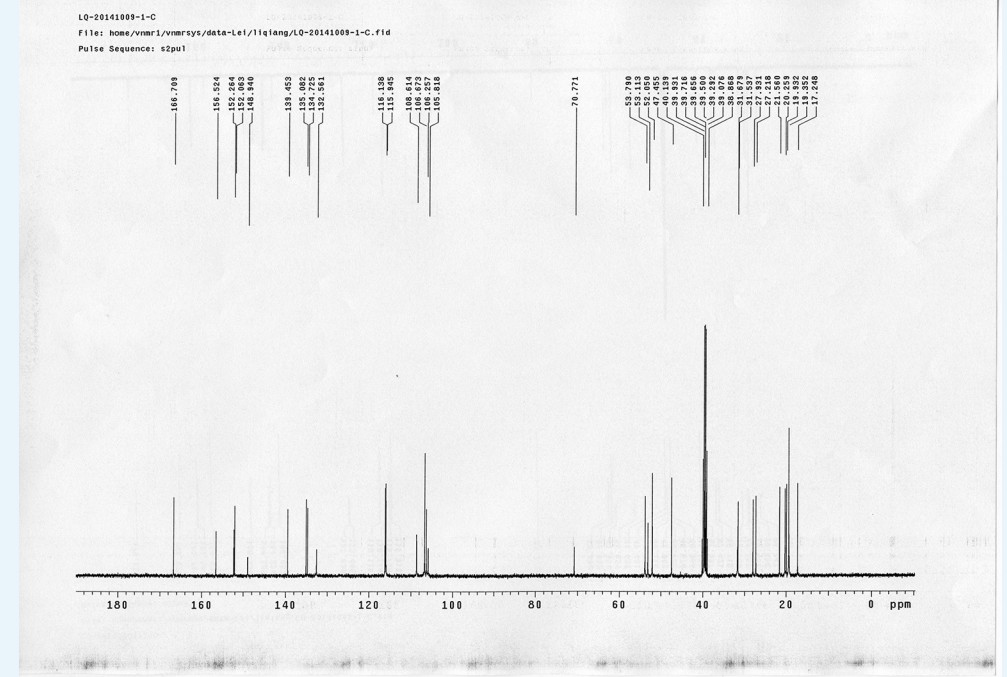

**Appendix 1—figure 35.** [13]C NMR spectra of compound 15.

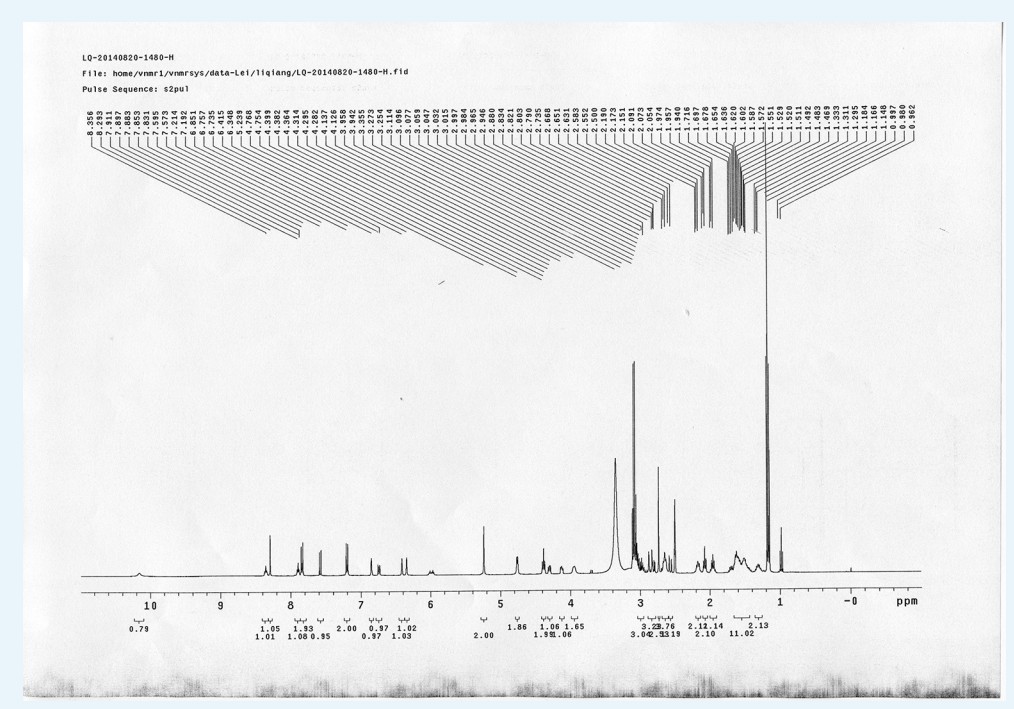

**Appendix 1—figure 36.** [1]H NMR spectra of bAL 1.

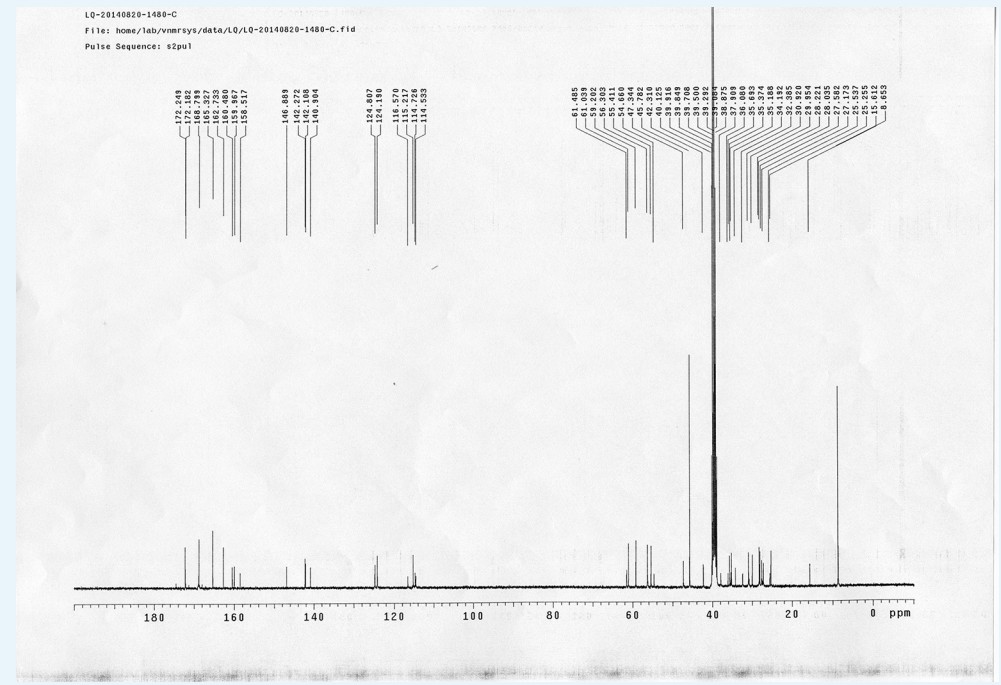

**Appendix 1—figure 37.** [13]C NMR spectra of bAL 1.

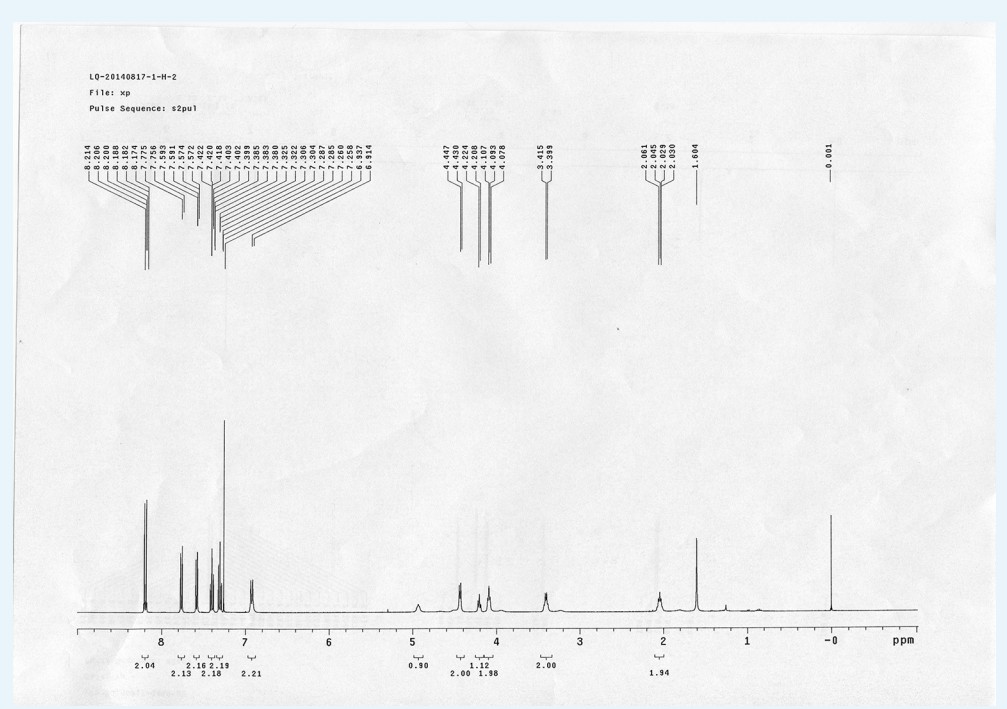

**Appendix 1—figure 38.** [1]H NMR spectra of compound 20.

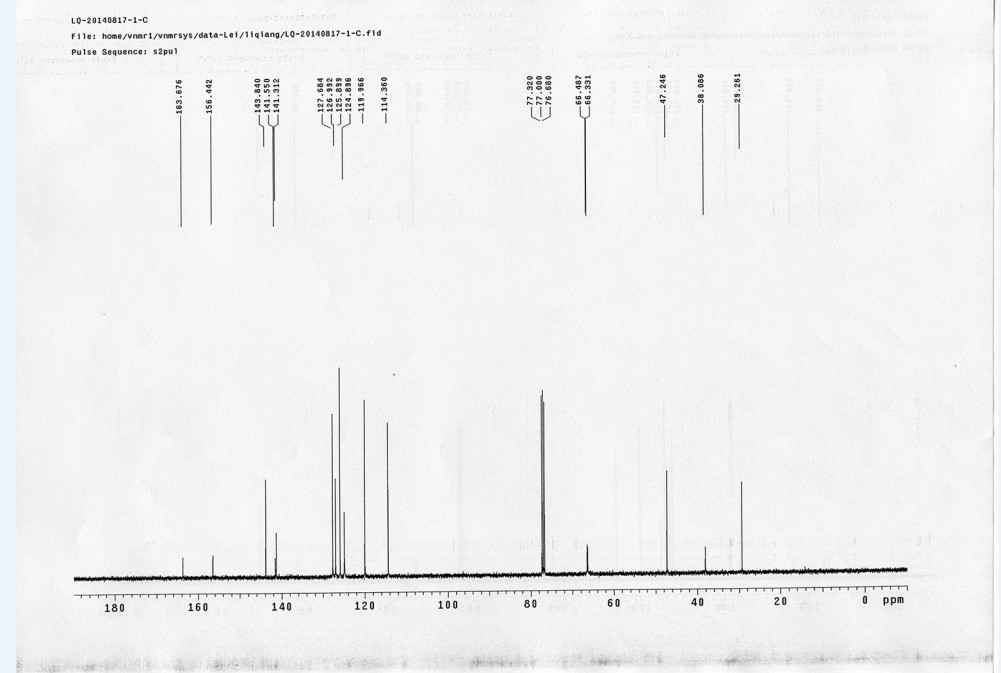

**Appendix 1—figure 39.** [13]C NMR spectra of compound 20.

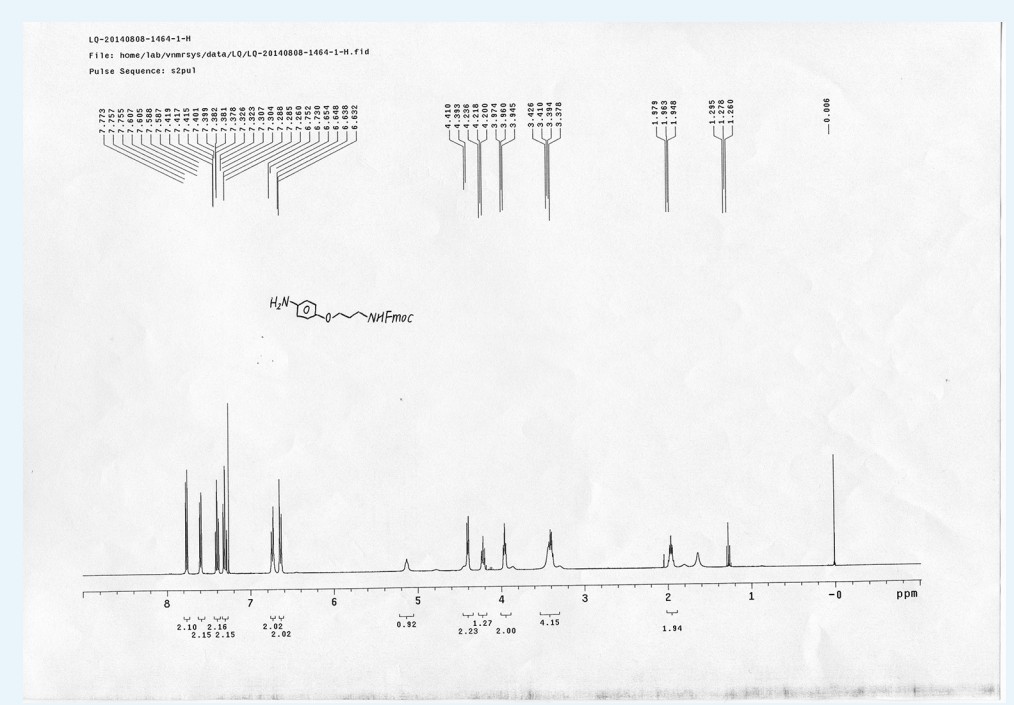

**Appendix 1—figure 40.** [1]H NMR spectra of compound 21.

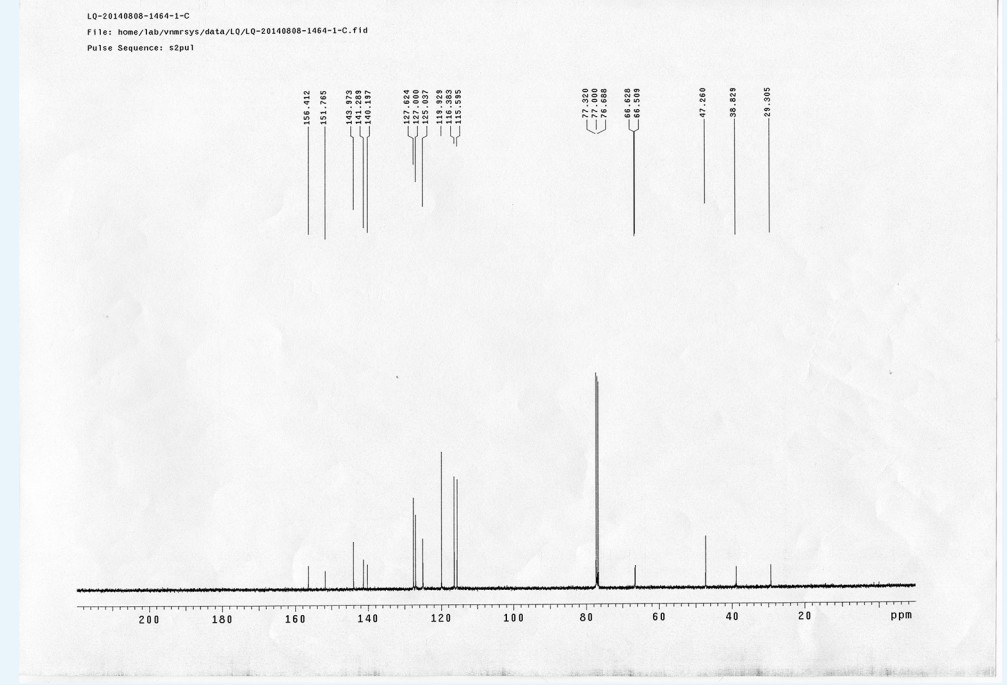

**Appendix 1—figure 41.** [13]C NMR spectra of compound 21.

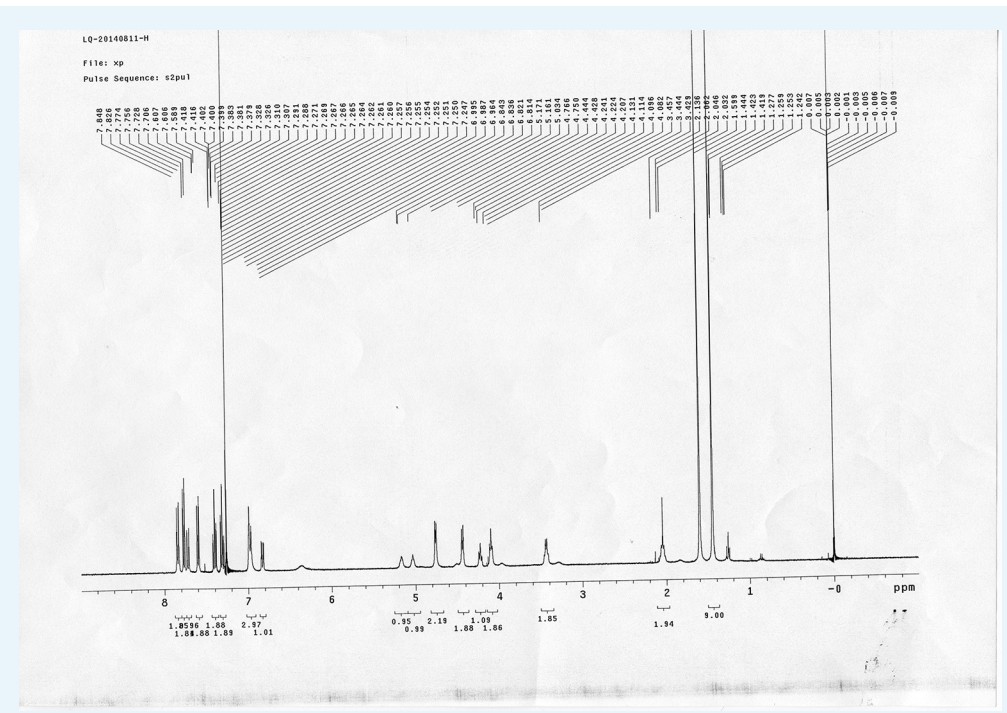

**Appendix 1—figure 42.** [1]H NMR spectra of compound 22.

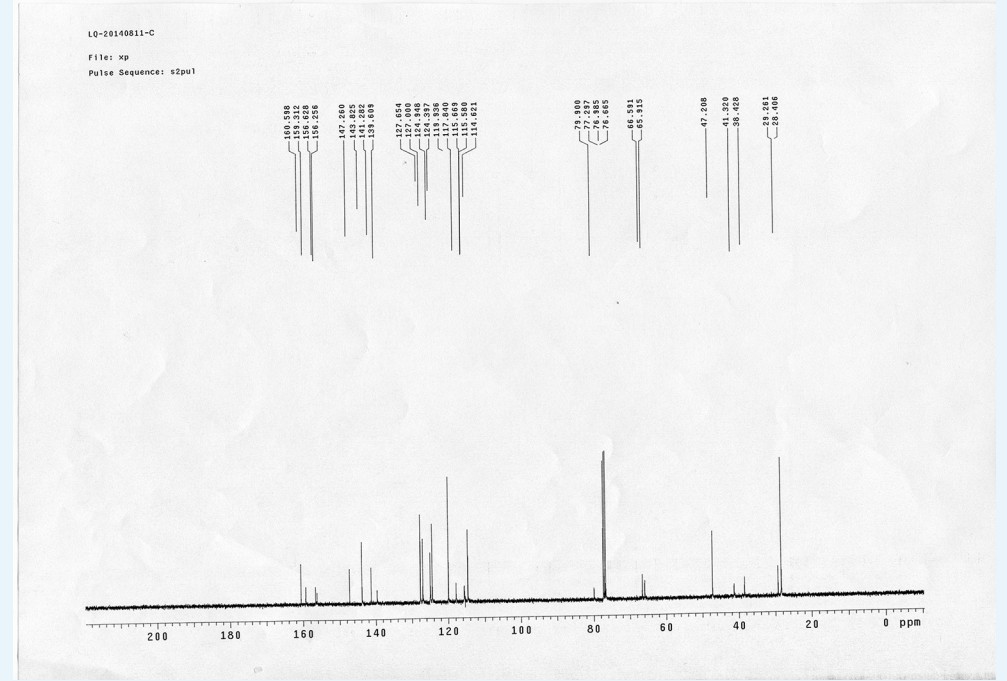

**Appendix 1—figure 43.** [13]C NMR spectra of compound 22.

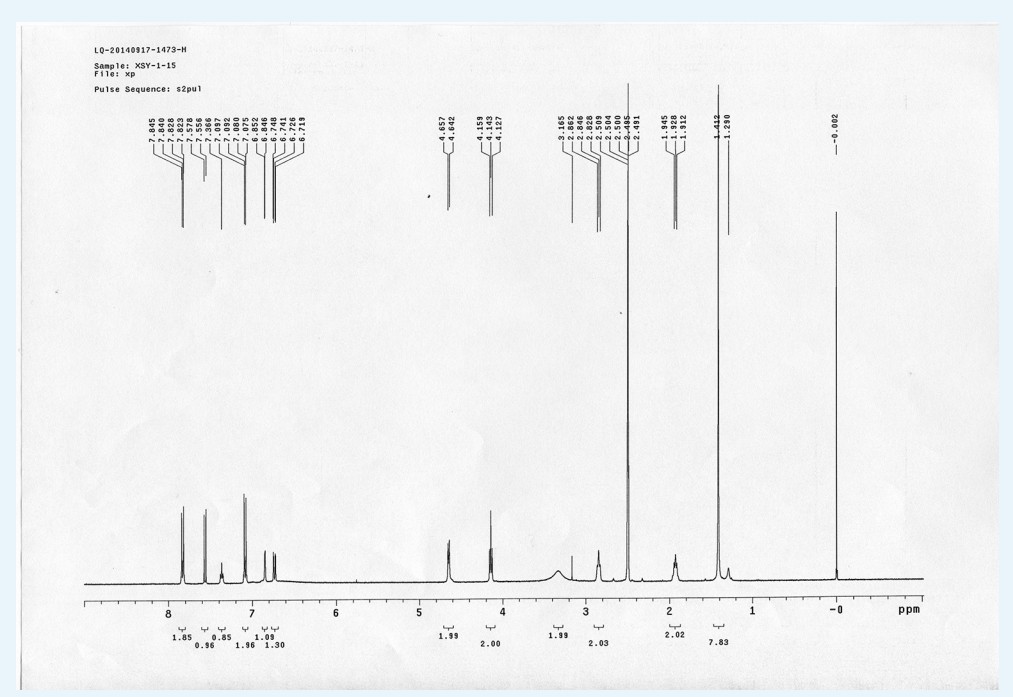

**Appendix 1—figure 44.** [1]H NMR spectra of compound 23.

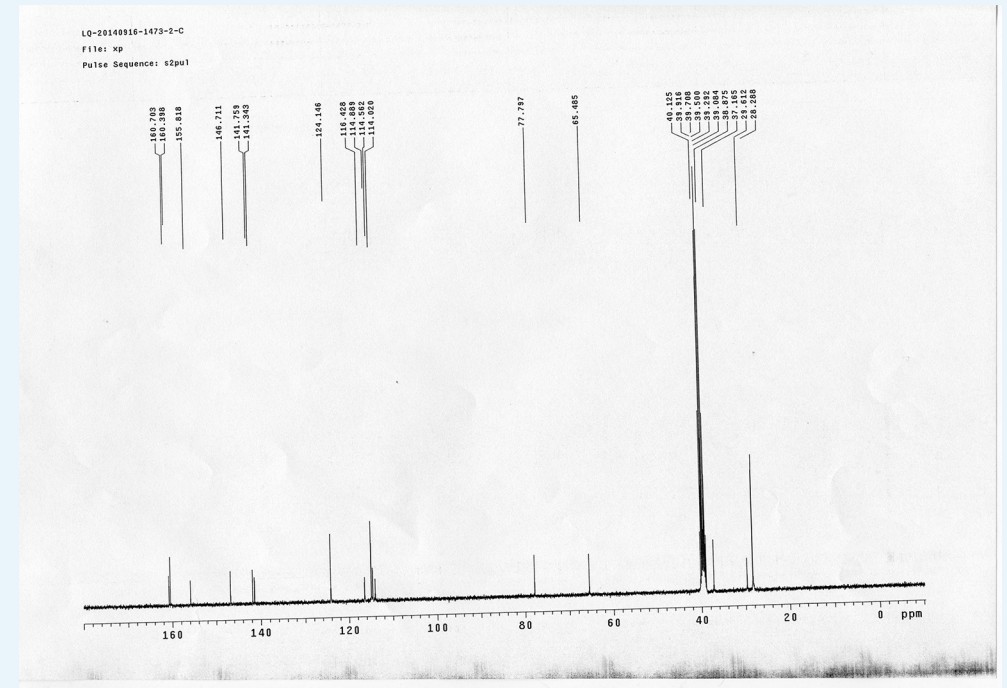

**Appendix 1—figure 45.** [13]C NMR spectra of compound 23.

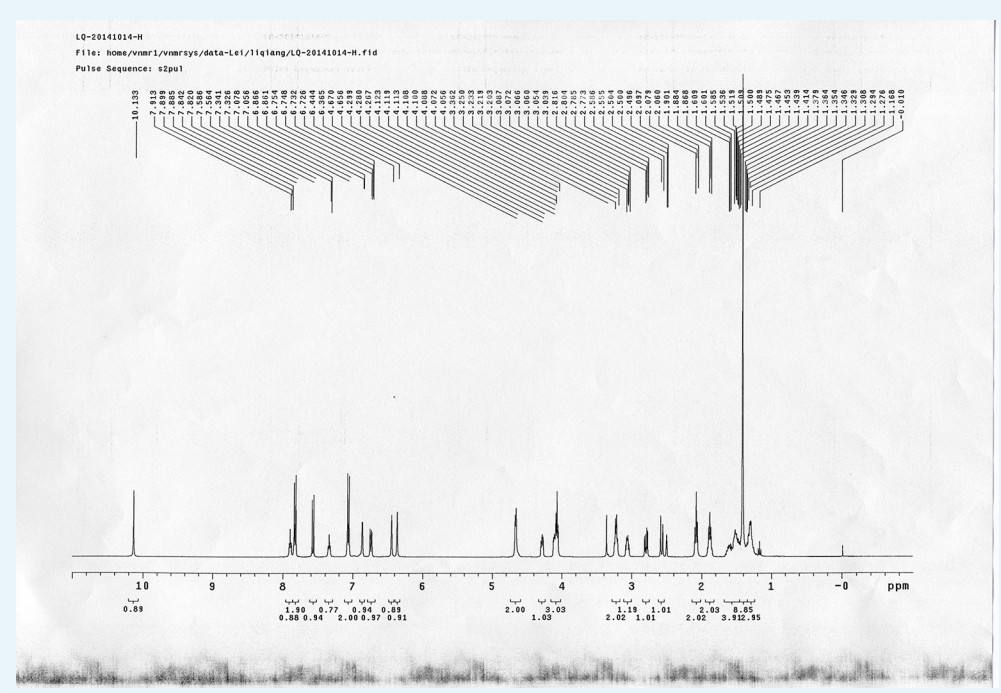

**Appendix 1—figure 46.** [1]H NMR spectra of compound 25.

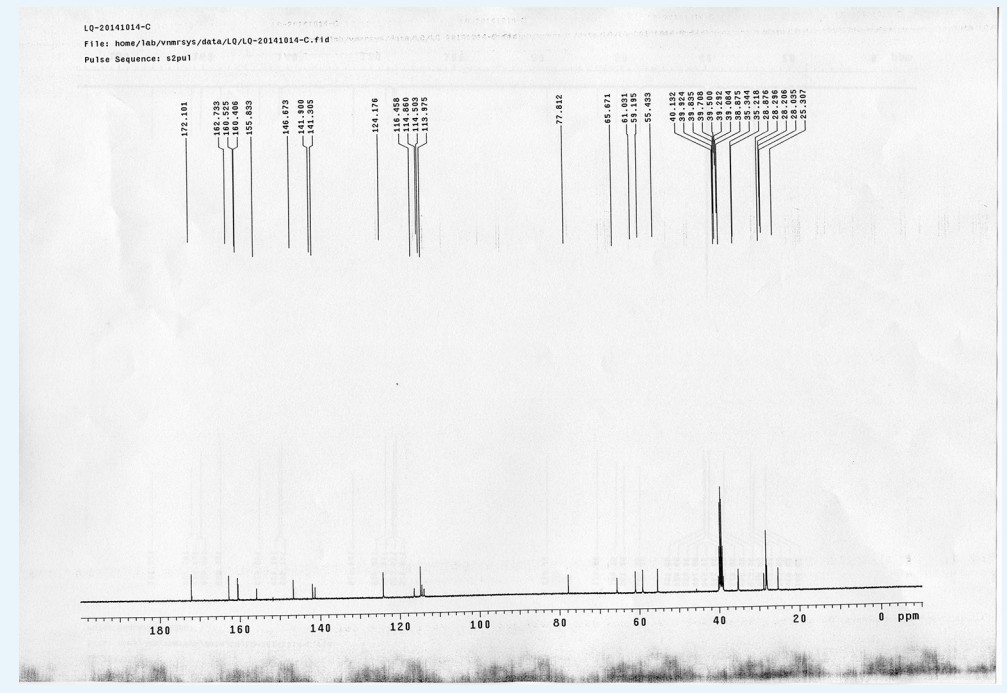

**Appendix 1—figure 47.** [13]C NMR spectra of compound 25.

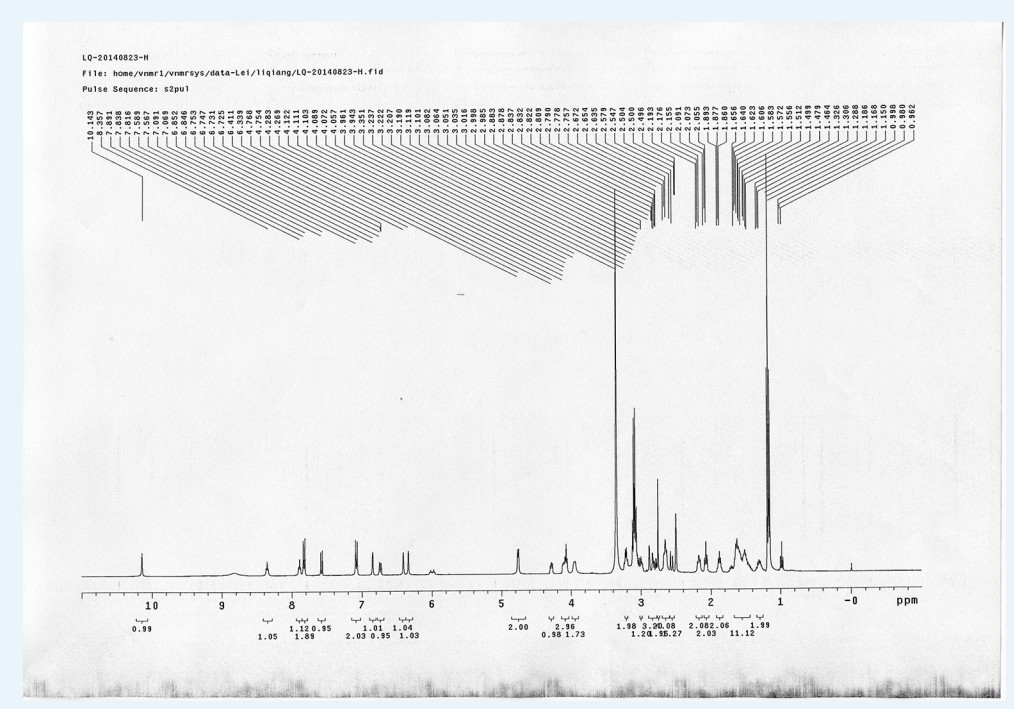

**Appendix 1—figure 48.** $^1$H NMR spectra of bAL 2.

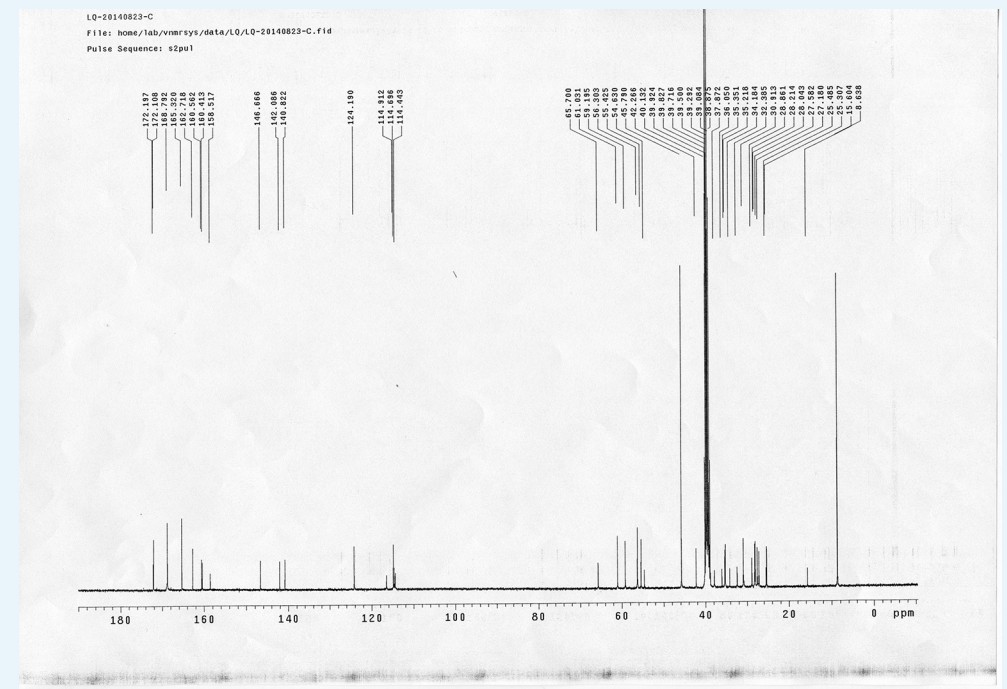

**Appendix 1—figure 49.** $^{13}$C NMR spectra of bAL 2.

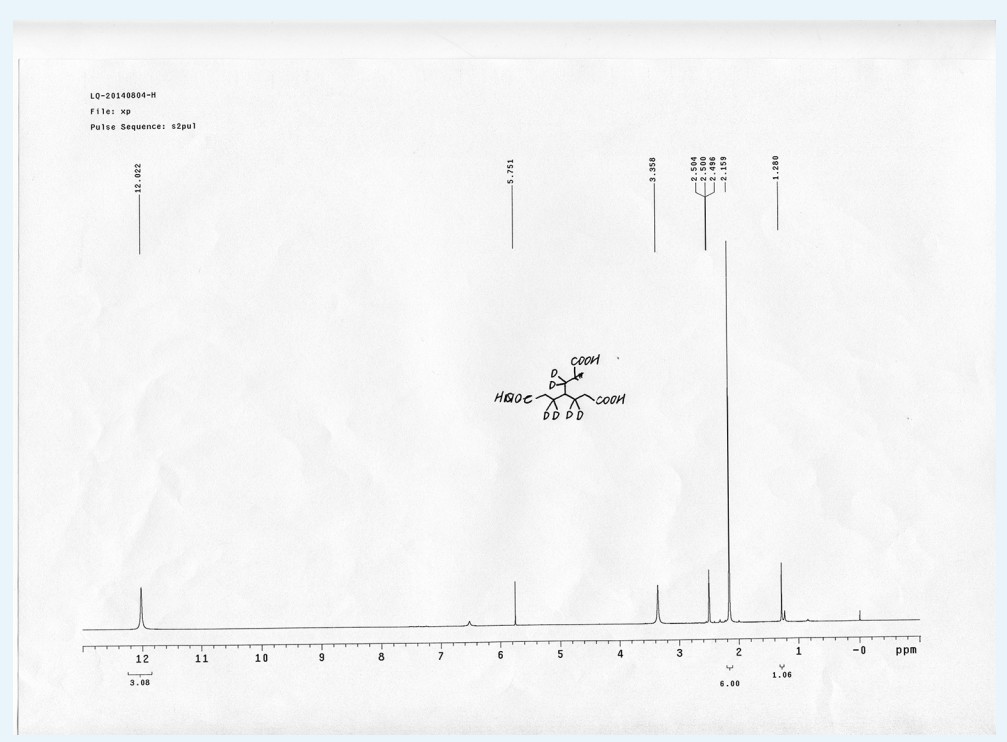

**Appendix 1—figure 50.** [1]H NMR spectra of compound 29.

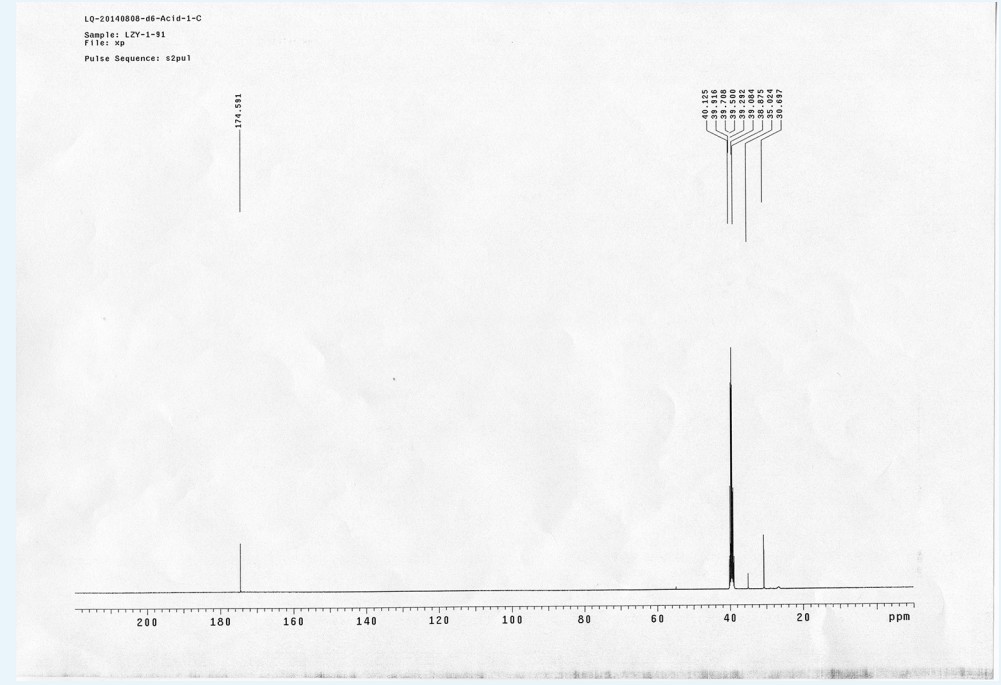

**Appendix 1—figure 51.** [13]C NMR spectra of compound 29.

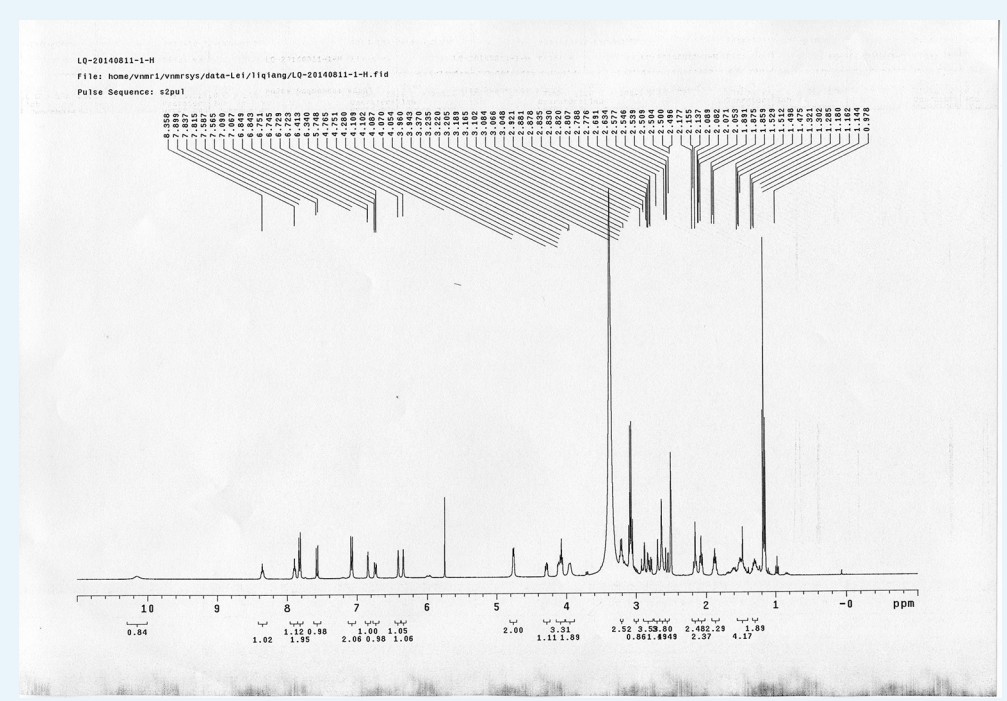

**Appendix 1—figure 52.** [1]H NMR spectra of d6-bAL 2.

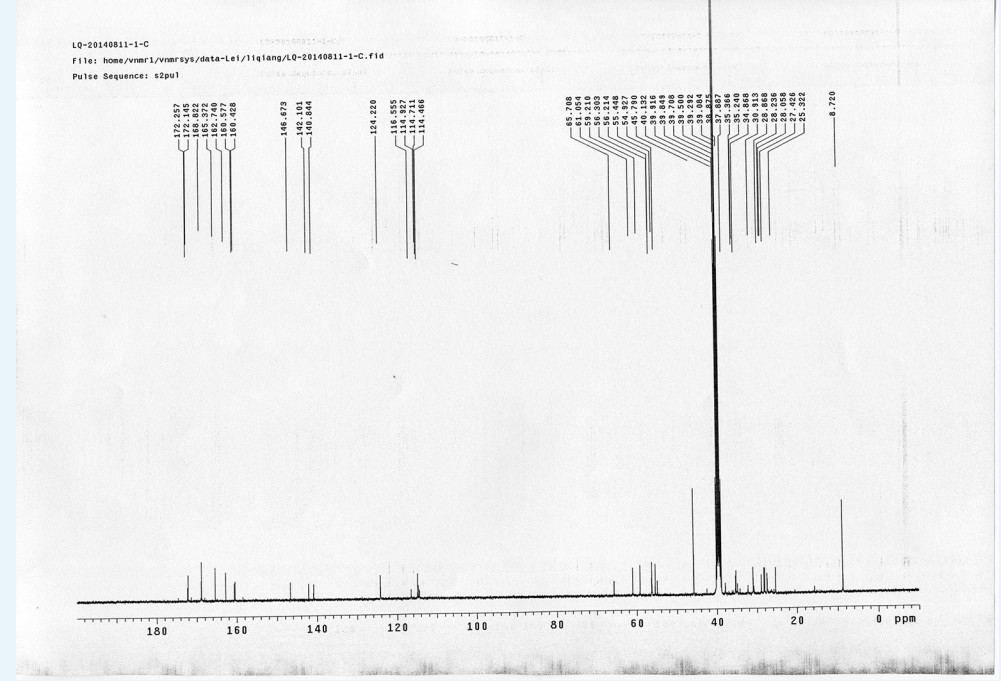

**Appendix 1—figure 53.** [13]C NMR spectra of d6-bAL 2.

