## [Decision Letter]

Thank you for submitting your work entitled "Leiker, A Cross-Linker for Mapping Protein-Protein Interaction Networks and Comparing Protein Conformational States" for consideration by *eLife*. Your article has been reviewed by four peer reviewers, one of whom was a guest editor and the evaluation has been overseen by John Kuriyan as the Senior Editor. One of the four reviewers has agreed to reveal his identity: Jeff Ranish.

The reviewers have discussed the reviews with one another and the Reviewing Editor has drafted this decision to help you prepare a revised submission.

Summary:

The authors describe the use of tri-functional crosslinking reagents for CX-MS studies with one of the reagent's functionalities being the biotin affinity tag and the other two being amine-reactive moieties. The basic idea here is to enrich crosslinked and crosslink-modified peptides from mixtures of peptides generated from protein assemblies. While this concept and its demonstration is not new, the value of the present study lies in its experimental tests and assessment of the utility of such trifunctional crosslinking reagents for enriching information-filled crosslink-modified peptides from a large background of unmodified peptides. In this respect, there are components of the manuscript that have considerable potential value (especially the section entitled "Leiker enabled robust enrichment of cross-linked peptides"). At the same time other components of the study are either not well enough described to have real value or the conclusions from these components are insufficiently justified (especially the section entitled "Application of Leiker to lysates"). Therefore, publication of this work can only be supported provided that the authors adequately address the detailed points listed below.

Detailed Comments:

1) With respect to the prior work the authors state "….as proposed previously (Trester-Zedlitz et al., 2003)." The paper referred to is a JACS article with a full description of a large range of modular trifunctional crosslinks of the kind repeated in the present study, so this earlier work cannot be considered merely a proposal, but rather a careful description and demonstration of the chemical synthesis of these reagents, as well as their application. The authors should more accurately represent their work in the light of the prior work, and certainly the re-naming of the strategy associated with the use of trifunctional crosslinking reagents as "Leiker" is inappropriate.

2) Also with respect to prior work, the authors apply trifunctional reagents to a study of crosslinking within exosome complexes isolated from *S. cerevisiae*. Exosome complexes have been studied previously in the Shi et al. 2015 paper that was referred to in the manuscript. In order to allow the reader to better assess the relative merits of the present approach versus those used formerly, the authors should compare and contrast their results with this prior work, which was performed with standard bifunctional crosslinking reagents. Points of interest here include a comparison of the number of inter-subunit crosslinks, the relative amounts of sample used, and any differences in the information gained in these two cases. This comparison should prove of greater value to the reader than the comparisons that are made in the manuscript, which are comparisons between the results provided by the author and early two-hybrid experiments as well as to just a single crosslink data point in the Shi et al. 2015 paper.

3) The authors state that there is a HCD reporter ion at *m/z* 122.0606 generated from their reagent (Figure 2). It would be helpful for the authors to state what use, if any, they make of this reporter ion and if it is always observed.

4) It is important to specify more precisely what is meant by "…a crude immunoprecipitate of TAP-tagged Rrp46" in terms of purity? What do the products of this affinity isolation look like on an SDS-PAGE gel or by mass spectrometric analysis?

5) The experiment shown in Figure 3 appears to be a convincing demonstration of the potential utility of the described technique for increasing the information content of crosslinking studies, especially in complex mixtures of proteins. I believe that this is the most convincing and valuable data that is provided by the authors. However, because the described technique involves additional steps to those carried out in more conventional crosslinking experiments, it is important to evaluate the cost in sensitivity (if any) that this gain in information content is accomplished. This is especially important for real world samples of affinity isolated complexes, where the amount of sample is often limiting. Thus, this study will be more valuable if the authors provide information on the data falloff as a function of decreasing the quantity of the ten protein mixture. Also, the individual amounts of each protein should be provided, not just the total amounts of all ten proteins and the names of the proteins should be supplied in the paper proper rather than just in the Supplementary data. Finally, because structures for most of these proteins are available, the α-carbon to α-carbon distance distribution for all the inter-links should be provided and assessed as a measure of x-link veracity.

6) The authors utilize identification of artefactual inter-protein crosslinks to assess the specificity of the trifunctional reagents versus BS_3_. It is also of interest to ascertain whether these larger reagents are more sterically hindered in accessing reactive amines. This requires comparing data obtained on standards such as the 10 proteins using e.g. BS_3_ versus the tri-functional reagents. Since this data already exists, it appears straightforward to make this comparison. Indeed from Figure 3 it appears that their technique yields approximately 50% less of crosslinks comparing to DSS. This point needs to be properly addressed.

7) In the experimental data shown in Figure 3, the authors found that even in a 100-fold excess of *E. coli* lysate over the protein standards approx. 97% of the identified peptides were crosslinked peptides – i.e., they observed something like a 100-fold enhancement of crosslinked peptides under these conditions. However in the *E. coli* lysate, the authors only observed a 4-fold increase in the crosslinks treated with tri-functional reagents compared to that treated with BS_3_ (Figure 3). The authors need to explain this observation? (This question is related to 16 below).

8) Crosslinks do not necessarily define direct protein-protein interactions due to their relatively long distance thresholds (here >25 angstroms), although in many cases they do. Thus, it is more accurate to interpret the data as defining "protein spatial connectivity" or "spatial restraints" instead.

9) The crosslink data appears sparse by current day standards. Discuss why you think this is the case. How much material was used in the ribosome crosslinking study?

10) Concerning the application of the described method to an analysis of crosslinking of proteins within lysates, it is of interest to provide data on the relative information provided in the three independent experiments to give the reader a feel for the amount of data that can be expected from a single experiment as well as their reproducibility.

The authors should further discuss 'half of the BS_3_ identified cross-links from *E. coli* were recapitulated in this study' (subsection “Application of Leiker to lysates”, second paragraph)? If after enrichment 50% of previously identified crosslinkers are not identified, does it mean that the generated crosslinks themselves vary dramatically or the enrichment is simply not exhaustive enough? For that reason the authors should state how much overlap there is between the crosslinks of the biological replicates.

11) Crosslink data analysis. For their large database searches, did the authors try to filter and estimate the FDRs separately for intersubunit and intersubunit crosslinks? (Trnka et al. MCP, 2014). If not, the authors could, for example, search their 70S dataset using two different databases: either a small database (containing only the ribosome proteins) or a large database (with hundreds of proteins) and compare the differences (mapped to the available atomic structure). In any event, the authors should to upload their inter-peptide spectra to allow the interested reader to assess the crosslinked peptides data quality (scores do not always allow this because the good score may be dominated by one of the two peptides within the pair).

12) It is in the first paragraph of the subsection “Application of Leiker to lysates” that most of the inter-molecule cross-links suggest novel protein-protein interactions. Exactly how was this assessed?

13) I find Figure 5 to be largely uninformative. The authors must find a more informative way of presenting this or part of this data. It seems likely that the observed inter-molecule interactions are strongly skewed by the relative abundance of the proteins in the lysates. An assessment of this data in terms of the abundance of the proteins could provide valuable information on the rules underlying the observation of interaction in such complex mixtures. Depending on the results, it might also be valuable to discuss other possible reasons for why these particular interactions are observed.

14) I could not make sense of the sentence “Applying the same procedure to the previously identified BS^3^ cross-links (Yang, et al., 2012), we obtained only three ribosomal proteins in the most highly connected modules (Figure 5—figure supplement 2)”, nor could I find Figure 5—figure supplement 2. Indeed I was unable to make proper sense of the second half of the subsection “Application of Leiker to lysates”. I also did not find the recitation of numbers in this section particularly useful or meaningful. The points that I think are ostensibly being made in this section need to be made much more clearly.

15) The data from the whole-proteome CX-MS analysis of *C. elegans* appears sparse (with just a few hundred crosslinks identified). What is the reason for this? (See also Q 16). Is this because of limitations of the software? If this is the case, perhaps it would help to use a smaller database containing only the top 1,000 most abundant proteins (since identifying crosslinks from the less abundant proteins would appear unlikely anyway)?

16) While I agree with the authors that better sample preparation and more sensitive instrumentation could help to reduce the crosslinking problem caused by the huge dynamic range of proteins within proteomes, I am not convinced that these are actually the most important barriers to overcome for proteome-wide crosslinking profiling for at least two reasons:

A) Within cells, the majority of protein complexes are not abundant. Thus, using the current crosslinking protocol without prior affinity enrichment of the complexes of interest, the majority of such complexes will not be efficiently crosslinked because of the low efficiency for crosslinking low abundance complexes.

B) And even if one could crosslink the proteome efficiently using more efficient reagents, glueing the cellular components together would likely lead to daunting challenges in inducing the needed proteolysis.

17) Figure 8. Reliable quantitative CX-MS analysis can be very challenging due to the low abundance of crosslink ions. What is the reproducibility of the quantitative data? Error bars from the repeat experiments would help here. Indeed without stronger quantitative data, it is not justified to entitle the paper: "Leiker, A Cross-Linker for Mapping Protein-Protein Interaction Networks and comparative cross-link analysis" at least with respect to the "aspect of comparative cross-link analysis".

[Editors' note: further revisions were requested prior to acceptance, as described below.]

Thank you for resubmitting your work entitled "Leiker, A Cross-Linker for Mapping Protein-Protein Interaction Networks and Comparing Protein Conformational States" for further consideration at *eLife*. Your revised article has been favorably evaluated by John Kuriyan (Senior editor) and a Reviewing editor. The manuscript has been improved but there are some remaining issues that need to be addressed before acceptance, as outlined below:

Concerning the title, it is not acceptable to us that you use the acronym LEIKER (Lysine-targeted enrichable cross-linker) in the title or Abstract of the paper. We note that you have now acceded to the request to properly attribute the 2003 JACS article, which previously described trifunctional reagents with the same basic properties as the present trifunctional reagent. We note that you still wish to name your process by an acronym (Lysine-targeted enrichable cross-linker (Leiker) even though this was thought to be inappropriate by the reviewers. The reviewers' objection to the acronym was that in giving a new name to a previously published strategy, you might effectively take possession of an idea that did not originate with the present work. Therefore, in order for this work to be published in *eLife*, you may use the acronym LEIKER in the main body of the paper, but not in the title or the Abstract.

---

## [Author Response]

*Detailed Comments: 1) With respect to the prior work the authors state "….as proposed previously (Trester-Zedlitz et al., 2003)." The paper referred to is a JACS article with a full description of a large range of modular trifunctional crosslinks of the kind repeated in the present study, so this earlier work cannot be considered merely a proposal, but rather a careful description and demonstration of the chemical synthesis of these reagents, as well as their application. The authors should more accurately represent their work in the light of the prior work, and certainly the re-naming of the strategy associated with the use of trifunctional crosslinking reagents as "Leiker" is inappropriate.* A) Leiker is not a collective term of all trifunctional cross-linkers; it is one type of design that we selected out of many and it truly meets the expectation for trifunctional cross-linkers. In the winner design, the azobenzene-based cleavage site has never been applied to the synthesis of cross-linkers previously. What distinguishes this study from other similar ones is the extensive, meticulous testing of multiple trifunctional cross-linkers and finding the best one for CXMS users. It is a heavy burden to test different cross-linkers. For example, we spent two years on the two-piece design before giving it up. No users would have the patience to test multiple cross-linkers in depth when they really want is one that works.

B) We have changed “…as proposed previously” to “…as pioneered previously”.

2) Also with respect to prior work, the authors apply trifunctional reagents to a study of crosslinking within exosome complexes isolated from S. cerevisiae. Exosome complexes have been studied previously in the Shi et al.

*2015 paper that was referred to in the manuscript. In order to allow the reader to better assess the relative merits of the present approach versus those used formerly, the authors should compare and contrast their results with this prior work, which was performed with standard bifunctional crosslinking reagents. Points of interest here include a comparison of the number of inter-subunit crosslinks, the relative amounts of sample used, and any differences in the information gained in these two cases. This comparison should prove of greater value to the reader than the comparisons that are made in the manuscript, which are comparisons between the results provided by the author and early two-hybrid experiments as well as to just a single crosslink data point in the Shi et al. 2015 paper.* We made a comparison of the two studies (Author response Table 1). Shi et al. used 15 μg of highly purified exosome complexes (44 proteins identified), and we used 40 μg of crude exosome complexes (740 proteins identified). The sample analyzed by Shi et al. included both cytosolic and nuclear exosomes and ours had only cytosolic exosome, with the overlap being the exosome core complex. Thus, the cross-links involving the ten exosome core subunits were compared. At FDR < 0.05, 59 out of 88 cross-links reported by Shi et al. were detected in this study (Figure 62).

Author response table 1.Comparison of this study and Shi et al. study of the exosome complex.**DOI:**
http://dx.doi.org/10.7554/eLife.12509.093LeikerDSS (Shi et al.)filtering criteriaFDR < 0.05FDR < 0.05,E-value < 0.00001, spectral count ≥ 3FDR < 0.05,manual inspection#total cross-links625195211#inter-subunit cross-links3624379#intra-subunit cross-links263152132amount of sample40 μg15 μg#proteins in the sample740 (FDR 0.1%)44 (FDR ~1%)sample puritypoorgoodtype of exosomecytosolic exosome(in the Rrp6 deletion background)cytosolic exosomeand nuclear exosomedatabase for pLink search740 proteins17 proteins

Author response image 1.Overlap of inter-links between subunits of the exosome core complex.**DOI:**
http://dx.doi.org/10.7554/eLife.12509.094

*3) The authors state that there is a HCD reporter ion at m/z 122.0606 generated from their reagent (Figure 2). It would be helpful for the authors to state what use, if any, they make of this reporter ion and if it is always observed.*

The *m/z* 122.0606 reporter ion is always observed with high intensity in the MS2 spectra of peptides modified by Leiker (after their release from the streptavidin beads by cleavage of azobenzene). It is either not observed or observed with low intensity in the MS2 spectra of regular peptides (not modified by Leiker) due to contamination of co-eluting Leiker-modified peptides in the 2-Da isolation window of precursor ions for MS2. The Intensity distribution is shown in Figure 63. The *m/z* 122.0606 reporter ion serves to verify the identification of Leiker-cross-linked peptides. It can also be used to filter spectra, however, we did not use it this way because after enrichment using streptavidin beads, the resulting spectra are predominantly of Leiker-modified peptides. We added a sentence to the manuscript to state its use.

Author response image 2.Intensity distributions of the reporter ion of *m/z* 122.0606 for different types of peptides.In each category, the FDR of peptide-spectrum match (PSM) is 5%. The data came from the ten standard proteins.**DOI:**
http://dx.doi.org/10.7554/eLife.12509.095

*4) It is important to specify more precisely what is meant by "…a crude immunoprecipitate of TAP-tagged Rrp46" in terms of purity? What do the products of this affinity isolation look like on an SDS-PAGE gel or by mass spectrometric analysis?*

For comparison, a SDS-PAGE gel of the crude immunoprecipitate of

TAP-tagged Rrp46 used in this study (left) is shown next to the SDS-PAGE of the purified exosome sample by Shi et al. (right) (Figure 64). The purity of the exosome sample used by Shi et al. is far better. In fact, a total of 740 (protein FDR 0.1%) and 44 (protein FDR ~1%) proteins were identified, respectively, in our sample and in the sample of Shi et al.. This is described in the Methods section, under the subsection “Identification of cross-linked peptides with pLink”.

Author response image 3.Silver-stained SDS-PAGE of the crude immunoprecipitate of TAP-tagged Rrp46 (left) and the SDS-PAGE of the purified exosome sample of Shi et al. (right).**DOI:**
http://dx.doi.org/10.7554/eLife.12509.096

*5) The experiment shown in Figure 3 appears to be a convincing demonstration of the potential utility of the described technique for increasing the information content of crosslinking studies, especially in complex mixtures of proteins. I believe that this is the most convincing and valuable data that is provided by the authors. However, because the described technique involves additional steps to those carried out in more conventional crosslinking experiments, it is important to evaluate the cost in sensitivity (if any) that this gain in information content is accomplished. This is especially important for real world samples of affinity isolated complexes, where the amount of sample is often limiting. Thus, this study will be more valuable if the authors provide information on the data falloff as a function of decreasing the quantity of the ten protein mixture. Also, the individual amounts of each protein should be provided, not just the total amounts of all ten proteins and the names of the proteins should be supplied in the paper proper rather than just in the Supplementary data. Finally, because structures for most of these proteins are available, the α-carbon to α-carbon distance distribution for all the inter-links should be provided and assessed as a measure of x-link veracity.*

A) We thank the reviewers for the suggestion of testing the sensitivity of our method—such information will be very valuable for users. From ourexperience, BS_3_ or DSS is a better choice for low-complexity samples of limiting amounts; Leiker is superior for high-complexity samples of adequate amounts; nothing works for minute amounts of high-complexity samples. We used the immunoprecipitated exosome sample, instead of the ten-protein mixture, for the sensitivity test for two reasons: 1. The ten-protein mix is too simple to mimic real-world samples of immunoprecipitated proteins; 2. as shown in Figure 3 dilution), Leiker has little advantage on this simple protein mixture over BS_3_. The input amount of the exosome sample was varied from 40 μg to 3 μg. We found that the number of inter-link identifications stayed about the same as the input decreased from 40 to 20 μg. A sharp decrease was observed when the input was reduced from 20 to 10 μg or lower. This information is added to the revised manuscript as Figure 4—figure supplement 6.

B) In [Supplementary-material SD1-data], we have added the following information: (1) the ten proteins were mixed at equal amounts by mass; (2) the pdb codes. Taking the undiluted ten-protein mix as an example, of the 61 identified Leiker cross-links that can be mapped to the pdb structures, 82% have Cα – Cα distance ≤22 Å and 93% have Cα – Cα distance ≤30 Å (FDR < 5%, E-value < 0.01). Of the 48 BS_3_ cross-links that can be mapped to the pdb structures, 94% have Cα – Cα distance ≤24 Å and 96% have Cα – Cα distance ≤ 30 Å (FDR < 5%, E-value < 0.01). This information has been added to [Supplementary-material SD2-data] and Figure 3—figure supplement 1.

6) The authors utilize identification of artefactual inter-protein crosslinks to assess the specificity of the trifunctional reagents versus BS_3_. It is also of interest to ascertain whether these larger reagents are more sterically hindered in accessing reactive amines. This requires comparing data obtained on standards such as the 10 proteins using e.g. BS_3_

*versus the tri-functional reagents. Since this data already exists, it appears straightforward to make this comparison. Indeed from Figure 3 it appears that their technique yields approximately 50% less of crosslinks comparing to DSS. This point needs to be properly addressed.*

It is a theoretical concern that Leiker may be more sterically hindered in accessing reactive amines because it is larger than BS_3_. However, we did not find clear evidence for it. Using the cross-links from the ten-protein mix (FDR < 0.05, E-value < 0.01), we find that the overlap between two BS_3_ experiments is similar to the overlap between a BS_3_ experiment and a Leiker experiment (Figure 65). The decrease in Leiker cross-links without enrichment might have to do with the negative effect of the biotin group, which is removed after enrichment. We are of the opinion that most of the differences are due to variations unrelated to cross-linking chemistry.

Author response image 4.Overlap between two CXMS experiments using BS_3_ or Leiker on the same 10-protein mix.**DOI:**
http://dx.doi.org/10.7554/eLife.12509.097

*7) In the experimental data shown in Figure 3, the authors found that even in a 100-fold excess of E. coli lysate over the protein standards approx. 97% of the identified peptides were crosslinked peptides – i.e., they observed something like a 100-fold enhancement of crosslinked peptides under these conditions. However in the E. coli lysate, the authors only observed a 4-fold increase in the crosslinks treated with tri-functional reagents compared to that treated with BS_3_*

*(Figure 3). The authors need to explain this observation? (This question is related to 16 below).*

The difference is due to the experimental design. In the experiment of Figure 3, the tryptic digest of a cross-linked ten-protein mixture was diluted with the tryptic digest of a non-cross-linked *E. coli* lysate. In the experiment of Figure 3, it’s the *E. coli* lysate that was cross-linked. The amount of non-cross-linked peptides in the first experiment greatly exceeded that in the second one. Because these non-cross-linked peptides were depleted after enrichment, the first experiment saw a greater enhancement in cross-link identification.

*8) Crosslinks do not necessarily define direct protein-protein interactions due to their relatively long distance thresholds (here >25 angstroms), although in many cases they do. Thus, it is more accurate to interpret the data as defining "protein spatial connectivity" or "spatial restraints" instead.*

We have replaced “direct protein-protein interactions” with “putative direct protein-protein interactions”. The term of protein spatial connectivity is very accurate but too technical for a general readership.

*9) The crosslink data appears sparse by current day standards. Discuss why you think this is the case. How much material was used in the ribosome crosslinking study?*

A) We are not sure what standards the reviewers are referring to. The number of *E. coli* cross-links identified in this study is four times greater than the number of PIR-identified inter-links (Chavez, et al., 2013) and eight times greater than the number of BS_3_-identified inter-links (Yang, et al., 2012). The number of cross-links identified from *C. elegans* is fewer that from *E. coli*. Among the possible reasons are greater sample complexity, the dominance of vitellogenin and cytoskeletal proteins in *C. elegans* lysates, fewer replicate experiments, less extensive peptide fractionation, and a search space several orders of magnitude larger.

B) 30 μg of ribosomes were used in each replicate experiment.

10) Concerning the application of the described method to an analysis of crosslinking of proteins within lysates, it is of interest to provide data on the relative information provided in the three independent experiments to give the reader a feel for the amount of data that can be expected from a single experiment as well as their reproducibility. The authors should further discuss 'half of the BS_3_

*identified cross-links from E. coli were recapitulated in this study' (subsection “Application of Leiker to lysates”, second paragraph)? If after enrichment 50% of previously identified crosslinkers are not identified, does it mean that the generated crosslinks themselves vary dramatically or the enrichment is simply not exhaustive enough? For that reason the authors should state how much overlap there is between the crosslinks of the biological replicates.*

A) The biological replicates of the *E. coli* whole-cell lysate yielded 1407, 1230, and 1474 cross-linked lysine pairs (FDR < 0.05, E-value < 0.01, and spectral count ≥ 3, same below), respectively, with 737 detected in all three replicates. The biological replicates of the *E. coli* ribo-free lysate resulted in 1512, and 1745 cross-linked lysine pairs, respectively, with an overlap of 1286 cross-links. This information is added to the revised manuscript as Figure 5—figure supplement 2.

B) We think that inadequate sampling of LC-MS/MS experiments is the major reason why half of BS_3_ cross-links of *E. coli* proteins were not identified using Leiker. In any sample there is always a population of cross-linked peptides that are barely detectable, and they may or may not be identified in an LC-MS/MS analysis. When identified, they typically have only one or a few spectral counts. As shown in Figure 65 (left panel), even for a sample containing only ten standard proteins, the overlap between two BS_3_ cross-linking experiments is around 50%. The overlap improves if a sample is analyzed repeatedly (technical repeats). For example, on *E. coli* lysates, the overlap between two ribo-free data sets, each with three technical repeats, is much better than the overlap between two whole-cell data sets, each with two technical repeats.

11) Crosslink data analysis. For their large database searches, did the authors try to filter and estimate the FDRs separately for intersubunit and intersubunit crosslinks? (Trnka et al.

*MCP, 2014). If not, the authors could, for example, search their 70S dataset using two different databases: either a small database (containing only the ribosome proteins) or a large database (with hundreds of proteins) and compare the differences (mapped to the available atomic structure). In any event, the authors should to upload their inter-peptide spectra to allow the interested reader to assess the crosslinked peptides data quality (scores do not always allow this because the good score may be dominated by one of the two peptides within the pair).*

A) The inter-subunit and intra-subunit cross-links were filtered together.

B) In the manuscript, MS2 spectra were searched against a small database consisting of 54 ribosomal proteins and 8 ribosome-associated proteins that were identified in the sample (54 + 8 = 62 proteins). As suggested, we added 500 other *E. coli* proteins to the database, repeated the search, and identified 192 cross-linked lysine pairs, 190 of which have been identified in the small database search (Figure 66). The two unique cross-links identified in the large database search are between a ribosomal protein and a non-ribosomal protein, and cannot be mapped to the crystal structures of ribosomes.

Author response image 5.Comparison of identified cross-links using a small database (62 proteins) and a large database (562 proteins).Filtering criteria: FDR < 5%, E-value < 0.00001, and spectral count ≥ 5.**DOI:**
http://dx.doi.org/10.7554/eLife.12509.098

C) We’ve deposited the raw data of the 70S ribosomes to http://www.huanglab.org.cn/donglab/RAW_6bio/.

*12) It is in the first paragraph of the subsection “Application of Leiker to lysates” that most of the inter-molecule cross-links suggest novel protein-protein interactions. Exactly how was this assessed?* We and others have demonstrated that inter-protein cross-links identified with high confidence generally indicate protein-protein interactions (Yang, et al., 2012; Herzog, et al., 2012; Liu, et al., 2015). This is well accepted in the field.

*13) I find Figure 5 to be largely uninformative. The authors must find a more informative way of presenting this or part of this data. It seems likely that the observed inter-molecule interactions are strongly skewed by the relative abundance of the proteins in the lysates. An assessment of this data in terms of the abundance of the proteins could provide valuable information on the rules underlying the observation of interaction in such complex mixtures. Depending on the results, it might also be valuable to discuss other possible reasons for why these particular interactions are observed.*

A) We have moved Figure 5 to Figure 5—figure supplement 3.

B) Yes, inter-molecular cross-links between high-abundance proteins are more readily observable than those of low-abundance proteins. In the *E. coli* whole-cell lysates, inter-protein cross-links are dominated by those of ribosomal proteins; when ribosome are removed (the ribo-free sample), cross-links of other proteins emerged (Figure 5). The 617 inter-molecular cross-links identified in the *E. coli* whole-cell lysate involve 341 proteins, 256 of which have their abundances determined in a recent study (Schmidt, et al., 2016). The number of inter-molecular cross-links observed is positively correlated with protein abundance (Figure 67).

Author response image 6.The number of inter-molecular cross-links observed for a protein in *E. coli* whole-cell lysates is positively correlated with the abundance of this protein.R, Pearson correlation coefficients; N, number of values.**DOI:**
http://dx.doi.org/10.7554/eLife.12509.099

*14) I could not make sense of the sentence “Applying the same procedure to the previously identified BS^3^ cross-links (Yang, et al., 2012), we obtained only three ribosomal proteins in the most highly connected modules (Figure 5—figure supplement 2)”, nor could I find Figure 5—figure supplement 2. Indeed I was unable to make proper sense of the second half of the subsection “Application of Leiker to lysates”. I also did not find the recitation of numbers in this section particularly useful or meaningful. The points that I think are ostensibly being made in this section need to be made much more clearly.*

We have replaced Figure 5 with Figure 5—figure supplement 2. The content of the subsection “Application of Leiker to lysates” has been revised to focus on the main point that Leiker is a good reagent for mapping protein-protein interaction networks.

*15) The data from the whole-proteome CX-MS analysis of C. elegans appears sparse (with just a few hundred crosslinks identified). What is the reason for this? (See also Q 16). Is this because of limitations of the software? If this is the case, perhaps it would help to use a smaller database containing only the top 1,000 most abundant proteins (since identifying crosslinks from the less abundant proteins would appear unlikely anyway)?*

A) To find out if this has to do with the database size, we repeated the search using two smaller databases, one containing only the top 1,000 most abundant proteins as the reviewers suggested, and the other containing 363 proteins for which intra-molecular cross-links had been identified in the initial pLink search. This resulted in a 10-20% increase in the number of cross-links identified (Figure 68), so the database size is a factor but not a critical one.

B) For the *C. elegans* samples, fewer replicate experiments were performed and the peptides were separated into fewer fractionations (Author response Table 2), accounting at least in part for the decrease of cross-link identifications.

C) Other possible explanations include the greater complexity of the *C. elegans* lysate and the dominance of vitellogenins and cytoskeletal proteins in the lysates.

Author response image 7.Comparison of identified cross-links using databases of different size.db9346, containing all the proteins identified in the sample; db1000, containing the top 1000 abundant proteins; db363, containing all the proteins for which intra-molecular cross-links had been identified. Filtering criteria: FDR < 5%, E-value < 0.01, and spectral count ≥ 3.**DOI:**
http://dx.doi.org/10.7554/eLife.12509.100

Author response table 2.Replicates of the *E. coli* and *C. elegans* lysates.**DOI:**
http://dx.doi.org/10.7554/eLife.12509.101SampleBiologicalReplicateTechnicalReplicateFractionation*E. coli* whole-cell lysates3210-11 fractions*E. coli* ribo-free lysates2310-12 fractions*C. elegans* whole-cell lysates128 fractions*C. elegans* mitochondrial proteins129 fractions

*16) While I agree with the authors that better sample preparation and more sensitive instrumentation could help to reduce the crosslinking problem caused by the huge dynamic range of proteins within proteomes, I am not convinced that these are actually the most important barriers to overcome for proteome-wide crosslinking profiling for at least two reasons:*

*A) Within cells, the majority of protein complexes are not abundant. Thus, using the current crosslinking protocol without prior affinity enrichment of the complexes of interest, the majority of such complexes will not be efficiently crosslinked because of the low efficiency for crosslinking low abundance complexes.*

We think that proteome-wide cross-linking are effective for high- and medium-abundance protein complexes if combined with extensive protein-level fractionation. Low-abundance protein complexes are always a challenge; even affinity purification may not succeed in every case.

Although many signaling protein complexes are of low abundance, many structural protein complexes (e.g. nuclear pore complex) and those that function in fundamental biological processes (e.g. RNA polymerase II complex, spliceosome, proteasome, exosome) are quite abundant.

*B) And even if one could crosslink the proteome efficiently using more efficient reagents, glueing the cellular components together would likely lead to daunting challenges in inducing the needed proteolysis.*

As demonstrated using ten standard proteins ([Supplementary-material SD2-data]), the cross-linking reaction of Leiker is as specific as that of BS_3_. We are not willing to increase reaction efficiency at the risk of reducing specificity. The proteolysis challenge may be solved by using multiple proteases as suggested by Leitner A et al. (Leitner, et al., 2012). Lys-C is a good choice as it allows digestion to be carried out in 8 M urea, which unfolds proteins and exposes more polypeptides to digestion.

*17) Figure 8. Reliable quantitative CX-MS analysis can be very challenging due to the low abundance of crosslink ions. What is the reproducibility of the quantitative data? Error bars from the repeat experiments would help here. Indeed without stronger quantitative data, it is not justified to entitle the paper: "Leiker, A Cross-Linker for Mapping Protein-Protein Interaction Networks and comparative cross-link analysis" at least with respect to the "aspect of comparative cross-link analysis".*

A) The experiment for Figure 8 (log phase vs. stationary phase *E. coli* lysates) was performed only once, but the forward and the reverse labeling experiments serve the purpose of biological replicates. Between the forward and reverse experiments, the Pearson correlation coefficients on inter- and mono-links are 0.55 and 0.60, respectively, suggesting that the two experiments are generally consistent with each other.

B) For L7Ae, three biological replicates were performed for both forward and reverse labeling experiments. The quantification results of each replicate are shown in [Supplementary-material SD11-data]. Shown below is a scatter plot with error bars for this data set (Figure 69).

Author response image 8.Abundance ratios of mono-links (F/B) in the forward (F_[d0]_/B_[d6]_) and the reverse labeling experiment (F_[d6]_/B_[d0]_).Mean ± SEM.**DOI:**
http://dx.doi.org/10.7554/eLife.12509.102

Figure 69

[Editors' note: further revisions were requested prior to acceptance, as described below.]

The manuscript has been improved but there are some remaining issues that need to be addressed before acceptance, as outlined below: Concerning the title, it is not acceptable to us that you use the acronym LEIKER (Lysine-targeted enrichable cross-linker) in the title or Abstract of the paper. We note that you have now acceded to the request to properly attribute the 2003 JACS article, which previously described trifunctional reagents with the same basic properties as the present trifunctional reagent. We note that you still wish to name your process by an acronym (Lysine-targeted enrichable cross-linker (Leiker) even though this was thought to be inappropriate by the reviewers. The reviewers' objection to the acronym was that in giving a new name to a previously published strategy, you might effectively take possession of an idea that did not originate with the present work. Therefore, in order for this work to be published in eLife, you may use the acronym LEIKER in the main body of the paper, but not in the title or the Abstract.

We have revised the title to “Trifunctional Cross-Linker for Mapping Protein-Protein Interaction Networks and Comparing Protein Conformational States”, and removed the acronym “Leiker” from the Abstract.